# Modeling attention and binding in the brain through bidirectional recurrent gating

**Saeed Salehi** [1,2,3] ✉, **Jordan Lei**[4], **Ari S. Benjamin** [5],
**Klaus-Robert Müller** [1,2,6,7] **& Konrad P. Kording** [8,9] ✉

Attention is a cornerstone of cognition and neural computation, enabling the brain to select relevant information, bind features into coherent objects, and guide behavior. However, we currently lack a unifying computational model that connects the diverse phenomena of attention, from spatial and feature-based selection to object-based binding, within a single, neurally plausible computational framework. Here, we propose a bidirectional recurrent gating mechanism integrated into a principled architecture of the ventral visual stream. In this architecture, feedforward pathways extract visual features, while top-down and lateral connections transmit context- and task-dependent modulatory signals that control information flow. We demonstrate that our model, trained on recognition and segmentation problems, successfully performs the canonical attention tasks of orienting, filtering, and visual search on complex scenes. It replicates key psychophysical phenomena, such as perceptual load and inattentional blindness, while its internal units develop neural properties consistent with primate physiology, including multiplicative gain modulation and border-ownership coding. Our work provides evidence that this diverse set of attentional and binding phenomena can emerge from error-backpropagation combined with architectural constraints upon information flow, offering a powerful tool for neuroscience and a compelling, bio-inspired alternative to standard AI architectures.

Attention is widely seen as a mechanism by which the brain selects a meaningful subset of incoming stimuli for perception, learning, or memory[1,2]. Empirical research reveals that selectively focusing on portions of the visual scene produces neural responses that are more localized, sparser, and less noisy[3–6]. In the context of learning, behavioral evidence shows that focusing on task-relevant stimuli improves generalization and performance[7–9]. Similarly, in memory, attention facilitates efficient encoding of information, supports the formation of structured representations, and plays an integral role in working memory[10–12]. The empirical evidence suggests that attention is indispensable to neural

information processing, making it a foundational construct in our understanding of learning, memory, and perception.

Understanding attention is particularly challenging because it encompasses a wide range of complex phenomena across scales and modalities[13] (also see ref. 14). One well-studied form is spatial ("spotlight") attention, in which focus is directed to a region of space. Behavioral experiments demonstrate that perception within the attended region tends to be faster and more accurate[6,7,15]. Feature-based attention, another extensively studied variant, leverages prior knowledge about relevant stimulus features to enhance perception

[1]Machine Learning Group, Technical University of Berlin, Berlin, Germany. [2]BIFOLD—Berlin Institute for the Foundations of Learning and Data, Berlin, Germany. [3]BCCN—Bernstein Center for Computational Neuroscience, Berlin, Germany. [4]New York University, New York, NY, USA. [5]Cold Spring Harbor Laboratory, Cold Spring Harbor, NY, USA. [6]Department of Artificial Intelligence, Korea University, Seoul, Korea. [7]Max Planck Institute for Informatics, Saarbrücken, Germany. [8]Department of Bioengineering, Department of Neuroscience, University of Pennsylvania, Philadelphia, PA, USA. [9]CIFAR LMB Program, Toronto, ON, Canada. ✉e-mail: ai.neuro.io@gmail.com; kording@upenn.edu

and visual search[16–20]. Object-based attention, arguably related to feature-based attention, involves the holistic selection of objects, including all their associated spatial and visual features[21–23]. A fundamental question here is how an object that is represented by activity across millions of neurons and many cortical regions can be perceived and attended to as a single entity[24]. This is known as the binding problem, which refers to how the brain integrates activity patterns related to the same object and separates them from others[25–29]. Understanding object-based attention is thus viewed as a critical step toward addressing the broader problem of feature binding in neuroscience[30,31] (for a contrasting view, see ref. 32).

Mechanistically, attention is believed to rely on a large set of interacting, complex processes. The first is feedforward (bottom-up) processing, which progresses from lower to higher areas of the visual cortex through hierarchical, mainly unidirectional connections, and establishes the foundational tuning and receptive field properties of visual neurons[33]. The second is the top-down process, wherein higher-order brain regions modulate activity in early visual areas, a mechanism shown experimentally to influence object perception, integration, and biased competition[33–39]. The third is lateral processing, which enables context-sensitive interactions such as perceptual grouping, contour integration, and noise reduction[40–43]. The fourth essential mechanism is recurrence, or iterative processing, which allows information to propagate bidirectionally across layers and time. Experimental and theoretical studies alike have emphasized the importance of recurrence in visual processing and attentional dynamics[31,44–47]. The fifth is divisive normalization, which is believed to be a canonical computation governing biased competition and attention[48–50].

Over the past several decades, numerous computational models of attention have been proposed, each offering different perspectives on how attention may be implemented in the brain. Computational models of visual attention from neuroscience can be broadly grouped into four paradigmatic hypotheses[51]: selective routing, saliency map, temporal tagging, and emergent attention. Selective routing models conceive of attention as a controlled routing of information through the visual hierarchy (i.e., gating signals that direct a subset of inputs to higher areas)[52–54]. Saliency map models compute attention by integrating multiple feature maps into a topographic saliency map, where the most conspicuous location wins the competition for attentional selectionthrough winner-take-all mechanism[55,56]. Temporal tagging models exploit time dynamics (oscillations or synchrony) to bind or select attended stimuli[57–61]. In contrast to all of the above, emergent attention models posit that attentional phenomena arise intrinsically from task-driven competitive interactions within a large neural system rather than from a single explicit mechanism[62–64]. Although empirical evidence favors each of these hypotheses to varying degrees[65–67], there is not yet proof that any one such model can account for the entirety of attentional phenomena observed in the literature.

In this study, we introduce bidirectional recurrent gating, a neuro-inspired and biologically plausible mechanism for attention and binding. We also propose a model inspired by the ventral visual pathway, which allows integration of the bidirectional recurrent gating mechanism into a principled architecture that includes bottom-up, top-down, and lateral processing and enables multitask learning. Our modeling approach aims to evaluate the emergent-attention hypothesis that a neural network with the right architecture and complexity, equipped with the fundamental components of attention[68], and trained on reasonable objectives would exhibit "attentive behavior." It is important to emphasize that our objective is not to improve on artificial neural networks per se, but to use them as computational instruments for exploring attention mechanisms inspired by the brain[69,70]. Our architecture adapts the U-Net framework[71] by incorporating biologically motivated motifs: divisive normalization[72] approximated via layer normalization[73], working memory using a dense recurrent layer[74–76], attention-driven neuromodulation, and top-down context inputs.

In our results, we systematically validate our proposed architecture through a comprehensive set of experiments. First, we establish the behavioral potential and flexibility of our multitask framework across a wide range of attention-focused tasks. These include object recognition, cued perceptual grouping, symbolic orienting, visual pop-out, figure-ground separation, inhibition of return, and top-down visual search—tasks that collectively cover the three canonical behavioral axes of attention: orienting, filtering, and searching[77]. Critically, we find that the proposed attention mechanism significantly improves both learning speed and validation accuracy in object recognition. Then, we bridge the gap to human perception by showing how our model qualitatively reproduces established psychophysical effects including attentional contrast gain, perceptual load effect, and inattentional blindness. Next, we look inside the model to assess its neural plausibility, presenting evidence that its internal units exhibit properties analogous to attention-invariant tuning and attention-driven neuromodulation observed in the brain. Finally, we analyze its neural representations and find key parallels with neurophysiological data from border-ownership coding and a plausible explanation for figure-ground separation. Given the extent of these findings, many experimental details and additional results are provided in the supplementary materials. Ultimately, this confluence of behavioral, psychophysical, and neurophysiological evidence supports our central claim that the proposed mechanism and architecture offer a unifying computational account for many hallmarks of biological attention.

## Results

Our architecture consists of a feedforward, contracting feature pathway that hierarchically processes and extracts learned features, alongside a top-down, expansive attention pathway that combines lateral and top-down connections to implement a task- and stimulus-dependent bidirectional recurrent gating mechanism (Fig. 1a). The lateral connections form a long-range bidirectional recurrent network, while the dense recurrent layers provide the model with working memory, enabling it to perform memory-dependent tasks and behaviors such as inhibition of return. Since recurrence is central to our architecture, the model requires a sequence of images as its input stimulus. Correspondingly, it outputs a sequence of attention maps and label predictions, generated sequentially in each iteration (Fig. 1c). In addition to the input stimuli, the network can also receive additional inputs, such as the task index for multitask learning and prompts for top-down tasks (e.g., visual search). Altogether, our model can effectively learn to perform multiple tasks on the same architecture (Fig. 1b) (see Methods section and Supplementary Materials for detailed architectures and single- versus multi-task analysis).

We use error backpropagation and gradient descent[78] to train the model on a combination of two objectives: (a) Object classification, optimized using cross-entropy (CE) loss, and (b) Object segmentation in the input layer, optimized using mean squared error (MSE) loss. Although these two objectives may initially appear limited for training on complex tasks, we show that the multitask learning paradigm enables the model to learn to attend to relevant features even under partial supervision. We refer to training as fully supervised when both classification and segmentation losses are used; otherwise, it is partially supervised. For regularization, we apply $L2$-norm (i.e., Ridge regularization) penalty on all learnable weights. During inference, we report classification accuracy and pixel error from the final iteration (see Methods section). Unless stated otherwise, the reported outputs for each experiment correspond to the features, predicted class label, and attention map at the last iteration (Fig. 1c).

### Behavioral results and multitask learning

Visual attention research encompasses an extensive range of behavioral findings[2,77], including: object recognition in cluttered scenes,

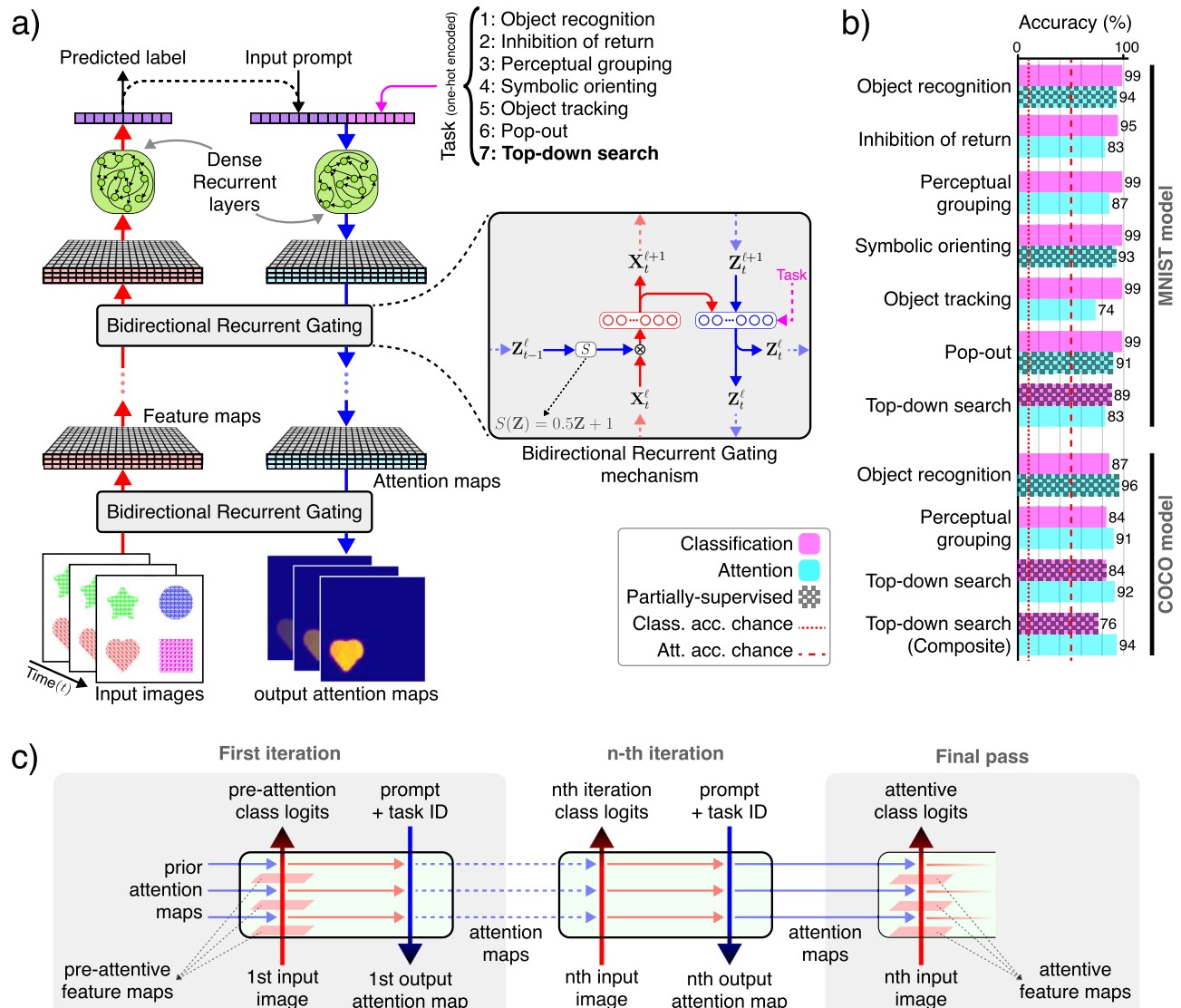

**Fig. 1 | Model architecture and multitask learning. a** Bidirectional recurrent gating is the core block of our model. The network comprises two main pathways: the bottom-up feature pathway, which hierarchically extracts learned feature representations (shown in red); and the top-down attention pathway, which combines top-down information, the task signal, and feature maps to generate attention maps (shown in blue). The attention maps consequently and multiplicatively modulate the feature maps of the next iteration. The two pathways meet at the bottleneck, which incorporates the dense recurrent and linear layers and outputs predicted labels (logits). In addition to the bottleneck, the two pathways communicate feature maps and attention maps through the lateral connections. The box shows the signal flow of the bidirectional recurrent gating mechanism. Subscripts indicate the iteration and superscripts denote the layer. The feature map $\mathbf{X}_t^\ell$ is multiplicatively modulated by the affine-scaled attention map $\mathbf{Z}_{t-1}^\ell$ from the previous iteration before going through a convolutional layer. The output of layer $\ell$, $\mathbf{X}_t^{\ell+1}$, is then passed to the corresponding layer in the attention path and concatenated with the attention map $\mathbf{Z}_t^{\ell+1}$. The concatenated signals are modulated by the task-embedding, when applicable (e.g., in multitask settings). The attention block then creates the attention map $\mathbf{Z}_t^\ell$ for the next iteration. **b** Our architecture enables effective multitask learning on both simple (i.e., digits from the MNIST dataset) and complex (i.e., animals from the COCO dataset) stimuli. Here we show the classification and attention accuracy for the two models trained on COCO and MNIST compositions. The red dotted line shows the chance level for classification accuracy (10%); the red dashed line marks the chance level for attention accuracy (50%). For some tasks, we use partially supervised training, meaning that we provide only one supervision signal during training, either target attention maps (i.e., segmentation) or target labels (i.e., classification). **c** Pre-attentive and attentive features. The model processes the input images over multiple iterations. In the first iteration, it receives flat, task- and input-agnostic attention maps, resulting in feature representations and class predictions that are considered "pre-attentive". In subsequent iterations, the model incorporates context and task information into attention maps, producing "attentive" features and predictions.

spatial cueing of attention and perceptual grouping, feature saliency and pop-out, top-down control, and inhibition of return, to name a few. In what follows, we provide a series of multitask experiments to examine whether our network is able to replicate these known behavioral findings. The first experiment, arguably using toy data, is performed on compositions of handwritten digits from the MNIST dataset[79], with various aspects extended to enable a broad set of tasks. The second experiment is conducted on a rather complex set of stimuli based on the COCO dataset[80], though on fewer number of tasks (see the Supplementary Materials for single-task training results).

Object recognition is one of the primary objectives of the visual system, which requires separating the target object from the background and foreground, and classifying it accurately. Although most artificial models of object recognition today rely primarily on feedforward processes, evidence from neuroscience suggests that attention and recurrence play a critical role[33,81]. Attention and recurrent

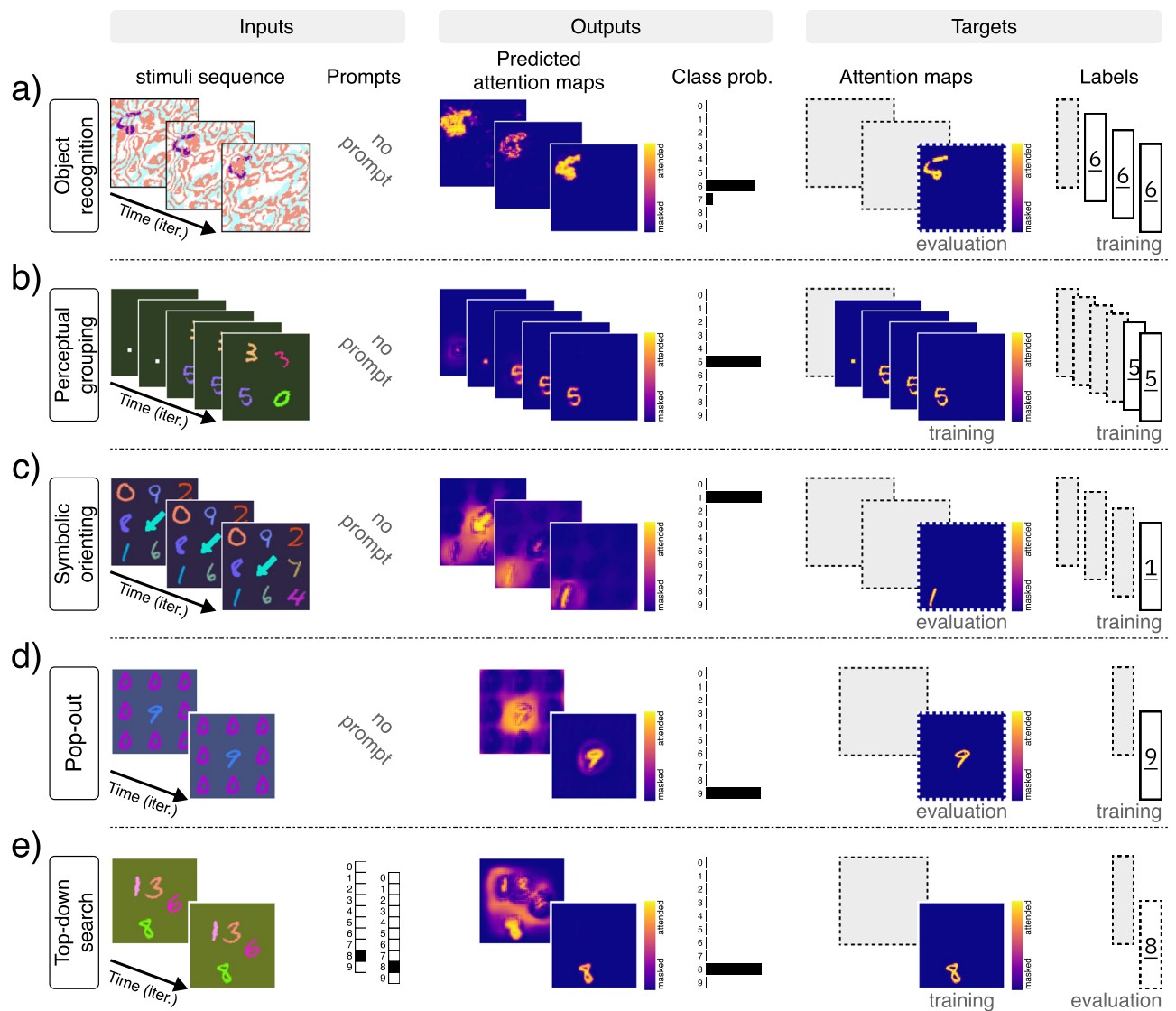

**Fig. 2 | Multitask training on MNIST composites (Part 1/2).** Results for a single model trained on seven tasks simultaneously. The figure includes input and output signals, as well as the target signals (i.e., the desired outputs of the model). If the target signal is used during training, it is marked by "training" subscript (e.g., attention maps for the top-down search task), otherwise marked by "evaluation" and framed by dashed outlines (e.g., class label for top-down search). Here we present the results for: **a** object recognition, **b** perceptual grouping using spatial cue, **c** orienting via symbolic cue, **d** pop-out saliency, and **e** top-down visual search. Note: colors for the input sequence in task (**a**) have been inverted to improve visualization. Digit images are adapted and modified from the MNIST dataset ©LeCun, Cortes, and Burges[79], available under a CC BY-SA 3.0 license.

processing can help resolve ambiguity at object boundaries, particularly where edges are shared between the target and other elements in the scene, ultimately improving accuracy and robustness[82,83] (in support of this claim, we show in the Supplementary Materials that incorporating attention into a model significantly accelerates learning and improves robustness in a composite object classification task based on CIFAR-100[84]). Moreover, the visual system tends to maintain a stable internal representation of the object, even when the foreground or background moves. To test whether our model can solve these problems, we designed an object recognition and permanence task, where a digit is placed on top of a background pattern and partially obscured by a foreground pattern (Fig. 2a). Both the background and foreground are randomly generated correlated noise patterns that move slowly across the image, while the object remains stationary. Training for this task includes only the target label, requiring the network to learn to uncover a meaningful attention map solely through classification loss and the multitask paradigm. Our model achieves 99% classification accuracy and a pixel error of 0.009 in recovering the

correct attention map in the final iteration on the test dataset ($n = 10,000$ samples). This demonstrates that the model can effectively solve object recognition, segmentation, and permanence tasks in the presence of significant levels of correlated noise and occlusion.

Spotlight attention is a well-studied model system for attention[6,15,56]. A common element in cognitive visual experiments is the use of a cue to spatially orient attention to a region and ask the subject to fixate on the cue and follow the instructions. Such cues can also be used to guide perceptual grouping[85,86]. Therefore, we tested our model to see if it could also learn where to direct its attention using spatial cues (Fig. 2b). This was crucial, as we intend to use a similar approach in the curve-tracing task later. Here, we trained the network to first attend to a cue (i.e., a dot) appearing in the first two iterations. The sequence is then followed by a random arrangement of a few digits, and the network must attend to the digit at the previously cued location. Our model achieves 99% classification accuracy and a pixel error of 0.002 in recovering the correct attention map in the final iteration on the test dataset ($n = 10,000$ samples). Additionally, we

show in the Supplementary Materials that incorporating attention and spotlight cueing during training can accelerate learning and enhance task performance. These results demonstrate that our model possesses spotlight attention capabilities qualitatively consistent with human subjects in similar tasks.

Apart from spatial cues, we also use symbolic visual cues to direct our attention to an object in the scene[87]. The most prominent example is pointing with the index finger. We have learned to first look at the finger, recognize the direction to which it is pointing, and then to orient our attention to the object in line with the finger's direction. Accordingly, we trained our model on a task where 8 digits are arranged around an arrow and the goal is to locate and classify the target digit (Fig. 2c). During training, only information about the class of the target digit to which the arrow points is provided. Since no attention map for the arrow is given, the model has to learn about the arrow only through classification loss and the multitask paradigm. Our model achieves 99% in classification accuracy and a pixel error of 0.021 on the attention map in the final iteration on the test dataset ($n$ = 10,000 samples). Interestingly, the results show that the model attends to the arrow first, to infer the location of the target digit (Fig. 2c). This suggests that it has learned, perhaps through other tasks, to treat digits and arrows as distinct visual entities, and therefore avoids learning the combination of all the arrow directions and digits, resulting in more robust behavior. Thus, the system can learn about and orient its attention to an abstractly cued location and use the multitask paradigm to improve generalization.

Salient features in a scene are thought to be the driving factors in pre-attentive processing[88–90], although the neural basis of this phenomenon is still debated[91–93]. We implemented a task in which the model is presented with a grid of nine digits, where all but one are of the same instance and color (Fig. 2d). The model is trained to return the class of the digit that is different from the rest, without explicit training on the target attention map. The network successfully learns to detect and classify the salient digit with 99% classification accuracy and a pixel error of 0.017 in recovering the correct attention map ($n$ = 10,000 samples), thus confirming that our network can learn to perform feature-based pop-out tasks.

So far, attention has been guided by the input visual stimulus and mediated by the task signal. However, it is often the case that we are given an explicit non-visual cue or description (i.e., a prompt) specifying what to search for in a cluttered scene[94]. Because the prompt specifies the class or other distinguishing attributes of the object, the model must output the location of the prompted object. Of the many kinds of visual search, here we focus on visual search through top-down attention. To demonstrate how our model can perform top-down search, we use a composition of digits as the visual input and a target class label as the input prompt that is fed to the bottleneck (i.e., the top) (Fig. 2e). The model is trained to attend to (i.e., segment) the prompted digit in the image. The model achieves 83% attention accuracy, 89% classification accuracy, and a pixel error of 0.002 in recovering the correct attention map in the final iteration ($n$ = 10,000 samples). Although here we restricted our prompt to class labels, we have also experimented successfully with combinations of other search attributes such as color and texture (see Supplementary Materials).

A model of attention should also be able to attend to an object as it moves through the scene. Spatial attention and awareness of the object's location have been shown to be important to the binding process[27]. Here, we trained the network to track and classify a moving digit (Fig. 3a). The task is to fixate on and classify a stationary single digit for two iterations, and then to track the same digit as it begins to move through the scene with multiple distractor digits. Training for this task is fully supervised, with target labels and masks provided during training. The network achieves 99% classification accuracy and a pixel error of 0.009 in recovering the correct attention map on the

test dataset ($n$ = 10,000 samples). Although the network was trained for only seven iterations of tracking, it can track the target digit for a longer period of time during inference, indicating that the network can learn to track an object through space and time.

When humans attend to multiple objects in a visual scene, they typically do so sequentially, moving from one object to the next[95]. A large body of experimental evidence supports the idea of inhibition of return (IOR), whereby recently attended regions of a visual input are inhibited, allowing the subject to move from one object to the next[56,96–98]. Without inhibition of return, a subject would remain fixated on the most salient object. Here, we demonstrate that our model can learn to perform IOR on a multi-digit composite using only implicit training (Fig. 3b). Our experiment is as follows: A sequence of identical compositions of $d$ augmented digits from the MNIST dataset is fed into the model. The network learns to arbitrarily select, attend to, and classify any of the digits in $k$ given iterations. It must then move on to the next digit, and avoid returning to previously attended digits. We trained the network in a fully supervised manner, simultaneously on $d$ = 1, 2, and 3 digits with $k$ = 2 iterations per digit. Our model achieves 95% accuracy in classifying all the digits in the scene and a pixel error of 0.004 in recovering the correct attention for every digit ($n$ = 10,000 samples). This demonstrates that the network can learn to cycle through objects in a multi-object scene.

**Bregman's perceptual illusion**. Shared perceptual illusions between humans and artificial neural networks can inform us about the similarity of the underlying processes that are otherwise inaccessible to us[99]. Here, we investigate whether our model can account for a well-known perceptual illusion for which the model has not been explicitly trained. Researchers have long known about the existence of the Bregman perceptual illusion, in which a visible occlusion can provide cues that influence object perception. In the Bregman illusion, a stimulus such as a letter is occluded by an ink blot (i.e., distractor) (Fig. 4a). When the distractor is removed, leaving a blank region, the stimulus appears fragmented and difficult to identify (Fig. 4b). However, when the distractor is present, the fragments are contextualized as objects occluded by a foreground distractor, making the entire scene more interpretable (Fig. 4a). The Bregman illusion is a classic demonstration of border-ownership in Gestalt psychology and perception[100–103].

To investigate this illusion, we start with the aforementioned model already trained on the MNIST tasks. It is important to note that we did not explicitly train or fine-tune the model on this illusion; we only manipulate the stimuli used in the object recognition task (i.e., the model is evaluated zero-shot on these variants) (Fig. 4c–e). In the original task, the background and foreground were moving (i.e., dynamic) and the model could slowly build a better representation of the object by integrating the information. Therefore, we expect the classification and attention accuracy to suffer when the foreground is made stationary (i.e., static). The question is whether removing the foreground, leaving a blank overlap, would result in a change in classification and attention accuracies. To further test whether the observed behavior is simply due to an out-of-distribution effect, we created a new set of control stimuli, distinct from the training set, that maintain a similar ratio of visible to invisible object regions (Fig. 4e). Since the invisible occlusion in the control set is organized as a grid pattern, we do not expect lower classification accuracy. Our hypothesis is that, despite the invisible occlusion, the grid structure introduces less fragmentation ambiguity and clearer boundaries than the irregular invisible occluder in the test condition. Our analysis suggests that the network does indeed perform better at the recognition task when the occlusion is visible rather than when it is invisible, which is consistent with the experimental findings of border-ownership (Fig. 4f). It also shows that the presence of visible occlusion has an initial negative impact on classification, but the network has learned to use attention to extract boundary information from the

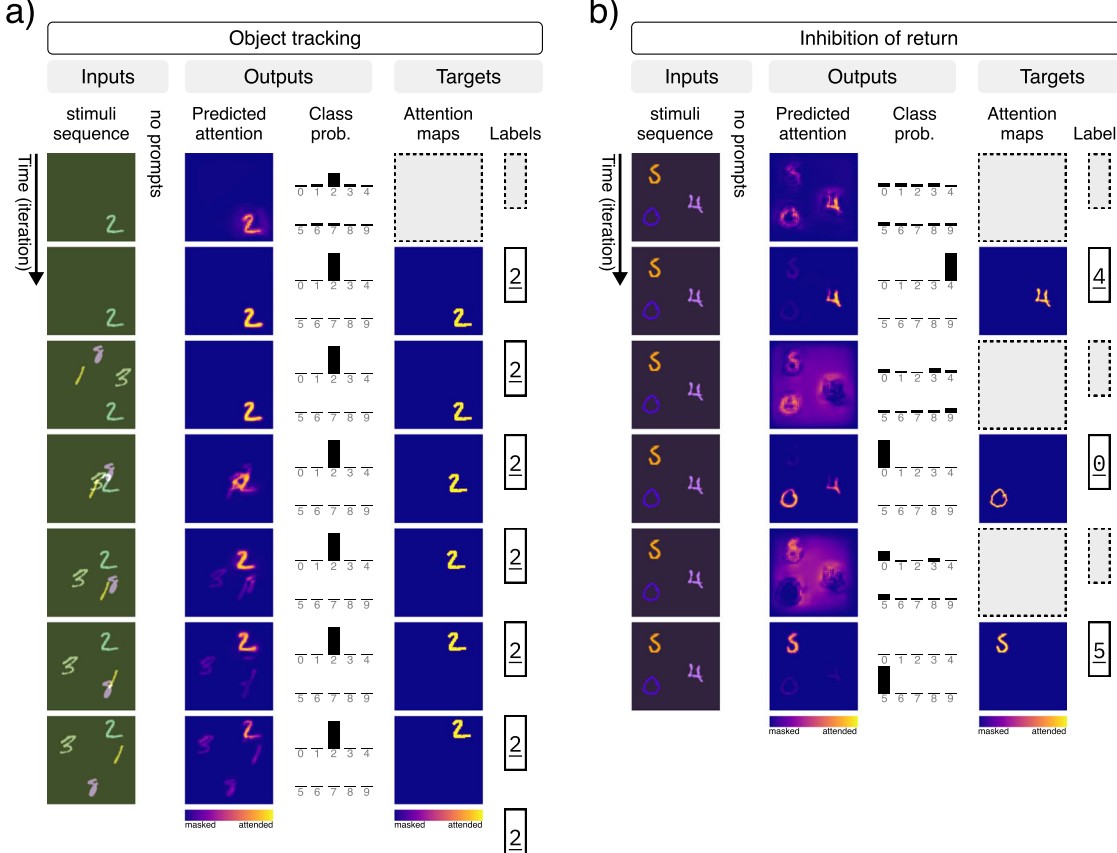

**Fig. 3 | Multitask training on MNIST composites (Part 2/2).** Results for a single model trained on seven tasks simultaneously. The figure includes input and output signals, as well as the target signals (i.e., the desired outputs of the model). The two tasks presented here are trained on both target signals (i.e., attention maps and class labels). Here we present the results for: **a** object tracking, and **b** inhibition of return (IOR). Digit images are adapted and modified from the MNIST dataset ©LeCun, Cortes, and Burges[79], available under a CC BY-SA 3.0 license.

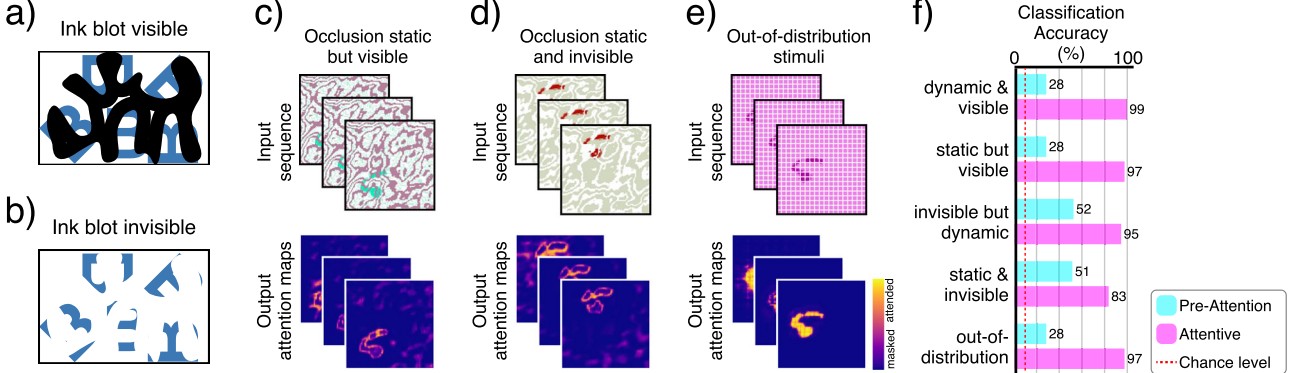

**Fig. 4 | Bregman's illusion.** The Bregman illusion is commonly used to demonstrate the concept of border-ownership. We refer to the occlusion as visible when the occluder is present; and invisible when it is removed, leaving a blank region. **a, b** Modified version of Bregman's illusion[101], illustrating that the visible ink blot (i.e., occlusion) helps with recognition of the letters. **c, d** Similarly, visible occlusion in our experiment seems to help with the recognition task and in recovering the digit's boundaries. **e** Out-of-distribution control stimuli, where the background is a random color, and the invisible occlusion is organized as a grid pattern rather than noisy splotchy texture. **f** Although visible occlusion appears to initially (i.e., pre-attentively) hinder classification accuracy, attention helps the model to integrate occlusion boundaries to achieve higher performance. Note: Colors for all input sequences have been inverted to improve visualization. Digit images are adapted and modified from the MNIST dataset ©LeCun, Cortes, and Burges[79], available under a CC BY-SA 3.0 license.

occlusion and build a better representation of the digit (Fig. 4f). Furthermore, the high classification accuracy on the control stimuli supports our claim that the illusion is not simply the result of unfamiliar input, but instead arises from the absence of expected structural features. We will further investigate border-ownership in the next section.

Although the MNIST dataset provides us with the freedom to design diverse tasks and evaluate the range of our model on multitask learning, the resulting compositions are far from those we encounter in nature. To further evaluate our model on natural stimuli, we train a scaled-up model on the animal subset of the COCO dataset[80]. We chose

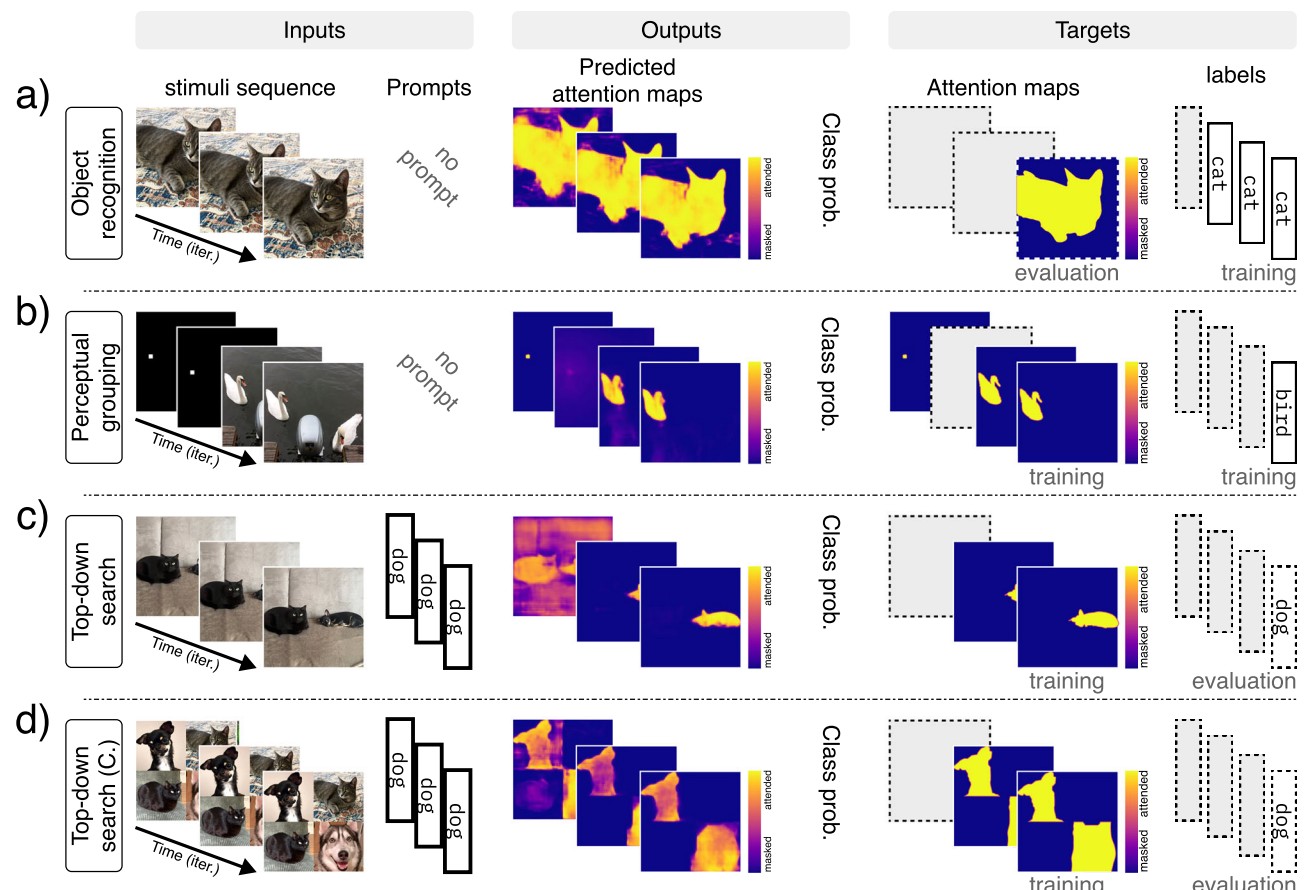

**Fig. 5 | Multitask training on COCO.** Results for a single model trained on three tasks simultaneously: **a** object recognition, **b** cued perceptual grouping, **c** top-down visual search, and **d** top-down visual search on compositional images. The figure includes input and output signals, as well as the target signals. If the target signal is used during training, it is marked by "training" subscript (e.g., attention maps for the top-down search task), otherwise marked by "evaluation" and framed by dashed outlines (e.g., class label for top-down search).

the COCO dataset because it contains fairly complex stimuli, but also includes the ground-truth segmentation maps. Here, we devised three tasks on which to evaluate our model: object recognition, perceptual grouping, and top-down visual search (Fig. 5). For the object recognition task, we used compositions of single animals placed on top of sliding background images from the BG-20k dataset[104] (Fig. 5a). For the perceptual grouping task, we used a regime similar to the MNIST spatial cueing task, where initially a visual cue is shown to indicate the target object (here, the target animal) (Fig. 5b). For the top-down visual search task, we used two sets of stimuli: the augmented natural images from MS-COCO (Fig. 5c), and compositions of four distinct animals into one image (Fig. 5d). Since most images contain only a single animal or multiple animals of the same species, we used the four-animal compositions to generate new multi-object samples to further improve training. Our model achieves 87% and 84% classification accuracy on the object recognition and perceptual grouping tasks, respectively. For object recognition, attention accuracy is 96% and a pixel error of 0.027; for perceptual grouping, 91% and 0.021; and for top-down search, 92% and 0.024. The test set contains 1,558 samples for recognition and perceptual grouping, and 646 samples for top-down search.

**Feature attention and masking**

One challenge in object recognition is the presence of spurious pre-attentive features in the scene. For example, most waterbirds are photographed near a lake or sea, which adds a blue background as a feature to the images. Studies in both human and artificial networks show that pre-attentive features like texture and color can lead to incorrect feature integration and misclassification[105–107]. Attention can help separate spurious and true correlations by masking the irrelevant features and amplifying the informative attributes.

To evaluate this theory, we use the CelebA dataset, which contains images of celebrity faces and 40 binary attributes (e.g., hair color and sex) per image[108]. The images are cropped such that the facial landmarks are approximately aligned across the dataset. Among the attributes in this dataset, there are some highly correlated features; namely, most celebrities with blonde hair color are labeled as female (Fig. 6a). Therefore, a naively trained neural network could use the blonde hair color as a low-level proxy for sex[109]. This makes CelebA an interesting dataset to test whether our model can learn to attend to the right features for classification.

To this end, we train our model on CelebA for a binary sex classification task on sequences of length two, with no target mask provided and without using class weights to address the class imbalance (see Supplementary Materials for the extension of this task to localized attributes). The results on the test dataset ($n = 8082$ samples) suggest that although pre-attentive features (i.e., hair color) remain predictive of sex, the network uses attention to integrate the relevant features while masking out the hair to achieve higher accuracy (Fig. 6b–d). This could have important implications for how networks should learn to deal with spurious correlations, since simply removing the spurious features has been shown to be problematic[110].

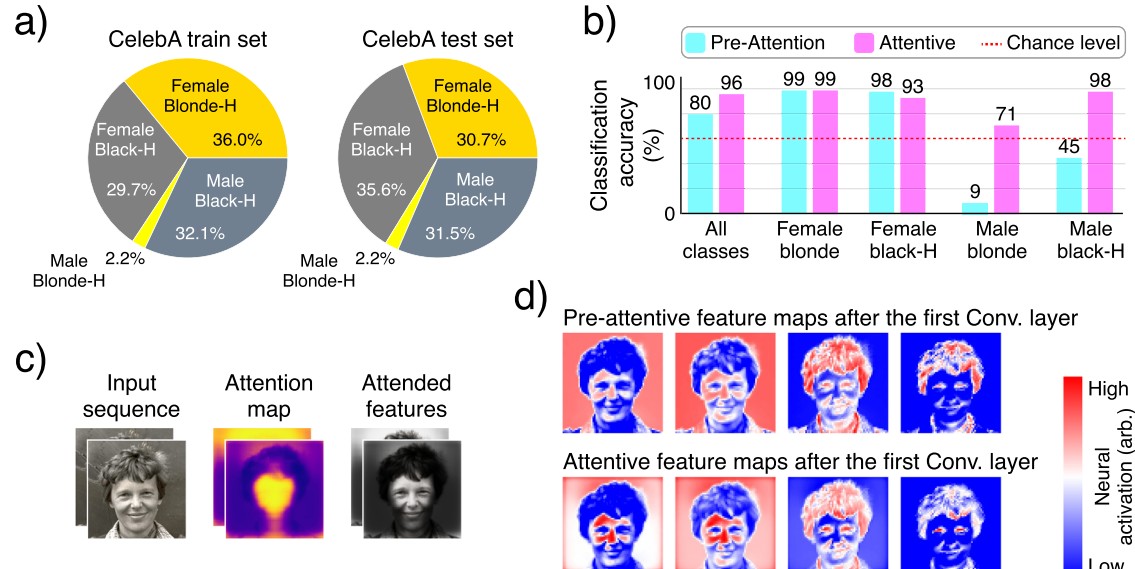

**Fig. 6 | Feature attention and masking. a** The CelebA dataset contains a strong spurious correlation between hair color and sex, which can be exploited by naive classifiers as a shortcut for sex classification. **b** Classification accuracy per class and feature. Pre-attentive results indicate that our model has not learned to correctly classify the expected sexes through the feature path alone, but instead uses recurrent attention to integrate the correct features. **c** Trained purely on sex classification, our model has learned to attend more to facial features and less to other attributes such as hair color. **d** Feature maps in the feature path before and after attention show the switch from hair attributes to facial features through attention. This figure includes a modified portrait of Amelia Mary Earhart captured by Underwood & Underwood provided by Smithsonian Institution under CC0 license.

## Psychophysical results

Psychophysical studies provide a powerful framework for characterizing attentional mechanisms and their influence on perception[6], making them critical for evaluating the biological relevance of computational models[111]. Drawing on this empirical foundation, we designed three experiments: contrast detection, contrast discrimination, and orientation change detection to test whether our model's behavior aligns with established human findings. For all experiments, we employed a spatial cueing paradigm. The model was presented with a 3 × 3 grid containing Gabor patches as target and distractor stimuli, and Gaussian patches as spatial cues to orient attention (Fig. 7). To mimic peripheral dynamics, targets appeared only in the eight off-center locations (Fig. 7a). While the same base architecture was used for all experiments, each model was trained independently on its specific task. For analysis, we fitted a Sigmoid function to the data and report 50% thresholds ($th_{50\%}$). Detailed specifications of the training procedures and stimulus parameters are available in the Methods section. For all experiments, the number of samples per condition is $n = 1024$.

**Attention enhances contrast sensitivity.** A key finding in psychophysics is that transient covert attention can enhance performance in visual discrimination tasks[112]. This enhancement manifests as a decrease in the contrast threshold required for perception, resulting in a leftward shift of the psychometric function (Fig. 7b). This effect is considered a signature of the "contrast gain" mechanism of attention[113]. To test whether our model exhibits this behavior, we designed a cued localization task. In each trial, a transient cue oriented the network's attention to a random location (Fig. 7c). Subsequently, a Gabor patch with variable contrast appeared in 50% of trials at one of the peripheral locations, independent of the cue's position. The model was trained to output the location label of the Gabor patch if present, or signal its absence (i.e., output the null label, which is class index 4). We then constructed psychometric functions by measuring performance as a function of target contrast, comparing trials where the target appeared at the cued (valid) location versus at an uncued

(invalid) location. Our results replicate the established psychophysical findings of contrast gain. As shown in Fig. 7d, attention markedly improved the model's performance. Specifically, the psychometric function for the attended condition ($th_{50\%} = 0.21$) is shifted to the left compared to the unattended condition ($th_{50\%} = 0.25$). This demonstrates a clear reduction in the contrast threshold, consistent with the contrast gain mechanism reported in humans[112].

**Attention enhances perceived contrast.** To test if our model accounts for attention's influence on perception, we sought to replicate the seminal work of Carrasco, Ling, and Read (2004)[114]. Their study demonstrated that transient covert attention increases the perceived contrast of a stimulus, a phenomenon quantified by the Point of Subjective Equality (PSE) (corresponding to the 50% point on the psychometric function). We implemented a cued, contrast comparison task. In each trial, a spatial cue was presented, followed by two Gabor patches of varying contrast at different locations (Fig. 7e). The model's objective was to identify which of the two patches had a higher contrast. To measure the 50% threshold during evaluation, we designated one Gabor as the "standard" (with a fixed contrast of 0.1 arb.) and the other as the "test" (with variable contrast). We then assessed the probability that the model would select the test patch as having higher-contrast, both when the test patch itself was cued and when the standard patch was cued. As shown in Fig. 7f, attending to the test patch systematically increased the likelihood that it was reported as having higher contrast (standard-cued: $th_{50\%} = 0.11$, neutral-cued: $th_{50\%} = 0.10$, and test-cued: $th_{50\%} = 0.09$). This indicates that attention increases perceived contrast for the attended stimulus, even when it has objectively lower contrastive than the unattended stimulus. This outcome, which demonstrates that attention enhances apparent contrast in our model, is consistent with the original findings of Carrasco et al. (2004) (compare with Fig. 7g).

**Perceptual load and inattentional blindness.** Two foundational concepts in attention research are perceptual load[115] and inattentional

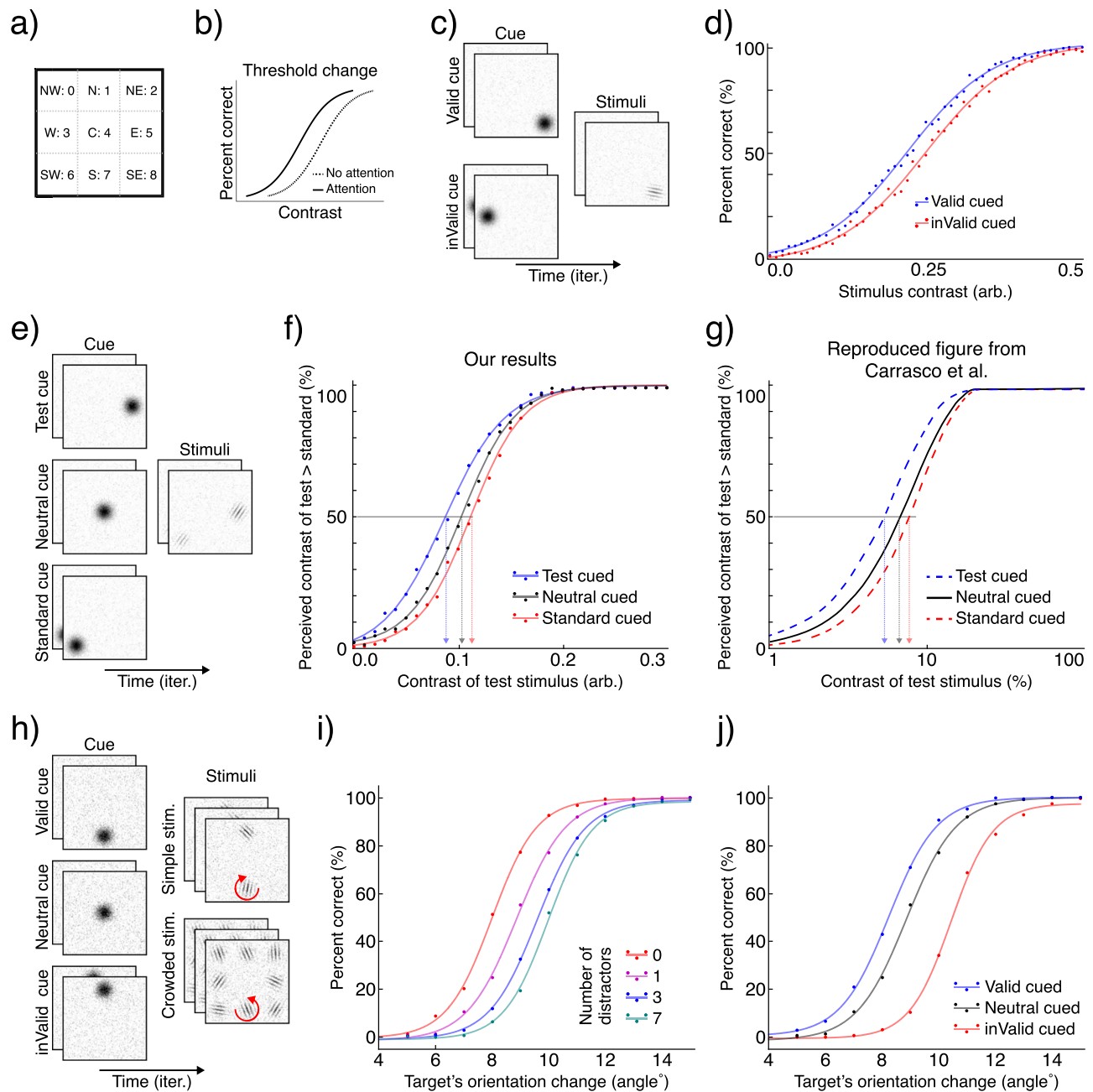

**Fig. 7 | Psychophysical results. a** Gabor patches (targets/distractors) can only appear in the eight off-center locations. For all experiments, the model is trained to return the class label of the location (i.e., 0–8). The center label (4) is reserved for null (i.e., absence of target). **b** Schematic illustration of the contrast gain mechanism, where attention shifts the psychometric function to the left, lowering the contrast threshold. **c** Sample sequence for the cued localization task. A cue precedes the stimuli, appearing at either the target's location (valid cue) or a different location (invalid cue). **d** Results showing contrast gain. The psychometric function for the attended (validly cued) condition is shifted to the left compared to the unattended (invalidly cued) condition, indicating a reduced contrast threshold for the attended location. **e** Sample sequence for the contrast discrimination task. The cue either precedes the "test" patch, the "standard" patch, or a neutral location (the center). **f** Our results qualitatively match the human data, showing a similar shift of the 50% threshold, demonstrating that attention increases the apparent contrast of the cued stimulus. **g** Original human data showing how attention alters perceived contrast. Reproduced figure from Carrasco et al.[114] for Gabor patches with 2 cpd (cycle per degree) and standard patch contrast of 6%. **h** Sample sequence for the orientation-change detection task. The number of distractor patches was varied from 0 to 7 to control the perceptual load. **i** The effect of perceptual load on performance. For neutrally cued trials, change-detection accuracy decreases as the number of distractor patches increases. **j** Inattentional blindness effect shown for a set size of two patches. Performance is highest with a valid cue and is significantly impaired when an invalid cue directs attention to a distractor, falling well below the neutral cue baseline.

blindness[116,117]. Perceptual Load Theory posits that the ability to process task-irrelevant information depends on the demands of the primary task; high load consumes attentional resources, leaving little capacity for processing the task-irrelevant stimuli[115]. A direct consequence of this is inattentional blindness: the striking failure to notice a salient, unexpected object when attention is engaged by a demanding task[118,119]. To determine if our model's behavior is consistent with these phenomena, we designed a cued orientation-change detection task. In the task, a spatial cue was followed by the presentation of one to eight Gabor patches (Fig. 7h). On 50% of trials, one

of these patches changed its orientation, and the model's objective was to detect and report the location of the changed stimulus. During training, the cue's location was correlated with the target patch's location with 50% validity, encouraging the model to use the cue. Our evaluation yielded two key results. First, we found a clear effect of perceptual load: as the number of distractor patches increased, the model's ability to detect the orientation-change systematically decreased, as illustrated by the rightward shift in the psychometric functions (Fig. 7i) (50% threshold as a function of number of distractors; 0: $th_{50\%}$ = 7. 9°, 1: $th_{50\%}$ = 8. 8°, 3: $th_{50\%}$ = 9. 5°, and 7: $th_{50\%}$ = 10. 0°). Second, to test for inattentional blindness, we analyzed performance based on cue validity. When the cue correctly located the future target patch (valid cue), performance was highest. However, when the cue directed attention to a distractor (invalid cue), performance dropped dramatically, falling well below the neutral-cued condition (Fig. 7j) (valid-cued: $th_{50\%}$ = 8. 2°, neutral-cued: $th_{50\%}$ = 8. 8°, and invalid-cued: $th_{50\%}$ = 10. 1°). This profound impairment in detecting a visible change when attention was misdirected is a clear analog of inattentional blindness.

## Neurophysiological results

**Object-based attention.** Inspired by the work of Roelfsema et al.[120], we tested our model's ability to perform object-based attention through a curve-tracing task and to examine the attention-mediated neuromodulation in the network. Here, an object is defined as a continuous and regular curve, following the Gestalt principle of continuity. Curve tracing is a task designed to investigate attentive neuromodulation while reducing the influence of pre-attentive and overt processes. In the original curve-tracing experiment, two curves were drawn that might or might not intersect. A cue was presented to determine the target curve, and the subject was trained to covertly trace the curve. In our version of the curve-tracing experiment, we use contiguity as a cue to indicate which curve the model should attend to. The task is as follows: a cue is shown to the model for two iterations, followed by three iterations of two continuous, smooth, and equally salient curves sampled from a third-order Bézier curve generator. One of the two curves is the target object, overlapping the previously shown cue, while the other curve is the distractor. Finally, the two curves disappear and two dots appear (see Methods for training details). The task is to first attend to the cue, then to the cued curve (target), and finally to the dot that overlaps with the target (Fig. 8).

An advantage of this experiment is that for any combination of two curves, we can use the initial cue to switch the target and distractor curves, giving us the control we require for the experiment. Our model not only learns to perform the task correctly (i.e., it can learn contiguity, continuity, and regularity to attend to the correct sequence of stimuli), but its neural activity is also consistent with findings from the primary visual cortex of the macaque performing a similar experiment[120] (Fig. 8). This shows that attention in our model operates similarly to biological attention, positively modulating the activity of neurons whose receptive fields fall on the target curve, compared to when the same curve is the distractor (i.e., unattended).

**Attention-invariant tuning.** Tuning curves are often used to depict how a neuron changes its activity (i.e., firing rate) in response to a change in a single attribute of its input. A common tuning curve for neurons in early visual areas, which are sensitive to stimuli similar to two-dimensional Gabor wavelets, is orientation tuning. To extract tuning curves, experiments are designed so that the neuron's activity is mostly driven by input from early visual areas, with recurrent or higher-level signals minimized. However, here we are interested in how the neuron changes its tuning in the presence of attention. Neurophysiological studies suggest that when a subject is attending to an object, neurons whose receptive fields contain the attended object show stronger activation than when the same object is unattended[121–124]. Furthermore, McAdams and Maunsell showed that this change in activation is multiplicative, meaning that other attributes such as width and preferred orientation are statistically unaffected, suggesting that neural tuning is attention-invariant[121]. Similar to biological neurons, neurons in the early layers of an artificial neural network trained on naturalistic images exhibit tuning curves that resemble Gabor wavelets. In the deeper layers, however, the learned filters become increasingly complex, resembling the tuning curves of neurons in the ventral visual pathway (Fig. 9a, b). Here, we aim to see whether the neurons in our model also exhibit attention-invariant tuning.

We start with the model trained on the curve-tracing task and follow the method described in ref. 121. As input, we use a composition of two bars (i.e., short straight lines). We use visual cues to switch attention between the two bars, creating two states, attended and unattended, for the target bar. We then extract neurons from the penultimate feature layer that have the target bar in their receptive field. Furthermore, we only consider those neurons that exhibit a bell-curve (i.e., Gaussian-like) orientation tuning (see Methods) (Fig. 9a). Similar to neurons in the visual cortex, neurons in our network have progressively larger and more complex receptive fields (less Gaussian-like) in deeper layers[125,126] (Fig. 9b). Finally, we show that attention affects the neural activation in later layers of our network through multiplicative scaling, without changing the overall shape of their tuning curves, similar to experimental findings in V4 cortical neurons[121] (Fig. 9c).

**Figure-ground separation.** Building upon recent studies on border-ownership and figure-ground perception in primate vision[127,128], we designed an experiment to investigate how our network performs figure-ground separation. Specifically, we aimed to test whether our model implements figure-ground separation via a similar feedback mechanism to that recently hypothesized by Jeurissen and colleagues[128] (i.e., positive feedback aligned with preferred figure location and negative feedback for background).

To that end, we trained a neural network, incorporating our bidirectional gating mechanism, on a multi-attribute classification task involving shape, color, and texture, without explicit attention supervision (see "Methods" and Supplementary Materials for details). From this trained network, we selected two distinct groups of neurons, each consisting of 128 units from neighboring bidirectional recurrent gating block: one group from layer 5 of the feature pathway (F-neurons) and another from layer 4 of the attention pathway (A-neurons) (Fig. 10a). We then determined the receptive fields of the F-neurons within the input space using methods described in ref. 129, as illustrated by the dotted red box in Fig. 10b.

We then created two sets of stimuli: stimulus (i), a colored, untextured novel object on a blank background (Fig. 10b, stimulus i); and stimulus (ii), an uncolored object (i.e., blank shape) on a background matching the object color from stimulus (i) (Fig. 10b, stimulus ii). We chose the object's shape carefully so that, regardless of rotation, it always occupied exactly half of the receptive field and ensured that, from the perspective of the F-neurons, stimulus (i) rotated by $\theta°$ appeared identical to stimulus (ii) rotated by $\theta + 180°$ (Fig. 10c, top row).

Finally, we analyzed the tuning curves and neural activities of the selected F- and A-neurons. We hypothesize that the pre-attentive activity of F-neurons would be object-agnostic, given that their receptive fields do not cover the entire visual scene. Consequently, the tuning curves of F-neurons responding to stimulus (i) should closely match those for stimulus (ii) rotated by 180° (Fig. 10d, e). In contrast to the F-neurons, the A-neurons exhibited markedly different behavior: their activities for both stimuli were highly similar and aligned in phase with their preferred orientations (Fig. 10f, g). Additionally, neurons in the attention pathway consistently showed strong positive modulation

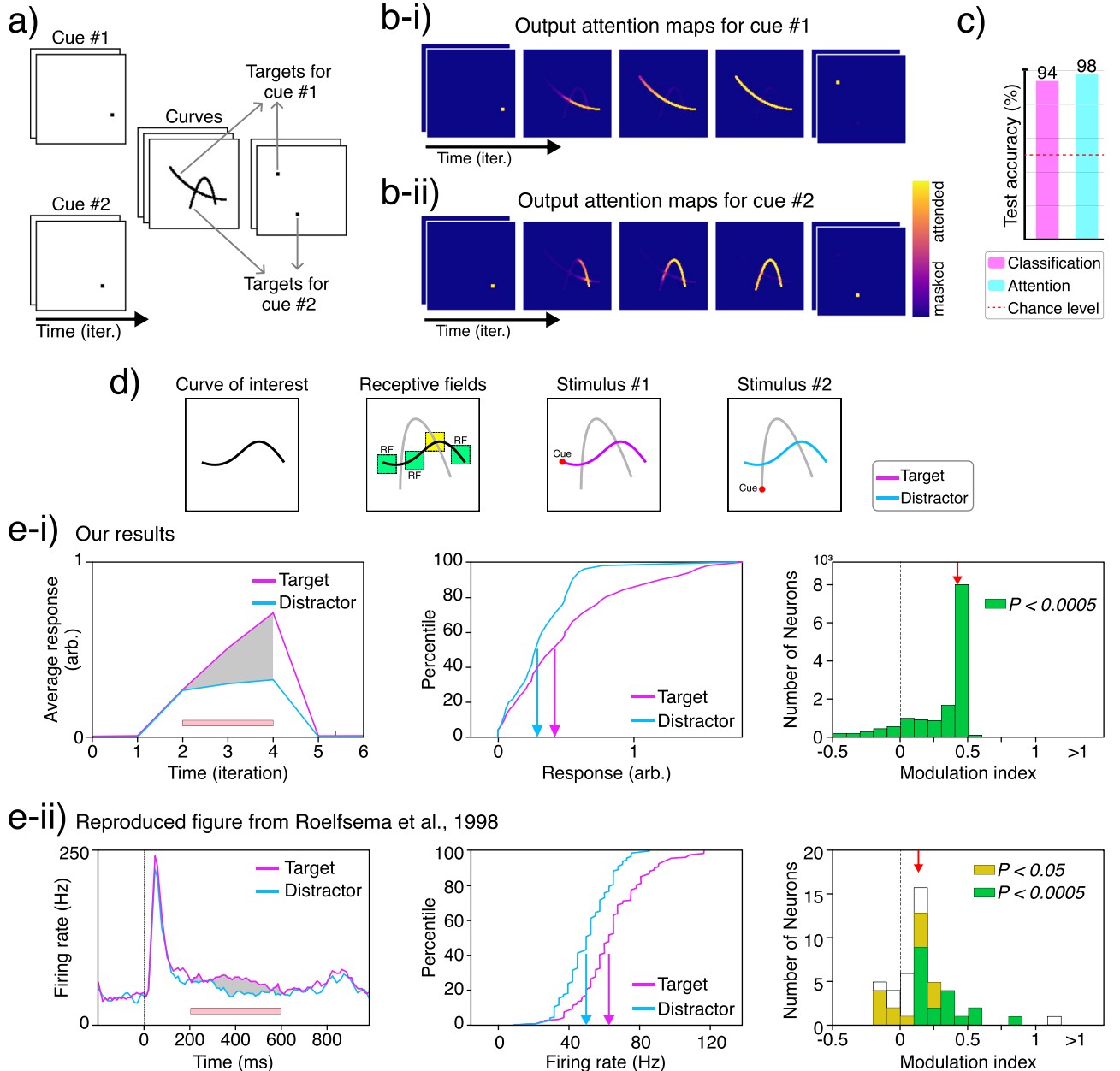

**Fig. 8 | Object-based attention in curve-tracing task. a** The input sequence for the curve-tracing experiment consists of three phases from left to right: (i) fixation (i.e., visual cueing) for two iterations; (ii) stimulus (i.e., curve tracing) for three iterations; (iii) saccade (i.e., attending to the target decision point) for two iterations. Note that for any two curves, we can use the cue to specify which curve is the target. **b-i** and **b-ii** Estimated attention maps for two sequences that share the same curves and decision points but differ in the cues. **c** Classification accuracy for the correct decision point and attention map accuracy for the target curve (number of test samples, $n = 1024$). **d** For any two curves, we identify all neurons whose receptive fields include one curve but not the other (here, green squares are considered acceptable receptive fields, while the yellow one is an example of a rejected receptive field). We can then use the visual cue to mark either curve as the target object (stimulus #1) or as the distractor object (stimulus #2). **e** Neural activity is

enhanced through attention. Here, we compare results from (**e-i**) neurons in the first layer of our model during the curve-tracing experiment with (**e-ii**) results from V1 neurons in macaques performing a similar task[120]. The left panels show neural responses over time for the target (i.e., attended) versus the distractor (i.e., unattended) curves. Our plot shows the average response across all selected neurons in the first layer. The pink bar marks the curve-tracing window used for statistical analysis. The middle panels show the response distributions for the target and distractor scenarios. The right panels depict histograms of the response modulation index, showing significant positive modulation of neuronal activity ($P < 0.0005$, sign-test; $n = 13,842$). The red arrows indicate the median modulation index (ours: 0.43; Roelfsema et al. 1998: 0.27). The plots are styled similar to those in Roelfsema et al. (1998)[120].

at their preferred compared to non-preferred orientations, irrespective of stimulus type (Fig. 10h).

Our primary analyses, as illustrated in Fig. 10, demonstrate that our computational model exhibits neuronal behaviors closely aligned with the recent biological findings by Jeurissen et al. (2024). Specifically, neurons within our model similarly separate into two major

groups: (1) feature-pathway neurons (F-neurons), displaying orientation-tuned responses analogous to biological orientation-tuned neurons (Fig. 10d, e); and (2) attention-pathway neurons (A-neurons), which are selectively tuned to object directions, consistent with the responses of border-ownership-tuned cortical neurons (Fig. 10f, g, h). Together, these findings suggest that our architecture successfully

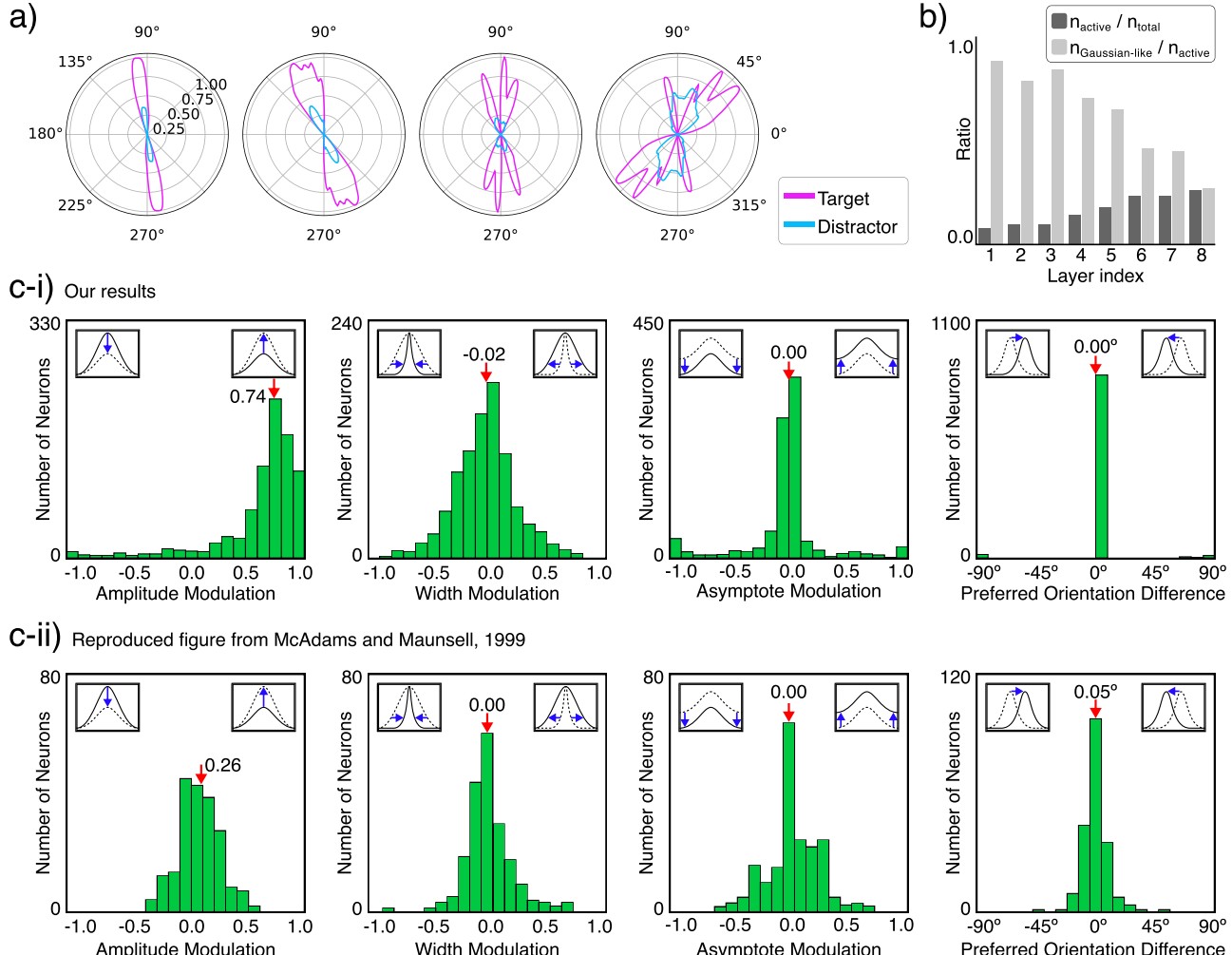

**Fig. 9 | Attention-invariant tuning. a** Orientation tuning curves of four neurons from layers 1, 3, 5, and 7 (from left to right) of our 8-layer model trained on the curve-tracing task. For the analysis of the attention-invariant tuning, we only consider neurons with Gaussian-like tuning curves (e.g., the two plots on the left) and reject the rest. **b** The ratio of neurons that respond to the target bar (i.e., the target bar is in their receptive field) increases in deeper layers, while their orientation tuning curves become more complex (i.e., less Gaussian-like). **c** Attention increases the response amplitude but has no significant effect on the width, asymptote, and preferred orientation of the tuning curves. **c-i** Our results from the penultimate feature layer of our model are compared to (**c-ii**) findings from V4 cortical neurons in macaques[121]. The red arrows and values indicate the median in each graph. Number of neurons used for the analysis, *n* = 883. The plots are styled similar to those in McAdams & Maunsell (1999)[121].

learns and implements figure-ground separation mechanisms comparable to those described in primate visual cortex.

## Discussion

In this study, we have presented an integrated computational model of visual attention and binding. Our main contributions can be understood along four primary axes: First, we address the long-standing challenge of unification[130–132], demonstrating that a single mechanism (i.e., bidirectional recurrent gating) can account for the diverse manifestations of spatial, feature-, and object-based attention. The canonical nature of cortical microcircuits[133] and the columnar circuitry of our mechanism lead to the intriguing hypothesis that while feature extraction is modality-specific, the core mechanism of recurrent gating for attentional selection may be a shared computational strategy in other sensory systems, such as auditory and somatosensory cortices. Second, we provide a mechanistic implementation of the emergent-attention hypothesis[51,68], showing how complex phenomena such as attention and binding arise intrinsically from the model's architecture and learning paradigm. Specifically, our results are consistent with the "binding by firing rate enhancement" theory[31], which proposes that features are integrated into coherent object representations through

the selective amplification of their corresponding neural activity. Third, we establish the model's cognitive plausibility by showing that it not only performs canonical attention tasks[77], but also reproduces characteristic limitations of human perception, such as perceptual load and inattentional blindness. Finally, we connect the model to neurophysiology at the circuit level, offering testable hypotheses about underlying neural computations and showing that its internal units develop representations analogous to those observed in the primate visual cortex.

Beyond replicating established phenomena, our model offers a set of specific, testable hypotheses for the underlying computations. We propose a mechanism in which the feedforward feature pathway performs feature extraction of the input stimuli, and the top-down attention pathway integrates global scene information with context and task objectives to determine which features are currently relevant[127,128]. The critical interaction occurs when the attention pathway broadcasts this relevance information back via multiplicative, top-down modulatory signals. As revealed in our neurophysiological analyses, this modulation manifests as a response gain that selectively amplifies the activity of feature-pathway neurons corresponding to the attended stimulus, without altering their fundamental tuning

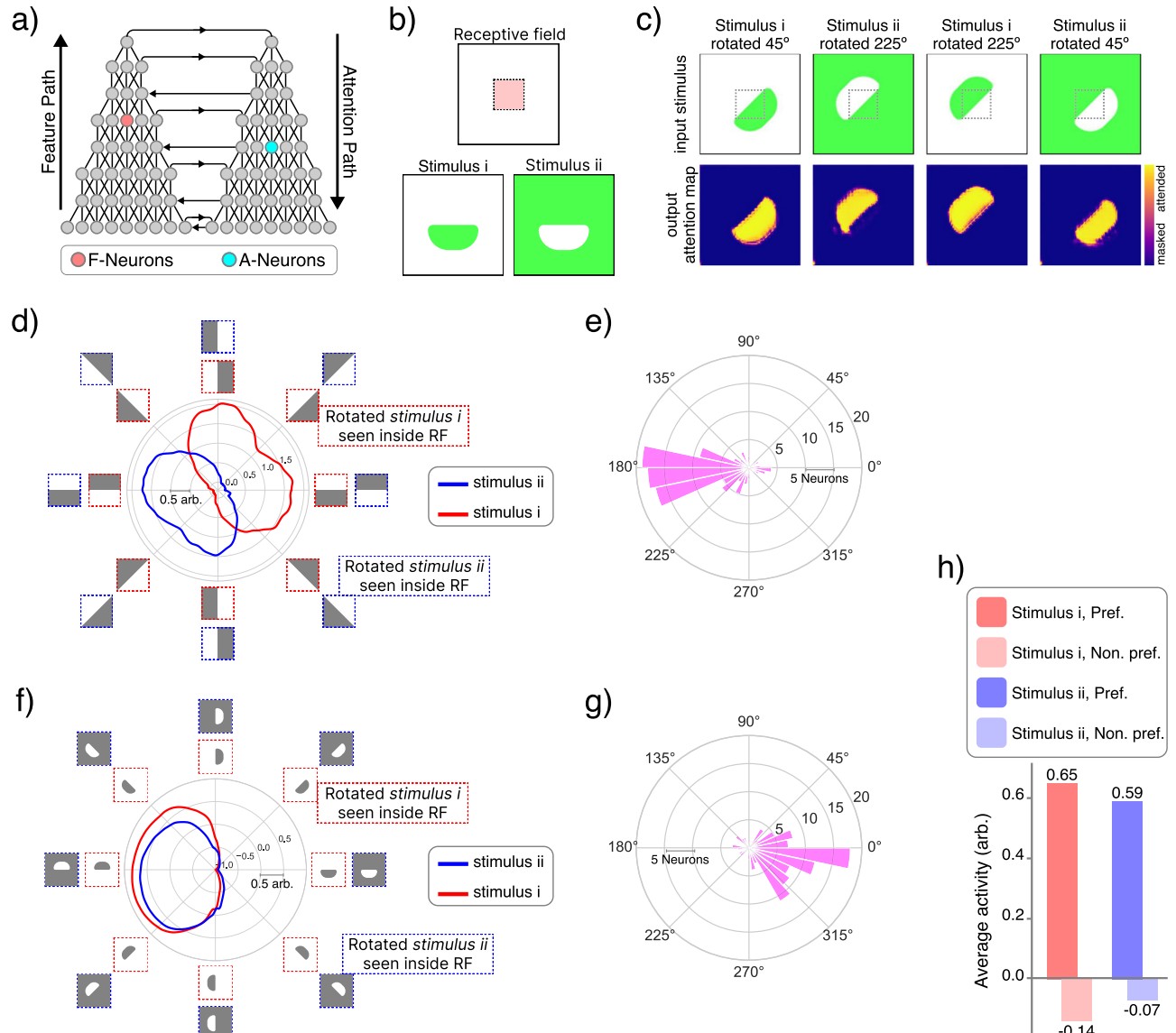

**Fig. 10 | Figure-ground separation. a** Schematic of the network architecture used for the multi-attribute classification task, highlighting both the feature and attention pathways. Colored circles indicate the locations of neurons analyzed in this study. **b** Top: Receptive field of neurons from the feature pathway (F-neurons), indicated by the dashed red box. Bottom: Two stimuli used for analysis; stimulus (i), a colored object over a blank background; and stimulus (ii), a colored background, matching the object color in stimulus (i), containing an uncolored stencil object. **c** From the receptive field's viewpoint, stimulus (i) rotated by $\theta°$ makes it identical to stimulus (ii) rotated by $\theta + 180°$. The network successfully generates accurate attention maps across all stimuli variations, achieving 91% attention accuracy. **d** Polar plot showing the activity of an example F-neuron as a function of stimulus rotation. Dashed boxes outside the plot illustrate receptive field content for stimulus (i) (red) and stimulus (ii) (blue). **e** Histogram illustrating the distribution of phase differences of preferred orientation for F-neurons between the two stimuli ($n = 119$ neurons). **f** Polar plot showing the activity of an example A-neuron as a function of stimulus rotation, with receptive field content indicated in the same way. Because the attention pathway receives input from the bottleneck, A-neurons have receptive fields spanning the entire input scene. **g** Histogram showing preferred orientation phase differences of A-neurons between the two stimuli ($n = 112$ neurons). **h** Average activity of A-neurons at their preferred versus non-preferred orientations, shown for both stimulus types.

properties[121,122]. Here, this response-modulated signal interacts with local divisive normalization and nonlinear circuits, biasing the competition towards the information relevant to current behavior[62]. This targeted amplification and normalization effectively increase the "contrast" of task-relevant features and reshapes subsequent processing. This process provides a mechanistic account for how top-down signals implement task- and context-relevant feature selection, and is further supported by the emergent specialization of neurons in our model, which mirrors the distinct roles of feature-tuned and context-aware border-ownership neurons found in the visual cortex[128].

Naturally, our model has limitations in its ability to fully capture the breadth of neural and behavioral phenomena. For instance, it does not incorporate overt attention mechanisms such as saccades and foveation, which are essential components of human visual attention[2]. Fortunately, effective models of these mechanisms exist and could, in principle, be integrated into our framework[134]. Additionally, our current implementation of inhibition of return (IOR) relies on an explicit training objective rather than biologically grounded mechanisms such as peripheral stimulation and oculomotor activation[96]. Nonetheless, we believe that the recurrent and modulatory nature of our architecture provides a strong foundation for incorporating these biologically motivated dynamics in future extensions.

A second limitation lies in the learning algorithm. Our model uses Backpropagation Through Time (BPTT)[135] to update its weights. While

BPTT is a powerful and widely used optimization method in machine learning, there is no direct evidence that the brain employs back-propagation, particularly not in a BPTT-like setting where information would need to propagate backward through time[136]. Nevertheless, we argue that our mechanism provides a strong inductive bias that enables attentive behavior to emerge and allows our network to perform well on multiple attention tasks, even if the underlying learning rule is not biologically grounded. However, biologically plausible alternatives to gradient descent have been extensively studied[137,138], and backpropagation itself has been shown to produce brain-like representations in visual tasks[139-143]. Similarly, weight sharing in convolutional layers lacks direct biological realism, even though it was originally inspired by the organization of simple and complex cells in the visual cortex[144]. Recent work, however, has proposed augmentations or wake-sleep cycles to train networks that exhibit emergent weight sharing[145]. In other words, we view backpropagation not as a biological mechanism per se, but as an effective computational tool for training our biologically inspired architecture.

Even though the proposed multitask learning paradigm can reduce reliance on full supervision for certain tasks, our model still requires supervised signals, including segmentation maps that lack a direct biological analog. We acknowledge that the use of abstract, binary object segmentation masks in some tasks is a departure from biologically plausible learning, and their inclusion reflects practical constraints, specifically the absence of richer sensory input and suitable datasets that could otherwise guide learning in a more naturalistic way. Indeed, humans leverage a variety of other information and mechanisms to infer objectness[146] and reasonable attention maps. For example, object motion provides a powerful auxiliary signal that when combined with the object persistence principle (e.g., through slow-feature analysis[147]), it can help separate objects from the background. In parallel, binocular vision provides depth information about the scene and, therefore, a useful signal for figure-ground segmentation. Similarly, additional strong cues exist for other aspects we use in supervised learning. For instance, abstract knowledge about object identity is often encoded across different modalities; for example, the presence of an object (e.g., a horse) is often associated with distinct multi-modal cues, such as characteristic sounds or smells.

A key strength of this framework is its ability to generate specific and falsifiable predictions to guide future empirical research. For instance, our model posits that a shared neural hierarchy is dynamically modulated by task demands. This leads to a clear prediction: while high-level control areas should exhibit strong task-specificity, neurons in early to mid-level visual pathways should maintain relatively stable tuning properties as an animal switches between different attentional modes, such as spatial cueing versus feature-based search. Furthermore, the model predicts that attentional processing is critically shaped by experience. We hypothesize that animals raised on a "diet" of tasks solvable without object integration would develop impoverished neural correlates for binding and segmentation compared to those with richer, more complex training. Finally, our model can be used in silico to design targeted experiments: specific patterns of neural activity during perceptual illusions or under varying levels of perceptual load can be simulated and then directly compared against neural recordings from subjects performing the same tasks. Any discrepancies would provide a systematic path for refining the model's underlying mechanisms.

Our finding that tasks can be either aligned (mutually supportive) or orthogonal (mutually interfering) invites speculation on the underlying principles governing their interaction (see Supplementary Materials). We hypothesize that task interference may arise from at least two sources. First, interference may stem from competition for finite computational resources within our architecture (a limitation related perhaps to the absence of distributed working memory or dorsal stream processing). For example, tasks requiring dynamic state processing, such as object tracking and inhibition of return, are likely to compete for the limited resources of the model's recurrent circuits[148]. Second, interference could be caused by conflict between bottom-up, stimulus-driven signals and top-down, goal-directed commands. For example, a task directing attention to a salient pop-out stimulus would align these two streams, whereas searching for a non-salient object would create competition between the exogenous pull of a salient distractor and the endogenous search goal[148]. In principle, it may be possible to predict the relationship between any two tasks by characterizing their demands along these axes of resource usage and signal congruence, offering a clear direction for future investigations into multitask learning.

Taken together, we propose our architecture as a pragmatic and biologically grounded alternative to the feedforward models commonly used to study visual cognition[149-151]. By embracing a multitask learning framework, the model not only learns efficiently but also develops more robust and generalizable representations, creating opportunities to probe emergent behaviors in settings that better approximate the complexity of natural vision[146,152-154]. We see this work as a flexible foundation for developing more comprehensive cognitive models, offering a platform to investigate how attention-transformed representations influence downstream processes like long-term memory and credit assignment. Crucially, incorporating neurally-grounded attention mechanisms is not just a matter of biological realism; it is essential for bridging computational neuroscience and clinical applications. A promising future direction is to use this framework to model visual impairments, providing a testbed for understanding deficits rooted in covert attention, feature integration, and binding processes[155].

Finally, this work contributes to a growing dialog between neuroscience and machine learning on the importance of object-centric processing[156,157]. By engineering attention not as a post-hoc module but as a fundamental computational primitive, we provide a blueprint for systems that can dynamically determine relevance based on the joint interaction of stimuli, tasks, and internal states. This approach directly embraces the compositional nature of the real world—a critical step beyond static pattern recognition toward more flexible and robust machine intelligence[158]. Looking forward, the central question is not merely whether human-like attention can enhance network performance, but how the integration of such cognitive mechanisms could lead to the next generation of AI systems capable of more general-purpose reasoning, adaptation, and a deeper understanding of the world they perceive.

## Methods

We follow the DOME recommendations and framework presented by ref. [159] with respect to assessment and reproducibility.

### Data

The data samples used in our experiments are task-specific and generated by augmenting and composing images from publicly available datasets: MNIST[79], COCO[80], CelebA[108], CIFAR-100[84], and STL-10[160]. We apply various image augmentations (e.g., random flipping, color jitter, and blurring) to improve generalization. Regarding train-test splits, the CelebA dataset provides predefined training, validation, and test sets, which we use as provided. However, MNIST, COCO, and CIFAR-100 do not include a publicly available labeled test set. For these, we split the official training set into training and validation sets and perform cross-validation. We then use the official validation set for testing. For STL-10, we split the official validation set into two non-random subsets for validation and testing, ensuring that the test subset is held out entirely during model validation. Data samples for the curve-tracing experiment were generated synthetically using randomly drawn Bézier curves. Data samples for the shape recognition task are provided in the GitHub repository. In all experiments, validation sets are used only

**Table 1 | Training hyper-parameters**

|  | n-Epochs | Batch-size | Learning rate $\eta$ | $\eta$ Scheduler milestones | scheduler $\gamma$ | L2-rate $\lambda$ |
|---|---|---|---|---|---|---|
| MNIST | 96 | 128 | $5 \times 10^{-4}$ | [32, 64] | 0.2 | $1 \times 10^{-6}$ |
| COCO | 48 | 128 | $5 \times 10^{-4}$ | 0.25 | OneCycleLR | $1 \times 10^{-5}$ |
| CelebA | 32 | 128 | $2 \times 10^{-4}$ | [16, ] | 0.1 | $5 \times 10^{-4}$ |
| Contrast Detect. | 32 | 64 | $1 \times 10^{-4}$ | 0.25 | OneCycleLR | $5 \times 10^{-4}$ |
| Contrast Discrim. | 32 | 64 | $1 \times 10^{-4}$ | 0.25 | OneCycleLR | $5 \times 10^{-4}$ |
| Ori. Change Detect. | 64 | 64 | $5 \times 10^{-4}$ | 0.25 | OneCycleLR | $1 \times 10^{-4}$ |
| Fig-Grnd-Sep | 64 | 64 | $2 \times 10^{-4}$ | 0.125 | OneCycleLR | $1 \times 10^{-4}$ |
| Curve Tracing | 64 | 128 | $5 \times 10^{-4}$ | – | – | $1 \times 10^{-6}$ |
| CIFAR-100 | 64 | 64 | $5 \times 10^{-4}$ | 0.125 | OneCycleLR | $1 \times 10^{-4}$ |
| Multi-Modal Search | 64 | 64 | $1 \times 10^{-4}$ | – | – | $5 \times 10^{-5}$ |

Contrast Detect.: Contrast detection. Contrast Discrim.: Contrast discrimination. Ori. Change-Detect.: Orientation-change detection.

during inference to select the best-trained model, while test sets are reserved for final reporting and visualizations.

## Optimization

All models were developed and trained in PyTorch[161] using a single NVIDIA A100 GPU. We used the Adam optimizer with weight decay for regularization. A grid search was performed to identify learning rates and regularization factors that yielded better validation performance. We found that very small learning rates often produced better results, specifically, more stable training and more interpretable attention maps. Learning rate warm-up also proved extremely helpful, particularly due to the use of layer normalization. For this, we used Cosine OneCycleLR scheduling (Table 1). Our loss function is defined as a weighted composition of three objectives:

$$L = \sum_{k=1}^{m} \sum_{i=1}^{n} \left[ \alpha_i^k \; \text{MSE}(\widehat{\mathbf{M}}_i^k, \mathbf{M}_i^k) + \beta_i^k \; \text{CrossEntropy}(\widehat{y}_i^k, y_i^k) \right] + \lambda \parallel \mathbf{W} \parallel^2 \tag{1}$$

where subscript $i$ denotes the iteration, and superscript $k$ indicates the task index. The first term inside the sum is the Mean Squared Error (MSE) loss between the estimated attention map $\widehat{\mathbf{M}}_i^k$ and the ground-truth attention map $\mathbf{M}_i^k$, if available. The second term is the cross-entropy loss used for classification of the output labels. For each experiment, the presence and relative importance of these loss terms at iteration $i$ are controlled using hyperparameters $\alpha_i^k$ and $\beta_i^k$. The final term is the $L2$ regularization applied to the learnable weights $\mathbf{W}$. In the COCO experiments, we additionally used class weights to improve performance on imbalanced categories.

Since different tasks involve varying numbers of iterations and task-specific hyperparameters, we trained the model using mini-batches that contained stimuli from a single task at a time. During training, batches were interleaved across tasks to ensure that no two consecutive batches came from the same task. This approach simplifies optimization but imposes a constraint on batching. In principle, this constraint could be lifted if all tasks shared the same number of iterations and the IOR task, due to its intermediate prediction steps, were excluded. Alternatively, recent developments in deep learning frameworks, such as Ragged or Jagged tensor operators in PyTorch and TensorFlow, now allow for efficient batching of variable-length sequences. These tools could enable more flexible multitask batching in future implementations.

Table 2 lists the number of iterations (recurrent loops) and loss hyperparameters (i.e., $\alpha_i^k$ and $\beta_i^k$) for each task and iteration. These values were selected primarily for simplicity, without extensive tuning or grid search. An exception is the COCO dataset, where we did perform tuning to ensure balanced performance across tasks. In earlier

experiments, the network quickly learned to perform object recognition and perceptual grouping, but struggled with top-down search. Therefore, we adjusted the $\alpha_i$ and $\beta_i$ values for these tasks to enhance the effectiveness of the top-down search error gradients.

**IOR experiment.** Training the Inhibition of Return (IOR) task requires a more nuanced approach compared to other tasks, where each input sequence typically has a single target label and attention map, allowing for straightforward optimization via cross-entropy (CE) and mean squared error (MSE) losses. In the IOR task, however, each input contains multiple target objects, and we explicitly aim to avoid imposing any fixed order of attention. This necessitates a training strategy that supports flexible attention sequencing while still enforcing accurate classification and attention for each object.

We describe the training procedure for a 3-object composite input, where the network is given two iterations per object, resulting in six iterations total. The input sequence, comprising repeated presentations of the same 3-object image, is fed to the network, which produces a sequence of predicted class labels and attention maps, one per iteration. Up to this point, the process mirrors that of other tasks. The key difference lies in how the loss is computed.

Specifically, we extract the outputs from the 2nd, 4th, and 6th iterations. At the 2nd iteration, we compare the predicted attention map to all three target attention maps using a sum-of-differences metric and identify the object with the closest match (i.e., the most attended object by the network). This object is designated as the first target, and its corresponding label and attention map are used to compute the CE and MSE losses for that iteration. At the 4th iteration, we exclude the first selected object and compare the predicted attention map to the remaining two targets. The closest match defines the second attended object, which is used for computing losses. The process is repeated at the 6th iteration, where only one object remains. This dynamic matching approach enables the network to learn a valid attention sequence without enforcing a fixed order, while implicitly guiding it to avoid re-attending to previously selected objects, thereby capturing the core behavior of inhibition of return.

**Psychophysical experiments.** For the psychophysical experiments, we used a 2D Gaussian function with amplitude = 1.0, and width-scale = 8.0 as the cue. For stimuli, we used a Gabor wavelet with $\sigma = 0.75$ and $k = 12$ with random phase (i.e., $\theta$) and amplitude. A key ingredient in the input images was the amount of background noise. Background noise was sampled from a unit uniform distribution with a multiplicative scaling factor (Table 3). For contrast detection task, the noise scaling factor was drawn from a uniform distribution with width [0.0, 0.25] during training and fixed to 0.25 for reporting. For contrast

**Table 2 | Task hyper-parameters**

|  |  | n-iterations | $\alpha_i$ | $\beta_i$ |
|---|---|---|---|---|
| MNIST | Object recognition | 3 | 0.0 | [0.0, 0.5, 0.5, 2.0] |
|  | Spatial Cue | 2 + 3 | [0.0, $\frac{1}{4}$, $\frac{1}{4}$, $\frac{1}{4}$, $\frac{1}{4}$] | [0.0, 0.0, 0.0, 0.0, 1.0, 1.0] |
|  | Symbolic orienting | 3 | 0.0 | [0.0, 0.0, 0.0, 1.0] |
|  | Pop-out | 2 | 0.0 | [0.0, 0.0, 1.0] |
|  | Top-down search | 2 | [0.0, 1.0] | 0.0 |
|  | Object tracking | 2 + 5 | [0.0, $\frac{1}{6}$, $\frac{1}{6}$, $\frac{1}{6}$, $\frac{1}{6}$, $\frac{1}{6}$] | [0.0, $\frac{1}{6}$, $\frac{1}{6}$, $\frac{1}{6}$, $\frac{1}{6}$, $\frac{1}{6}$, 1.0] |
|  | Inhibition of return | 2 × 3 | [0.0, 1.0, 0.0, 1.0, 0.0, 1.0] | [0.0, 1.0, 0.0, 1.0, 0.0, 1.0] |
| COCO | Object recognition | 3 | 0.0 | [0.0, 0.125, 0.125, 0.5] |
|  | Perceptual grouping | 2 + 2 | [$\frac{10}{3}$, 0.0, $\frac{10}{3}$, $\frac{10}{3}$] | [0.0, 0.0, 0.0, 0.0, 0.25] |
|  | Top-down search | 3 | [0.0, 5.0, 5.0] | 0.0 |
| PsycPhys | Contrast detection | 2 + 2 | [1.0, 1.0, 0.0, 0.0] | [0.0, 0.0, 1.0, 1.0] |
|  | Contrast discrimination | 2 + | [1.0, 1.0, 0.0, 0.0] | [0.0, 0.0, 1.0, 1.0] |
|  | Change detection | 2 + 3 | [1.0, 1.0, 0.0, 0.0, 0.0] | [0.0, 0.0, 0.0, 1.0, 1.0] |
| CIFAR | Object recognition | 1 | 0.0 | [0.0, 1.0] |
|  | Spatial Cue | 3 | 0.0 | [0.0, 0.0, 0.0, 1.0] |
| MMS | Object recognition | 3 | 0.0 | [0.0, 0.05, 0.05, 0.2] |
|  | Top-down search | 3 | [0.0, 0.5, 0.5] | 0.0 |
|  | Figure-ground separation | 1 | 0.0 | [0.0, 1.0] |
|  | CelebA sex classification | 2 | 0.0 | [0.0, 1.0] |
|  | Curve tracing | 2 + 3 + 2 | [0.0, $\frac{1}{3}$, 0.0, 0.0, $\frac{1}{3}$, 0.0, $\frac{1}{3}$] | 0.0 |

Task specific parameters used in the experiments. *n-iterations* refers to the number of recurrent loops per stimulus in the task. We use + and × to indicate the different phases of the task, for example in object tracking we have two iterations of fixation followed by five iterations of tracking. For $\alpha_i$ and $\beta_i$ we use a list to describe the values for each iteration. If the value used for $\alpha_i$ or $\beta_i$ is a scalar 0.0, it means this signal was not provided during training. Furthermore, for some tasks the length of lists for $\beta_i$ is one more than the number of iterations, if we perform a final feature-pass using the last attention map.

**Table 3 | Background noise factor in psychophysical tasks**

|  | Contrast detection | Contrast discrimination | Orientation-change-detection |
|---|---|---|---|
| Training | [0.0, 0.25] | [0.0, 0.2] | [0.0, 0.25] |
| Evaluation | 0.25 | 0.0625 | 0.125 |

During training, the noise scaling factor was drawn from a uniform distribution with the given width.

discrimination task, the noise scaling factor was drawn from a uniform distribution with width [0.0, 0.2] during training and fixed to 0.0625 for reporting. For orientation-change detection task, the noise scaling factor was drawn from a uniform distribution with width [0.0, 0.25] during training and fixed to 0.125 for reporting. The orientation-change detection task also included "rotation-noise" during training which was in the range of [−5, 5]°.

The model architecture for the experiments was chosen to be minimal (i.e., small number of channels per block). Although the architecture was mostly consistent between the three experiments, we used a single dense RNN layer as working memory for the orientation-change detection experiment. We used SciPy's least-squares curve fitting tool[162] and Sigmoid function as our psychometric function to fit the data:

$$\text{Sigmoid}(x) = d + \frac{c}{1 + \exp(-a(x - b))} \qquad (2)$$

where $a$, $b$, $c$, and $d$ are the parameters to be fitted.

## Model
Although all of our models share a similar core architecture (Fig. 11a), each network is scaled and configured differently based on task requirements and stimuli complexity (Fig. 11b−d). For instance, in the MNIST model, we incorporated RNN layers to provide working

memory, a requirement for tasks like inhibition of return (IOR) (Fig. 11e). Additionally, we used the GELU (Gaussian Error Linear Unit) activation function in the COCO model, as it has been shown to improve training stability and performance[163].

Here, we describe how to build a model based on our proposed Bidirectional Recurrent Gating (BRG) mechanism. We begin with the modular backbone architecture (Fig. 11a), where a key design decision is the network's depth (i.e., the number of stacked BRG blocks). In our experiments, we found no clear benefit to using very deep networks. Instead, we selected depth based primarily on two factors: the spatial dimensions of the input images (height $h$ and width $w$) and the visual complexity of the objects (e.g., handwritten digits versus natural images from COCO) (see Table 4). The only exception was the curve-tracing task, which required a deeper network to successfully learn the task under the given hyperparameter settings.

In our experiments, we implemented three variations of the BRG block, illustrated in Fig. 11b-d. For downsampling within BRG blocks, we tested both max pooling and strided convolution. We found that longer strides in the convolutional layers consistently led to better performance.

Both ReLU and GELU were effective as nonlinear activation functions; however, GELU offered a slight performance advantage in our trials. It is important to note that the final activation function in the attention pathway of all BRG blocks is always a hyperbolic tangent, ensuring that the pre-scaled modulation values remain in the range (−1, 1). For all experiments, we apply affine scaling to the attention maps using the function Scale($a$) = 1.0 + 0.5 · $a$ where $a$ is the output attention map from the BRG block (Fig. 11b−d).

While layer normalization generally improves training stability, we recommend omitting layer normalization at the lateral connections in the first BRG block (as shown in Fig. 11b). We found that the BRG variant shown in Fig. 11c is better suited for simpler stimuli (e.g., handwritten digits), whereas the deeper variant in Fig. 11d was used for more complex tasks like COCO.

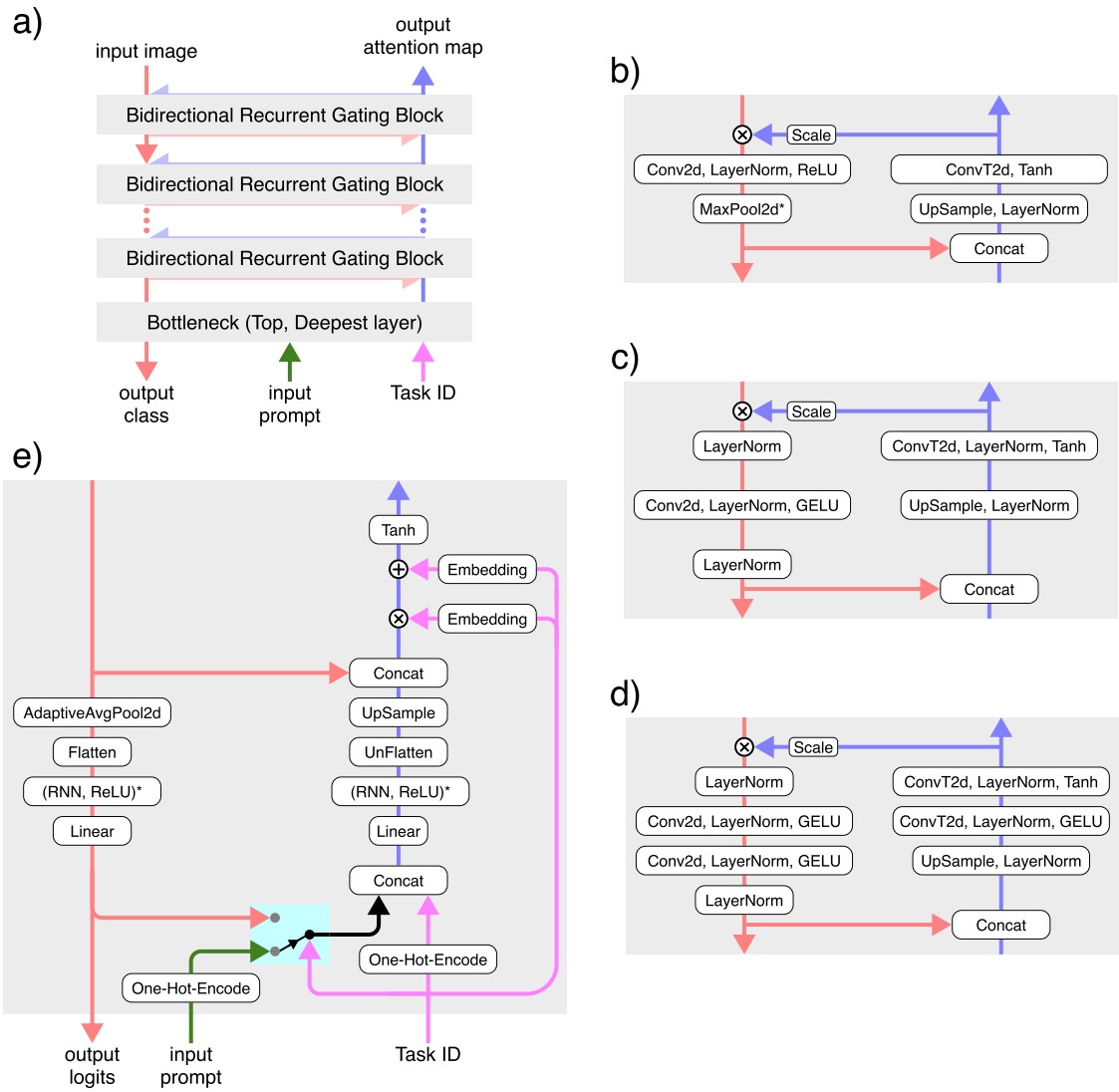

**Fig. 11 | Architecture backbone, building blocks, and elements.** The terminology used here follows PyTorch layer conventions. Sequential operations are denoted in order, for example: [Conv2d, LayerNorm, GELU] represents the operation **Y** = GELU(LayerNorm(Conv2d(**X**))). Asterisks (*) indicate optional layers. **Layer definitions:** *Conv2d*: 2D convolution; *ConvT2d*: Transposed 2D convolution (equivalent to ConvTranspose2d in PyTorch); *LayerNorm*: Layer normalization; *GELU*: Gaussian Error Linear Unit; *ReLU*: Rectified Linear Unit; *Tanh*: Hyperbolic tangent; *MaxPool2d*: 2D max pooling; *AdaptiveAvgPool2d*: Adaptive average pooling; *Concat*: Channel-wise tensor concatenation; *UpSample*: 2D upsampling; *Flatten* / *Unflatten*: Reshape between 4D (batch, channel, height, width) and 2D tensor

shapes (batch, channel × height × width); *(RNN, GELU)*: Vanilla recurrent layer with GELU activation; *Linear*: Fully connected layer; and *Embedding*: Embedding layer; **a** The modular backbone of our architecture. The number of BRG blocks is chosen based on input size and task complexity. **b** A BRG block variant with an optional MaxPool2d layer, suited for first BRG block in the network. **c** A lightweight BRG block using strided convolutions, preferred for simpler stimuli. **d** A deeper BRG block also using strided convolutions, used for more complex stimuli like natural images. **e** The bottleneck module (optionally with an RNN), where all signals converge and interact. A logic-switch denotes conditional feedback from output logits to the attention pathway when no external prompt is provided.

**Table 4 | Number of BRG blocks for each network**

| Model | MNIST | COCO | CelebA | Shapes | CIFAR-100 | Curve-tracing | PsycPhys |
|---|---|---|---|---|---|---|---|
| Image dimensions (h, w) | (96, 96) | (256, 256) | (128, 128) | (128, 128) | (96, 96) | (128, 128) | (96, 96) |
| Number of BRG Blocks | 5 | 6 | 7 | 5 | 5 | 8 | 5 |

Finally, the bottleneck, where all signals converge, is shown in Fig. 11e. We use a logic-switch symbol to indicate conditional feedback: unless the task requires an external prompt, the predicted output logits are routed into the attention pathway. For the MNIST model, we also incorporated RNN layers in both the feature and attention pathways at the bottleneck to support tasks requiring working memory.

In Fig. 12 we provide the detailed model architectures used for the MNIST and COCO experiments. The main differences are the presence

of a dense RNN layer in the MNIST model (Fig. 12a), and the double stacked BRG layers in the COCO model (Fig. 12b).

For instance, in the MNIST model, we incorporated RNN layers to provide working memory, a requirement for tasks like inhibition of return (IOR) (Fig. 12a). Our ablation studies further showed that dense RNN layers improved attention accuracy across all tasks, even when not strictly necessary. In contrast, the COCO model was larger overall but did not include RNN layers (Fig. 12b).

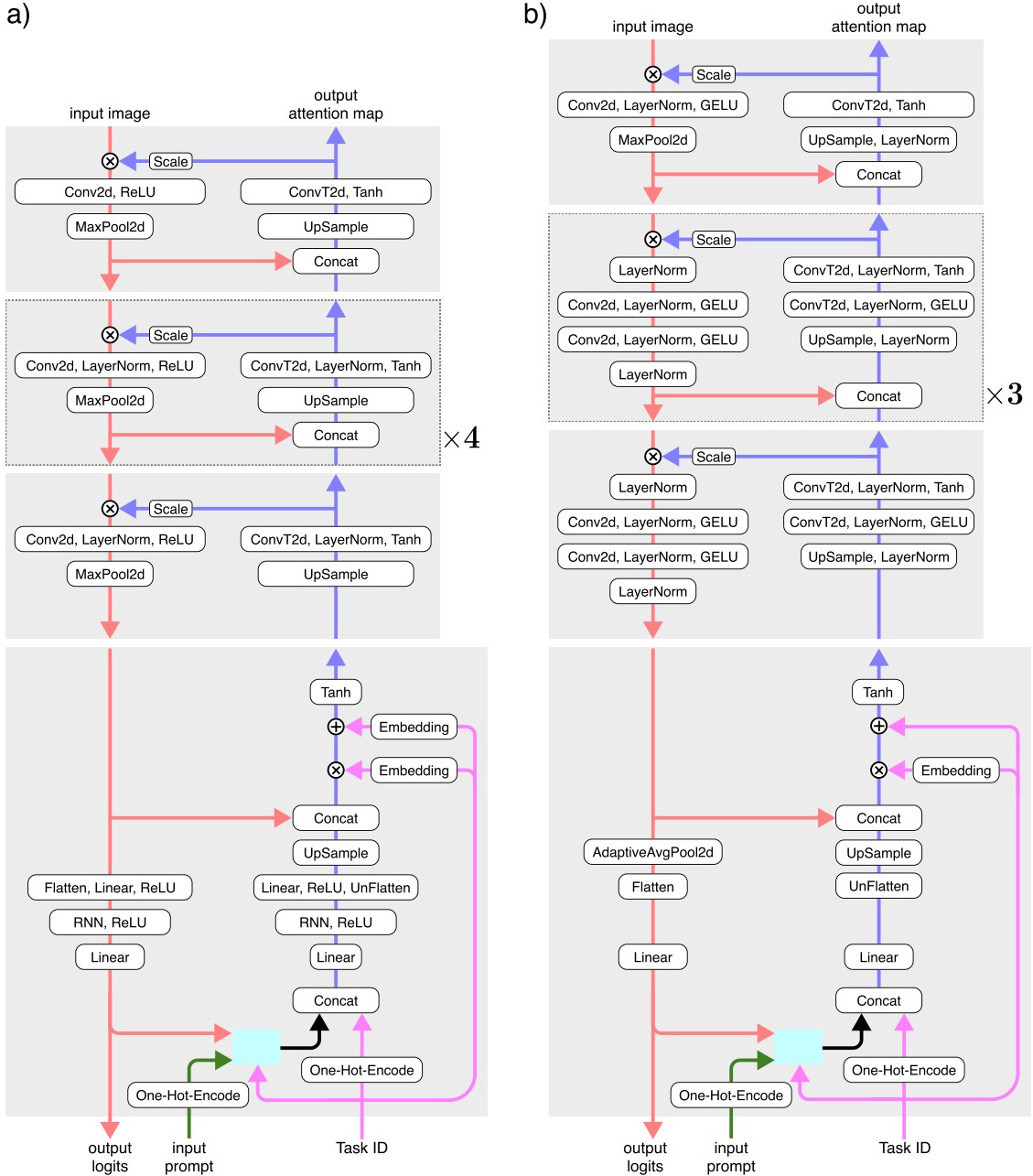

**Fig. 12 | Architecture backbone, building blocks and elements. a** Detailed architecture used for the MNIST experiment. For the MNIST model, we use RNN layers in the bottleneck. **b** Detailed architecture used for the COCO experiment. For the COCO model, we use stride of 2 in convolutional layers for downsampling instead of max-pooling.

## Evaluation

The code and trained models are available for validation and further investigation (see the Code Availability section). For classification tasks, we use cross-entropy loss. The MNIST dataset is sufficiently balanced, so no class weighting is applied. For COCO, however, we use class weights to mitigate the effects of imbalanced class distributions. In the CelebA experiment, although the dataset is also imbalanced, we deliberately avoid using class weights to highlight the network's ability to learn without prior knowledge of class distributions or spurious correlations. For segmentation tasks, we train using mean squared error (MSE) loss, but report results using attention accuracy and pixel error, as these metrics are more intuitive and easier to interpret.

**Attention accuracy.** Attention accuracy measures the normalized accuracy of a pixel being correctly attended to or masked, that is, whether the target and estimated pixel values have the same sign. To ensure consistency across samples and tasks with varying target-to-scene area ratios, we apply a normalization step such that the chance level is always 50% (e.g., when the entire scene is either attended or masked). This metric is conceptually similar to the Intersection over Union (IoU) commonly used in computer vision[164]. The following equation defines the measure:

$$\text{AttentionAccuracy} = 100\% \frac{1}{h \times w} \sum_{i,j}^{h,w} \sigma(\widehat{m}_{ij}, m_{ij}) \text{ where} : \sigma(a,b) = \begin{cases} 1 & \text{if Sign}(a) = \text{Sign}(b) \\ 0 & \text{otherwise} \end{cases}$$

$$(3)$$

where $\hat{m}_{ij}$ and $m_{ij}$ denote the $ij-$th pixel of estimated and true attention maps respectively, while $h$ and $w$ are the height and width of the attention map.

**Pixel error**. Pixel Error denotes the mean squared difference between estimated and true attention maps.

**Modulation index**. We use the Modulation Index (MI) similar, but not identical to, to those described in refs. 120 and 121:

$$\text{ModulationIndex} = \frac{T-D}{T+D} \qquad (4)$$

where $T$ and $D$ denote the averaged response (i.e. activity) of neurons to the target and distractor curves respectively.

**Bell-curve fitting**. For the invariant tuning experiment, we used the following method to accept or reject a tuning curve based on its shape, specifically, whether it is bell-shaped. Starting with the raw tuning curve $y$ of length 180 (i.e., defined over a domain of $0°$ to $179°$), we first apply a 1D Gaussian filter to smooth the curve. Next, we compute basic tuning attributes using simple algebra: the asymptote, amplitude, and preferred orientation. Based on the half-width at half-maximum (HWHM) of the curve, we estimate a reference Gaussian curve and calculate the Mean Euclidean Distance (MED) between this reference and the smoothed curve. We retain only those neurons for which both the target and distractor orientation tuning curves yield an MED smaller than 0.01. Please see the public code for the implementation and details.

**Preferred orientation**. To calculate the preferred orientation in the Figure-Ground Separation experiments, we used SciPy's least-squares curve fitting tool[162]. The target function is a cyclic Gaussian of the form:

$$y(\theta) = C + R_p \cdot \exp\left(-\frac{\sin^2(f \cdot (\theta - \theta_p))}{2\sigma^2}\right) \qquad (5)$$

Here, $C$ is the baseline offset, $R_p$ is the peak activity (i.e., the response at the preferred orientation), $f$ is the cyclic frequency, $\theta_p$ is the preferred orientation, and $\sigma^2$ is the variance. The fitting procedure minimizes the squared error to estimate the optimal values of $C, R_p, \theta_p$, and $\sigma$ for a given frequency $f$.

### Reporting summary

Further information on research design is available in the Nature Portfolio Reporting Summary linked to this article.

## Data availability

All datasets are either publicly available on their corresponding online repositories or the code to generate and compose them is on GitHub (https://github.com/ssnio/bio-attention). Shapes dataset: https://github.com/ssnio/bio-attention/tree/main/data. The MNIST dataset is publicly available on http://yann.lecun.com/exdb/. The MS-COCO dataset is publicly available on https://cocodataset.org/. The CelebA dataset is publicly available on https://mmlab.ie.cuhk.edu.hk/projects/CelebA.html although for non-commercial research purposes only. The CIFAR datasets are publicly available on https://www.cs.toronto.edu/~kriz/cifar.html. Source data are provided with this paper.

## Code availability

We used Python (version 3.9) as the primary programming language. The following libraries (modules) were used and are required to execute the analysis: PyTorch, NumPy, SciPy, Matplotlib, and Pillow. For the COCO experiment, the pycocotools module is necessary. The source code for all experiments and corresponding result notebooks are available on GitHub: https://github.com/ssnio/bio-attention and also upon request.

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

## Acknowledgements

This work was partly funded by the German Ministry for Education and Research (under refs 01IS14013A-E, 01GQ1115, 01GQ0850, 01IS18056A, 01IS18025A and 01IS18037A) and the German Research Foundation (DFG) within the RTG 2433 DAEDALUS. Furthermore, KRM was partly supported by the Institute of Information & Communications Technology Planning & Evaluation (IITP) grants funded by the Korea government (MSIT) (No. 2019-0-00079, Artificial Intelligence Graduate School Program, Korea University and No. 2022-0-00984, Development of Artificial Intelligence Technology for Personalized Plug-and-Play Explanation and Verification of Explanation).

## Author contributions

Saeed Salehi (S.S.), Jordan Lei (J.L.), Ari Benjamin (A.B.), Klaus-Robert Müller (K.R.M), Konrad Kording (K.K.) S.S., J.L. and K.K. conceptualized the model. S.S. and K.K. designed the experiments. S.S. performed the experiments, analyzed the data and prepared the figures. S.S., J.L., K.R.M, K.K. wrote the manuscript. All authors contributed to review and revision of the manuscript. K.R.M. and K.K. supervised and funded the project.

## Competing interests

The authors declare no competing interests.
