## [Transparent Peer Review file · Nature Communications]

Modeling Attention and Binding in the Brain through Bidirectional Recurrent Gating

Corresponding Author: Mr Saeed salehi

Version 0:

Reviewer comments:

Reviewer #1

(Remarks to the Author)

Salehi and co-authors aim at modelling attention in the brain by means of a deep neural architecture using bidirectional neural gating in the model. The training relies on backpropagation through time.

Modelling attention in deep neural networks (DNN) has not been very fruitful, as most DNNs can solve recognition and segmentation tasks sufficiently well without concepts of attention. Thus, it is certainly of interest how performance of DNN can be improved by brain-inspired attention mechanisms. However, my major concern is, that the manuscript does neither explain attention in the brain, even though the authors make this claim, nor does it provide a demonstration of the usefulness of attention in DNNs.

1) Developing a neural model of attention in the brain typically requires the following procedure. A) Anatomical and functional considerations will be incorporated into the design of the structure of the model. B) The model will be grounded in empirical data, such as behavioural or neurophysiological data. C) The model will be used to either predict new data or shed light on open issues, such as in conflicting data or to help understanding a research question by means of the complex interactions of mechanisms within a model.

I will now evaluate the manuscript on basis of these three points.

A) The proposed DNN is too abstract to allow for a good match with the anatomy of the brain. In particular, I became concerned of the “output attention maps”, which do not mimic any relevant part of the brain. Further, isn't the assumption of “Target Attention Maps” in the brain somewhat unrealistic? I am not aware of any brain area that represents such highly segmented objects. Area V1 only shows a mild attentional modulation, but no segmentation at all. Concepts of recurrent processing and gating are certainly biologically inspired, but the authors do not provide a clear description in how far their implementation maps well to at least abstract concepts of recurrent and top-down processing in the brain.

B) The authors do not make much effort to ground their model in experimental data. The section on behavioural results and task learning rather explores a few abstract principles without a clear replication of behavioural data taken from particular experimental conditions. Moreover, throughout the whole manuscript, references to models and to experimental data are mixed up without distinguishing between conclusions drawn from data and from particular model implementations. To give an example, already in the first paragraph providing an introduction about what attention is, models and data are combined in a very unreflected way. As far as the Bregman illusion is concerned, I fear different conditions of training have not been fully explored. It is not surprising to see a drop in performance, if previous information present during learning is not present during recall (invisible condition). I am missing a more solid elaboration in how far different training conditions will affect “the illusion”. Further, the authors just claim that they observe the illusion in the model without offering an explanation in how far the model can offer an explanation of the illusion in the brain. With respect to the neurophysiological results some effort is made to compare model neural activity with neural recordings. However, the curve tracking data and multiplicative attentional tuning data are not really the most informative data of attention in the brain, and I am not sure if the authors aim to suggest, that brain area V1 is comparable to their first model layer. More relevant data sets would comprise biased-competition experiments or dynamic receptive field shifts.

C) The authors do not use their model to explain brain function, but rather seem satisfied when their results can be interpreted in the context of demonstrating attention in the model. However, they do not elaborate on the function and purpose of attention in the brain. They do not answer any pressing research question in the field of attention by using their

model. For example, what can we learn from the model in the line tracking experiment about how attention is involved in line tracking? I did not find any hint about how this model can guide and improve understanding of attention in the brain.

In summary, the model rather mimics attention in the brain without explaining attention in the brain.

2) As far as the literature of models of attention is concerned, references to models from the field of computational neuroscience are largely ignored, as they “are generally not scaled to real-world problems and there are thus many behavioural and neural phenomena that current models cannot explain”, according to the view of the authors. However, I do not agree with this view. If models target the implementation and function of attention in the brain, they are supposed to be relevant even though some implementation details may vary. In particular, with regard to the proposed model a comparison among models can help better understanding how the brain may implement top-down, lateral and recurrent processing. Of course, Salehi and co-authors are particularly interested in computational models that operate on images. However, to my view, a comparison among different models is not sufficiently worked out in the present manuscript. A good reference on early models is the scholarpedia article “Computational models of visual attention” from Tsotsos.

Another overview about early and recent deep-learning models is:

de Santana Correia, A., Colombini, E.L. Attention, please! A survey of neural attention models in deep learning. *Artif Intell Rev* 55, 6037–6124 (2022).

A quite recent biologically motivated model of attention operating on images is:

Burkhardt M, Bergelt J, Gönner L, Dinkelbach HÜ, Beuth F, Schwarz A, Bicanski A, Burgess N, Hamker FH. A large-scale neurocomputational model of spatial cognition integrating memory with vision. *Neural Netw.* 2023 Oct;167:473-488.

Another recent brain-inspired model, but a bit more on the DNN side is:

Adeli, H., Ahn, S., & Zelinsky, G. (2023). A brain-inspired object-based attention network for multi-object recognition and visual reasoning. *Journal of Vision*, 23(5), 16-16.

I hope that the above references inspire the authors to better compare the mechanisms implemented in their model to those of other models. I am sure, despite of some differences, that there are certainly many similarities including the three key elements proposed by the authors.

3) Different models are used for different experiments, including different learning regimes. I understand that the authors aim to simplify and take different model versions in different tasks. However, this suggests that there is no unifying model.

4) The experimental studies and the function of the model shall be better explained. What are the exact conditions the model has to learn in each task? What properties in the architecture allow the model to learn those tasks? The labels of some results figures do not match well with the labels in Fig. 1, e.g. Target Label. What does pre-attention and post-attention refer to? Please provide for each experiment the particular loss function being used, including their final parameters (alpha, beta) after learning.

How could the manuscript be improved? If the authors aim at demonstrating their model being a good model of attention in the brain they may improve A, B and C (see above). They may also further elaborate better on the proposed three key elements of attention: top-down process, lateral processes and communication and recurrence. Even though they make this claim already in the third paragraph of their introduction, they do not elaborate the proposed key elements in their manuscript. They could better work out that indeed these three aspects are essential for explaining experimental data and discuss the role of each key element in more detail including a comparison of different implementations of these key elements in different models. However, bear in mind that many models of attention rely on recurrent and top-down processing and use mechanisms of neural gain-modulation. These terms are not new at all. Another direction would be that the authors rather focus on demonstrating their model being a good model of attention in DNNs, putting less emphasis on modeling the brain. So far, DNNs rather ignore methods of attention and it would be interesting to explore, what kind of tasks require attention in DNNs and which tasks could be solved without attention. In the present manuscript, it is only shown that the model can learn tasks, that are somewhat related to attention. This itself is not very surprising, given the strong learning ability of such DNNs. However, the ultimate question is, does the consideration of mechanisms of attention allow DNNs to improve in certain tasks?

(Remarks on code availability)

Reviewer #2

(Remarks to the Author)

Review of Modeling Attention and Binding in the Brain through Bidirectional Recurrent Gating

Summary: Salehi et al. propose a novel deep learning attention model. A key challenge with modeling attention is the ability to incorporate both feedforward and feedback (top-down) connectivity. Salehi et al overcome this challenge by developing a

model with a bidirectional recurrent gating mechanism. The model exhibits many known attentional phenomena on simple (MNIST) and more complex (COCO, CelebA) datasets. The results show that multi-task training enables the model to “attend” to relevant aspects of the input, thereby increasing its performance. Moreover, model neurons show effects associated with attention in biological neurons. The model is interesting, and several aspects of the results are well-substantiated.

There are a few places where the manuscript could be improved. I have listed my suggestions, both major and minor, below:

(Major Points)

Motivation:

1) A key effect of biological attention is the speeding up of reaction times. Showing the trained model performs faster (lesser number of ‘iter.’s) when using attention could strengthen the model’s biological plausibility. In other words, attention maps converge faster when this network is used than models without the Attention pathway.

2) The abstract and the Introduction mentions that the study replicates the neural phenomenon of “relatively late onset attention”, while none of the results appear to substantiate this claim (the word “late-onset” is not mentioned anywhere else in the manuscript). Could the authors clarify and elaborate on what they mean by this phenomenon? Are they referring to the fact that top-down attention takes longer to deploy?

3) The model does not currently appear to be able to fit behavioral or neural data. While many model architectures may be able to qualitatively explain behavioral patterns to varying degrees the ability to also quantitatively fit at least some aspects of behavior in actual experimental tasks would be a valuable addition. Similarly, quantitatively fitting some aspects of neural responses would also be an important addition.

4) More broadly, such an addition could also make the model widely relevant clinically, for example to help provide a basis for understanding mechanisms of attention disorders. This could enable quantifying inter-participant or inter-group differences in attention through their model, for example, by fitting the behavior of neurodivergent and neurotypical populations and comparing the model weights between the two groups. Such an extension could be discussed as motivation for such modeling.

Model architecture and training:

5) The paper is missing comparisons with other sota models:

a. Comparisons with existing models which explain fewer aspects of attention would strengthen the argument of multi-task training.

b. Showing that existing architectures like transformers or conv-lstms perform worse than the proposed model would provide stronger evidence for biological relevance of the model

6) The authors could clarify why the task signal passed to every layer in the Attention pathway. Is there evidence that all the different cortical regions receive such task information or is this a model-specific artifice to avoid task information being lost in the deeper layers?

7) A suggestion: In the MNIST experiments, why is the RNN layer used in the Attention pathway? RNN in the Feature pathway could hold the information required to perform loR-type tasks.

8) Could the authors provide the following details:

a. What was the “time” i.e., iter used for different tasks for different datasets?

b. How were the values for the hyper-weights α_i and β_i chosen for different tasks?

c. How was the model trained for the loR task? Line 176 (Figure 4b) states that the model is trained to “avoid returning to previously attended digits”. In the sequential classification task, the model must avoid attending to a formerly classified digit, which automatically forces it to attend to the remaining ones. Even though target labels and target attention maps are not provided wouldn’t BPTT guarantee this behavior?

9) Did the images used for training the model for Spatial cue task in the COCO dataset always comprise multiple animals? If not, the object recognition task could be trivial for the model to solve as it could have seen the target attention map in the Spatial cue task.

Results and inferences:

10) Could the authors elaborate further regarding figure 1b? What were the exact signals used for partially supervised training for different tasks? Specifically, isn’t object recognition always partially supervised?

11) For the MNIST pop-out task, how was the “correct” attention map determined? Was it a spotlight over the digit’s location or in the exact shape of the digit?

12) Attribution methods like CAM/grad-CAM are able to produce salience or attention heat maps despite not training the network to segment. Comparing the proposed model with these alternatives would strengthen its relevance to the machine learning community.

13) Attention-invariant tuning:

- a. Figure 10 – panel a: Could the authors clarify which layers these neurons were chosen from?
- b. Figure 10 – panel b-i: Because the model architecture uses multiplicative attention, this result appears trivial. Could the authors elaborate?
- c. Could the authors clarify how they quantified Gaussian-likeness for Figure 10?

14) Please clarify the target attention map for the first timestep for the task in Figure 17 d.

15) Could the authors further elaborate on the results and inferences for the multi-modal top-down visual search experiments (lines 722-725)?

16) On a related note, Figure 19 appears to be lacking a caption! Moreover, the second column shows that the model failed to capture “yellow”. Could the authors comment on why could that be?

(Minor Points) Typos and corrections:

- a. Figure 1 caption: duplicate word “see”
- b. Line 148: duplicate round bracket for figure reference & inconsistent style
- c. Line 206: specie -> species
- d. Figure 6: panels b and d are not mentioned in the main text
- e. Figure 8 caption: duplicate round brackets when referring to panel b-ii

(Remarks on code availability)

Reviewer #3

(Remarks to the Author)

This study presents a neural architecture integrating bottom-up and top-down visual processing through bidirectional recurrent gating. While the model successfully unifies multiple aspects of visual attention in a single framework - a notable advance over previous single-task approaches - several fundamental questions remain unaddressed.

Major points:

Multi-task Learning Analysis

The paper demonstrates multi-task capability but lacks critical analysis of its implications:

- * No comparison of performance between multi-task and equivalent single-task networks
- * No evaluation of computational efficiency gains/costs
- * No investigation of potential task interference or facilitation effects
- * No analysis of whether shared representations emerge across tasks

Without such comparisons, we cannot determine whether the multi-task approach offers meaningful advantages beyond demonstrating that a large network can learn multiple independent tasks.

Scalability Concerns

- * The reported 80% classification accuracy needs context: How many classes (I guess 10)? How does this compare to state-of-the-art?
- * What are the specific challenges for scaling this approach to larger datasets or more complex tasks?
- * How does computational cost scale with additional tasks or increased complexity?

Minor issues

- * The demonstrated behavior may not truly represent IOR, which requires temporal history effects. Alternative mechanisms could produce similar task performance.
- * The learning mechanism should be explicitly acknowledged as biologically implausible. The claim about observing moving objects providing supervised learning signals is problematic - this represents unsupervised learning.
- * While the model qualitatively reproduces human-like behaviors, quantitative comparisons to human performance are absent but this claim is made. The claim is incorrect and the comparisons would be stronger with a quantitative comparison (but I understand that the authors did not do this).
- * Fig 1. Variables are undefined (in this section) and chance level should be added to fig 1b).
- * Figure 2a-c: Consider adding training phase for clarity
- * Super et al. (2010) citation misrepresents the literature - Lamme & Roelfsema (2000) would better support the recurrent processing claim.
- * Task Implementation

Clarification needed: Is task cycling performed per stimulus? Can stimuli be randomized?

(Remarks on code availability)

well organised GitHub with clear notebooks. Most data is from other repositories and this is well referenced.

Version 1:

Reviewer comments:

Reviewer #1

(Remarks to the Author)

Salehi and co-authors aim at modelling attention in the brain by means of a deep neural architecture using bidirectional neural gating in the model. The training relies on backpropagation through time.

The clarity of the manuscript has improved significantly during the revision. In particular, the teaching and input signals used in the individual studies are now more comprehensible. Overall, the proposed model represents a substantial improvement over more traditional deep learning architectures, which often neglect attentional phenomena. However, I remain unconvinced that the current approach offers substantial insight into the mechanisms of attention in the brain.

While some of my earlier concerns have been addressed, others remain unresolved. My remaining concerns are as follows:

1) I criticized that in the introduction, models and data are combined in a very unreflected way. I still spotted such sentences in the revised manuscript which are:

“Empirical research reveals that selectively focusing on portions of the visual scene produces neural responses that are more localized, sparser, and less noisy [Posner et al., 1980; Tsotsos and Bruce, 2008].” These references do not support the claim. Posner et al. (1980) is a behavioral study, and Tsotsos and Bruce (2008) is a computational modeling paper. Neither analyzed neural responses.

„In the context of learning, behavioral evidence shows that focusing on task-relevant stimuli improves generalization [Rutishauser et al., 2004; Walther et al., 2005].“ These references are primarily computational. They do not provide behavioral evidence.

2) While the tone throughout the manuscript is mostly balanced, I find the following statement in the abstract to be an overstatement and recommend removing it: “Most importantly, our proposed model unifies decades of cognitive and neurophysiological findings of visual attention into a single principled architecture.” The model is a valuable and interesting contribution to computational modelling of attention. However, it cannot reasonably be described as a unification of decades of empirical findings. Such a claim overreaches the evidence presented and may raise unrealistic expectations.

3) A key concern remains that the model is used primarily to demonstrate attentional behaviors, rather than to explain them. The manuscript does not articulate how the model sheds light on the function or purpose of attention in the brain. It also does not appear to address any major open research questions in attention neuroscience.

In their response letter, the authors state their research question as: “How can a biologically inspired system successfully operate on real-world stimuli while exhibiting core attentional behaviors observed in humans?”

While this is indeed a legitimate research question, it is one focused on building a model, rather than one that can guide or be guided by experimental neuroscience. In its current form, the model does not clearly serve as a tool for hypothesis generation or for interpreting neural data. For example, while the model shows strong performance on grouping and segmentation tasks, this is likely a result of supervised training on pre-segmented data, rather than an emergent property rooted in biologically plausible mechanisms.

The discussion would benefit greatly from addressing how the model could inform future experiments or contribute to theoretical debates about attention in the brain.

In conclusion, the manuscript shows meaningful progress in the domain of modelling attention in deep neural networks. However, it still falls short in offering novel insights into the biological or cognitive mechanisms of attention. Strengthening the connection to neuroscience—either through more precise referencing or by framing the model in a way that supports experimental interpretation—would significantly enhance the impact of the work.

(Remarks on code availability)

Reviewer #2

(Remarks to the Author)

Salehi et al. present a novel model and introduce innovative methods for incorporating established attentional findings into a machine learning framework. The model exhibits many expected patterns of behavior, although it is not strongly motivated by brain architecture or evolutionary principles.

The authors have addressed many of my concerns through additional experiments and by providing more detailed descriptions of their methodology and I thank them for the careful response. The revised figures enhance clarity, and I appreciate their comprehensive outline of the training procedure for the IoR task.

These are a few additional suggestions, major and minor.

Major points:

1. It would help to target the model to address an important open question in the attention literature, to make systematic predictions about mechanisms of unexplained behavioral or neural phenomena or to provide an integrative or synthetic view of these diverse phenomena. Morgan et al (2025) provides a recent example of such a model:

<https://www.biorxiv.org/content/10.1101/2024.11.09.622721v1>

2. In symbolic reorienting task (MNIST), can the same digit appear more than once in an image during training? Please clarify.

3. In the top-down search task (MNIST), was it necessary to explicitly provide the target attention map? Given that the model is trained across multiple tasks, shouldn't it have developed an internal representation of what the digit '8' looks like and thus be capable of generating the attention map autonomously?

Minor points

1. Space after period: Lines 123 and 764

2. Figure 3a caption: the word 'trained' appears to be mistakenly included or misplaced

3. Bracket after panel number in captions of Figure 4, Figure 7 and Figure 26

4. Figure 6 caption: c-d -> c-e, bold f

5. Figure 8c) "purelyd" -> "purely"

6. Figure 10c) did the authors mean "F-neurons"

7. Line 379: B capitalized

8. Line 760: The word "Table" missing from the reference

9. Formatting: inconsistent casing across section titles and across figure titles

10. Figure 14: ";" placed incorrectly before 'width', fist -> first

11. Lines 845-846: better to refactor

12. Line 989: spacing after open bracket

13. Figure 24: caption for panel e is missing

14. Figure 25: disctractor -> distractor

15. Details of panels c-e could be added in Figure 29

(Remarks on code availability)

Reviewer #3

(Remarks to the Author)

I had some major points that the authors addressed throughly

multi-task learning analysis  added substantial experiments including single vs. multi-task training comparisons, transfer learning studies, and representational similarity analysis).

efficiency  scaling analyses showing how their approach scales with task count and complexity.

scaling  conducted experiments with CIFAR-100

and more (task interference, shared representations etc).

So in sum they have taken my comments and added substantial changes and updates.

The authors also show that certain tasks can be aligned or orthogonal to each other. I would be interested in reading in the discussion speculations on the source of this. Would it be possible to predict for two tasks if they are one or the other?

Beyond that I have no additional comments and enjoyed reading the paper very much. I think it is a very relevant step in the field.

Two final remark, I can imagine the authors are thinking about ablation studies in the future?

Also, is the repository up to date with this version of the submission? Looking at it, it might be of the last version?

Best, H.Steven Scholte

(Remarks on code availability)

The code has a readme file and clear instruction on how to run, and explanation of the repository structure and a tutorial on how to run everything.

It also has a demo.

I suspect (but have not run it) that the repository is of the first submission and not updated yet.

Version 2:

Reviewer comments:

Reviewer #1

(Remarks to the Author)

According to my opinion the authors have sufficiently exploited the capabilities of their approach. They showed that their model allows to replicate data from different experimental settings. However, to my view present deep neural network approaches are too abstract to capture the essence of visual attention, since they are lacking sufficient temporal dynamics and their match to particular brain areas is very limited. Thus, the focus is more on replication of data and less on explaining attention in the brain. Anyway, the authors have significantly improved the presentation of their results and overall, the manuscript provides a valuable contribution to the field of modelling attention.

(Remarks on code availability)

Reviewer #3

(Remarks to the Author)

The authors have fully addressed my questions and processed my comments. Questions that are still open are, in all reasonable consideration, as far as I would say, targets of future research.

And the authors make an excellent point. Given that the models are open, that future research can be done by many.

(Remarks on code availability)

This is a well documented GitHub resource.

Letter

Dear Editor,

We sincerely appreciate the reviewers' constructive and insightful comments.

In response to the reviewers' constructive and insightful feedback, we have made extensive revisions to our manuscript. To clarify the scope and intent of our study, we revised the introduction and discussion to emphasize that our goal is not to improve deep neural networks per se, but to use biologically inspired architectures to model and investigate mechanisms of visual attention. We clarified the experimental setups and training configurations across tasks, and revised nearly all figures and captions for enhanced clarity.

In response to Reviewer #1's call for more neurobiological grounding, we added a new figure-ground separation experiment inspired by recent neurophysiological work [Jeurissen et al., 2024] and improved our discussion of top-down, recurrent, and lateral processes in the brain. We also revised our references and comparisons to include a broader range of computational neuroscience models. At the suggestion of Reviewers #2 and #3, we conducted several new analyses, including learning with and without attention, multi-task vs. single-task comparisons, transfer learning evaluations, and a representational similarity analysis to examine shared features across tasks.

Overall, we believe the manuscript now offers a clearer, more rigorous, and well-contextualized contribution that effectively bridges neural plausibility with task performance. We are confident that the revised version will meet the journal's expectations, and we appreciate your time and effort in overseeing the review process.

Best regards,

Saeed Salehi, Prof. Klaus-Robert Müller, and Prof. Konrad P. Kording

Reviews

We would like to cordially thank the reviewers for the constructive and insightful comments on our paper. We feel that they have greatly helped us to improve the manuscript. In response to all reviewers comments we extensively rewrote the paper, added new analysis and computational experiments and clarifications. Our responses are in blue-colored text. The revised texts from the manuscript are in gray-colored text.

We do realize that our responses are long and detailed. To help the reviewers navigate the responses we want to start with a summary table here:

Category	Revision Summary and Key Changes
Modeling Intent	We now explicitly state that our goal is to explore biologically inspired attention mechanisms using DNNs as computational tools, not to simply achieve SOTA performance.
Biological Grounding	A new figure-ground separation experiment has been added to demonstrate border-ownership tuning and figure-ground segregation. In the Discussion, we clarify that the use of explicit target attention maps is a practical compromise given current dataset limitations.
Multi-Task Learning	Additional experiments now compare single-task versus multi-task training, including transfer learning studies and a Representational Similarity Analysis to demonstrate shared representations and task facilitation/interference.
Efficiency & Scalability	A new computational scaling analysis (Fig. fig_time) quantifies training time versus task count and recurrence iterations, showing sublinear cost growth. Memory requirements are also discussed.
Training Details	We now provide detailed explanations of the loss functions and hyperparameter values (α and β) used for each task in the supplementary materials. Figure captions and the Methods section have been revised for clarity regarding training sequences and supervision.
Structural Improvements	Figure labels and captions have been standardized with clear definitions of terms such as “pre-attentive” and “attentive.” We have reorganized the manuscript and added further references (e.g., Tsotsos, de Santana Correia et al., Adeli et al.) to better contextualize our work.
Transparency	Revised text excerpts and updated figures are included in our responses to demonstrate the depth of revisions and our commitment to fully addressing reviewer concerns.

Table 1: Executive Summary of Revisions. This table provides a concise overview of the major changes made in response to reviewer feedback.

Reviewer #1

Salehi and co-authors aim at modelling attention in the brain by means of a deep neural architecture using bidirectional neural gating in the model. The training relies on backpropagation through time.

We sincerely appreciate the reviewer’s valuable suggestions, and such detail and thorough review of the manuscript.

Modelling attention in deep neural networks (DNN) has not been very fruitful, as most DNNs can solve recognition and segmentation tasks sufficiently well without concepts of attention. Thus, it is certainly of interest how performance of DNN can be improved by brain-inspired attention mechanisms. However, my major concern is, that the manuscript does neither explain attention in the brain, even though the authors make this claim, nor does it provide a demonstration of the usefulness of attention in DNNs.

We acknowledge that our framing in the introduction may have been unclear. To clarify, our primary

goal is to investigate mechanisms of attention in humans, rather than improving deep neural networks per se. DNNs are used in our work primarily as a modeling tool to explore these mechanisms. We have revised the introduction to make this distinction more explicit. Here is the excerpt from the introduction: It is important to emphasize that our objective is not to enhance artificial neural networks per se, but to use them as computational instruments for exploring attention mechanisms inspired by the brain [Doerig et al., 2023; B. A. Richards et al., 2019].

1) Developing a neural model of attention in the brain typically requires the following procedure. A) Anatomical and functional considerations will be incorporated into the design of the structure of the model. B) The model will be grounded in empirical data, such as behavioural or neurophysiological data. C) The model will be used to either predict new data or shed light on open issues, such as in conflicting data or to help understanding a research question by means of the complex interactions of mechanisms within a model.

I will now evaluate the manuscript on basis of these three points.

In the following the reviewer sets up a set of desiderata for modeling attention in the brain. We largely agree with the setting. To better respond to some of the comments, we have conducted an elaborate experiment inspired by Jeurissen et al., 2024 to motivate the role of top-down processing in attention, and more specifically, figure-background separation. This experiment is now added to the main text of our manuscript. We would like to thank the reviewer for their comments, which encouraged us to make this very valuable addition to our study. We discuss how our model is meaningful in factors A, B and C and how we edited the manuscript to clarify this.

Figure-Background Separation

Building upon recent hypotheses that unify border ownership and figure-ground segregation in primate vision [Jeurissen et al., 2024; Self et al., 2019], we designed an experiment to investigate how our network performs figure-background separation. Specifically, we aimed to test whether our model implements figure-ground segregation via a similar feedback mechanism—positive feedback aligned with preferred figure location and negative feedback for background—as recently proposed by Jeurissen and colleagues.

To that end, we trained a neural network incorporating our bidirectional gating mechanism on a multi-modal classification task involving shape, color, and texture, without explicit attention supervision (see supplementary materials for details). From this trained network, we selected two distinct groups of neurons, each consisting of 128 units from the same bidirectional recurrent gating block: one group from layer 5 of the feature pathway (F-neurons), and another from layer 4 of the attention pathway (A-neurons) (Fig. 1a). We then determined the receptive fields of the F-neurons within the input space using methods described by [Araujo et al., 2019], as illustrated by the dotted red box in Fig. 1b.

We then created two stimuli: stimulus (i), consisting of a novel object (i.e., a new shape) with solid texture and color on an empty background (Fig. 1b, stimulus i); and stimulus (ii), consisting of the same shape without texture or color (i.e., blank shape), placed on a background matching the object’s color from stimulus (i) (Fig. 1b, stimulus ii). We chose the object’s shape carefully so that, regardless of rotation, it always occupied exactly half of the receptive field, and ensuring that stimulus (i) rotated by θ° would always appear identical, from the perspective of the F-neurons, to stimulus (ii) rotated by $\theta + 180^\circ$ (Fig. 1c, top row).

Finally, we analyzed the tuning curves and neural activities of the selected F- and A-neurons. We hypothesize that the pre-attentive activity of F-neurons should be object-agnostic, given that their receptive fields do not cover the entire visual scene. Consequently, the tuning curves of F-neurons responding to stimulus (i) should closely match those for stimulus (ii), but with a 180° phase shift (Fig. 2a, c). In contrast, A-neurons exhibited markedly different behavior: their activities for both stimuli were highly similar and aligned in phase according to their preferred orientations (Fig. 2b, d). Additionally, neurons in the attention pathway consistently showed strong positive modulation at their preferred orientations compared to non-preferred orientations, irrespective of stimulus type (Fig. 2e).

Our primary analyses, as illustrated in Fig. 2, demonstrate that our computational model exhibits neuronal behaviors closely aligned with these biological findings. Specifically, neurons within our model similarly separate into two major groups: (1) feature-pathway neurons (F-neurons), displaying orientation-

Figure 1: Stimuli and receptive field. **a)** Schematic of the network architecture used for the multi-modal classification task, highlighting both the feature and attention pathways. Colored circles indicate the locations of neurons analyzed in this study. **b)** Top: Receptive field of neurons from the feature pathway (F-neurons), indicated by the dashed red box. Bottom: Two stimuli used for analysis: stimulus (i), a colored object over an empty background; and stimulus (ii), a colored background (matching the object color in stimulus (i) containing an uncolored stencil object). **c)** From the receptive field's viewpoint, rotating stimulus (i) by θ° makes it perceptually identical to rotating stimulus (ii) by $\theta + 180^\circ$. The network successfully generates accurate attention maps across all stimuli variations, achieving 91% attention accuracy.

Figure 2: Figure-Ground Separation. **a)** Polar plot showing the activity of an example F-neuron as a function of stimulus rotation. Dotted squares outside the plot illustrate receptive field content for stimulus (i) (red) and stimulus (ii) (blue). **b)** Activity of an example A-neuron as a function of stimulus rotation, with receptive field content indicated in the same way. Because the attention pathway receives input from the bottleneck, A-neurons have receptive fields spanning the entire input scene. **c)** Histogram illustrating the distribution of preferred orientation phase differences for A-neurons between the two stimuli. **d)** Histogram showing preferred orientation phase differences of A-neurons between the two stimuli. **e)** Average activity of A-neurons at their preferred versus non-preferred orientations, shown for both stimulus types.

tuned responses analogous to biological Ori-tuned neurons (Fig. 2a, c); and (2) attention-pathway neurons (A-neurons), which are selectively tuned to object directions, thus mirroring the behavior of

biological BO-tuned neurons (Fig. 2b, d, e). Furthermore, A-neurons integrate inputs from higher-level areas and adjacent F-neurons, subsequently providing top-down modulation to earlier layers. Together, these findings suggest our architecture successfully learns and implements figure-ground segregation mechanisms comparable to those described in primate visual cortex.

Here is the excerpt from the supplementary materials:

Figure-Ground Separation

We trained a network based on bidirectional gating mechanism on a multi-modal classification task involving shape, color, and texture, without any attention supervision. Each input image contained a single object defined by three independent features: shape (triangle, square, circle, cross, hexagon, pentagram, heart, hexagram, and crescent), color (red, green, blue, yellow, cyan, and magenta), and texture (salt-and-pepper noise, soft-splotchy patterns, structured patterns, and solid fill). The object was randomly positioned on a background with color and texture drawn from the same feature distribution as the object (Fig. 3a). To ensure that figure-ground separation was always possible, we designed the dataset such that the object and background differed in at least one feature—color, texture, or both.

This setup allowed us to examine whether bottom-up processing alone (i.e., purely feedforward) is sufficient for this task, and whether top-down processing provides a functional advantage. We trained three versions of the network independently: (1) a feedforward model using only the feature pathway (i.e., a single forward pass) trained on samples from set (i); (2) an attention-based model incorporating both feature and attention pathways, also trained on set (i) and (3) a control condition in which a feedforward model was trained on set (iii), images without backgrounds—i.e., no figure-ground separation required. All models shared the same feature pathway architecture to ensure a fair comparison. Additionally, three specific feature combinations—triangle + red + salt & pepper, square + green + structured, and circle + blue + splotchy—were excluded from the training set to evaluate generalization.

Our results show that the attention-based model not only learns faster and achieves higher classification accuracy (Fig. 3b), but also generalizes better to unseen feature combinations (Fig. 3c). Moreover, despite being trained solely for classification, the attention-based network produces plausible and interpretable attention maps—even for novel, randomly generated shapes (Fig. 3a). Finally, the control case confirms that the inferior performance of the feedforward model is not due to hyperparameter choice, but rather the presence of distracting background features.

A) The proposed DNN is too abstract to allow for a good match with the anatomy of the brain.

We agree that our model is abstract and does not currently capture many anatomical details of visual attention—most notably, it lacks mechanisms for overt attention. This abstraction is intentional: our primary goal was to develop a minimal model of biological visual attention capable of solving non-trivial tasks on reasonably complex stimuli. Although the full range of anatomical features relevant to visual attention is still not completely understood, the architectural choices in our model are grounded in well-supported hypotheses from neuroscience. For example, we use convolutional neural networks to model feedforward hierarchical processing in the visual system [Lindsay, 2021], we incorporate multiplicative attentional modulation [McAdams and Maunsell, 1999] via scaled feedback signals. Our emphasis on top-down, bottom-up, and recurrent processing reflects the substantial evidence for these interacting pathways in the brain [Kar et al., 2019; Kietzmann et al., 2019; Lamme and Roelfsema, 2000]. While the precise anatomical implementation of these mechanisms remains unclear, we view our model as one possible formalization of how such interactions might be orchestrated. Finally, we respectfully disagree on the idea that abstraction in modeling is inherently problematic. Computational neuroscience has a long tradition of using abstract models to gain insight into brain function. In that spirit, our model aims to strike a balance between biological inspiration and computational tractability, contributing to a conceptual understanding of attention mechanisms rather than anatomical fidelity.

In particular, I became concerned of the “output attention maps”, which do not mimic any relevant part of the brain. Further, isn’t the assumption of “Target Attention Maps” in the brain somewhat unrealistic? I am not aware of any brain area that represents such highly segmented objects. Area V1 only shows a mild attentional modulation, but no segmentation at all.

Figure 3: Multi-Modal Classification task. **a)** Sample composites (top row) and corresponding attention maps (bottom row) generated by the attention-based model. (Note: input image colors are inverted for better visualization.) The model is trained on samples from set (i). Set (ii) contains samples where the object and background share at least one feature (i.e., same color or texture). Set (iii) includes images without backgrounds. Set (iv) includes randomly generated novel shapes. **b)** Validation cross-entropy loss and accuracy during training. The attention-based model converges faster and achieves higher validation accuracy than the feedforward model trained on the same dataset. Dotted curves indicate performance of the feedforward model trained on the objects on blank background. **c)** Shape recognition accuracy across different test sets for models with and without attention. The attention-based network generalizes better to out-of-distribution samples and performs more robustly on ambiguous stimuli (set ii).

We thank the reviewer for raising this important point. We agree that our use of binary object segmentation maps as training targets departs from biologically plausible learning mechanisms. We have acknowledged this limitation in the Discussion. Our revised text reads:

We have demonstrated that the proposed multitask learning paradigm can reduce reliance on full supervision for certain tasks. Nonetheless, our model still requires supervised signals, including segmentation maps that lack a direct biological analogue. We acknowledge that the use of abstract, binary object segmentation masks in some tasks is a departure from biologically plausible learning, and their inclusion reflects practical constraints—specifically, the absence of richer sensory input and suitable datasets that could otherwise guide learning in a more naturalistic way.

That being said, we demonstrate that in recognition tasks such as shape classification and CelebA attribute recognition, our model is able to localize relevant regions without explicit supervision via

segmentation maps. This suggests that even in the absence of strongly supervised attention signals, the model can learn to attend appropriately. Regarding the biological plausibility of segmented object representations, we agree that no single brain area encodes fully segmented object maps in the way our output maps suggest. However, there is substantial evidence of attentional modulation contributing to perceptual grouping and segmentation-like processes—even in early visual areas—through mechanisms such as border ownership, figure-ground segregation, and contextual modulation [Jeurissen et al., 2024; Lamme, 1995; Self et al., 2019; Super et al., 2010; von der Malsburg, 2024].

Concepts of recurrent processing and gating are certainly biologically inspired, but the authors do not provide a clear description in how far their implementation maps well to at least abstract concepts of recurrent and top-down processing in the brain.

We could have been clearer in explaining how these elements map onto abstract neurobiological concepts. To address this point and comments made in the next paragraphs, we performed the figure-ground separation experiment (See *Figure-Background Separation*). Our experiment clearly shows the emergence of Border-Ownership (BO) tuned cells in the attention pathway, similar to BO-tuned cells in V4 recently reported by [Jeurissen et al., 2024]. Here we reiterate the paragraphs we added to the main text in response to this comment:

Our primary analyses, as illustrated in Fig. 2, demonstrate that our computational model exhibits neuronal behaviors closely aligned with these biological findings. Specifically, neurons within our model similarly separate into two major groups: (1) feature-pathway neurons (F-neurons), displaying orientation-tuned responses analogous to biological Ori-tuned neurons (Fig. 2a, c); and (2) attention-pathway neurons (A-neurons), which are selectively tuned to object directions, thus mirroring the behavior of biological BO-tuned neurons (Fig. 2b, d, e). Furthermore, A-neurons integrate inputs from higher-level areas and adjacent F-neurons, subsequently providing top-down modulation to earlier layers. Together, these findings suggest our architecture successfully learns and implements figure-ground segregation mechanisms comparable to those described in primate visual cortex.

Furthermore, our model expands upon the hypothesis that the ventral visual stream comprises two distinct yet interconnected pathways: a feature pathway, progressing hierarchically from V1 to higher visual areas such as IT, responsible for parsing visual scenes into increasingly abstract and robust representations [Lamme and Roelfsema, 2000; Roelfsema, 2023]; and an attention pathway, receiving global scene and task-relevant information from higher visual areas, which modulates lower-level neurons via top-down signals. Our results reinforce this hypothesis by demonstrating that neurons within these pathways exhibit properties analogous to biological observations [Jeurissen et al., 2024; Klink et al., 2017; Self et al., 2019]: feature-pathway neurons display orientation-tuned responses similar to Ori-tuned neurons in the primate visual cortex, while attention-pathway neurons reflect border-ownership tuning behavior, selectively enhancing neural responses to preferred object directions.

B) The authors do not make much effort to ground their model in experimental data. The section on behavioural results and task learning rather explores a few abstract principles without a clear replication of behavioural data taken from particular experimental conditions.

We acknowledge that our initial manuscript may not have clearly communicated the extent to which our model is grounded in experimental findings. We have revised the Discussion to better highlight these connections and have also added a new experiment that directly addresses this concern. Specifically, we now include an experiment demonstrating the role of top-down modulation in figure-ground separation, which qualitatively mirrors recent neurophysiological findings [Jeurissen et al., 2024; Self et al., 2019]. This experiment highlights the role of bottom-up attentional signals in border-ownership and figure-ground separation.

More broadly, our model is grounded in experimental data and hypotheses in the following key ways:

- 1) Architectural principles: The model architecture is based on decades of neurophysiological evidence emphasizing the importance of top-down and recurrent processes in visual attention and perceptual binding [Kar et al., 2019; Kietzmann et al., 2019; Lamme and Roelfsema, 2000; Treisman, 1998].
- 2) Mechanism of attention: We implement attention via top-down modulation of bottom-up features, reflecting known mechanisms of neural gain control and feature-specific enhancement observed in primate vision [McAdams and Maunsell, 1999; Roelfsema et al., 1998]. Moreover, the newly added experiment on border ownership and figure-ground separation offers a concrete demonstration of how our model

can replicate phenomena seen in experimental paradigms, further strengthening the bridge between our computational approach and empirical data.

3) Behavioral relevance: we tried to implement many known tasks covering all the three aspects of attention: *orienting, filtering, and searching* [Ward, 2008].

We stress that our work is not only inspired by abstract principles but also connects with and build upon established experimental findings in visual neuroscience. While our work takes important steps toward bridging computational modeling and experimental findings, we agree that further work is needed to more directly align model behavior with specific experimental paradigms and datasets.

Moreover, throughout the whole manuscript, references to models and to experimental data are mixed up without distinguishing between conclusions drawn from data and from particular model implementations. To give an example, already in the first paragraph providing an introduction about what attention is, models and data are combined in a very unreflected way.

We have extensively revised the manuscript to ensure a clearer and more consistent distinction between empirical findings and model-based conclusions. We now follow a more structured approach when discussing models versus experimental data, explicitly indicating the source of each claim. These changes aim to improve transparency and interpretability throughout the manuscript. Here is our revised introduction:

Attention is widely regarded as a mechanism by which the brain selects a meaningful subset of incoming stimuli for perception, learning, or memory [Carrasco, 2011; Hommel et al., 2019; Itti et al., 2005; Lindsay, 2020; Maunsell, 2015; Moore and Zirnsak, 2017; Tsotsos, 2021]. Empirical studies suggest that by focusing on a subset of the world, attention can produce more localized, sparse, and less noisy neural responses to sensory input [Posner et al., 1980; Tsotsos and Bruce, 2008]. In the context of learning, behavioral evidence shows that focusing on task-relevant stimuli improves generalization [Rutishauser et al., 2004; Walther et al., 2005]. Similarly, in memory, attention appears to facilitate the encoding and reuse of efficient representations—for example, by emphasizing reusable object parts or reducing signal interference [Kruijne et al., 2021; Olivers and Roelfsema, 2020]. These findings underscore the central role of attention in shaping perception, learning, and memory. Hence, empirical evidence suggests that attention plays an indispensable role in neural information processing, and understanding attention is critical to advancing our comprehension of learning, memory, and perception.

What makes attention particularly challenging to model is the diversity of phenomena it encompasses. Experimental research has identified different forms of attention that act across multiple scales and modalities [Buschman and Kastner, 2015; Itti et al., 2005; Maunsell, 2015; Moore and Zirnsak, 2017; Posner, 2016]. One well-studied form is spatial or "spotlight" attention, in which focus is directed to a region of space, enhancing perception and neural processing in that region [Itti and Koch, 2001; Wolfe and Horowitz, 2004]. Behavioral experiments demonstrate that perception within the attended region tends to be faster and more accurate [Chun and Wolfe, 2005; Posner, 1980]. Feature-based attention, another extensively studied variant, leverages prior knowledge about relevant stimulus features to enhance perception [Bichot et al., 2015; Treisman, 1998; Treisman and Gelade, 1980]. Object-based attention, arguably related to "feature-based" attention, involves the holistic selection of objects, including all their associated spatial and visual features [Baldauf and Desimone, 2014; Behrmann et al., 1998; Scholl, 2001; Treisman, 1998]. A fundamental question here is how an object that is represented by activity across millions of neurons and many cortical regions can be perceived and attended to as a single entity [Von der Malsburg, 1999]. This is known as the "binding problem", which refers to how the brain integrates activity patterns related to the same object and separates them from others [Feldman, 2013; Robertson, 2003; Singer, 2001; Treisman, 1996; Von der Malsburg, 1995]. Understanding object-based attention is thus viewed as a critical step toward addressing the broader problem of feature binding in neuroscience [Roelfsema, 2023; Treisman, 1998].

Mechanistically, attention is believed to rely on a set of interacting, complex processes. The first is the *feedforward (bottom-up)* processing, which progresses from lower to higher areas of the visual cortex through hierarchical, mainly unidirectional connections, and establishes the foundational tuning and receptive field properties of visual neurons [Lamme and Roelfsema, 2000]. The second is the *top-down process*, wherein higher-order brain regions modulate activity in early visual areas—a mechanism shown experimentally to influence object perception, integration, and biased competition [Beck and Kastner, 2009; Freeman et al., 2003; Gilbert and Li, 2013; Klink et al., 2017; Lamme and Roelfsema, 2000; Paneri and Gregoriou, 2017; Thiele and Bellgrove, 2018]. The third is *lateral processing*, which enables

context-sensitive interactions such as perceptual grouping, contour integration, and noise reduction [Field et al., 1993; George et al., 2020; Gray et al., 1989; Han et al., 2005]. The fourth essential mechanism is *recurrence*, or iterative processing, which allows information to propagate bidirectionally across layers and time. Experimental and theoretical studies alike have emphasized the importance of recurrence in visual processing and attentional dynamics [Kar et al., 2019; Kietzmann et al., 2019; O’Reilly et al., 2013; Roelfsema, 2023; van Bergen and Kriegeskorte, 2020].

As far as the Bregman illusion is concerned, I fear different conditions of training have not been fully explored. It is not surprising to see a drop in performance, if previous information present during learning is not present during recall (invisible condition). I am missing a more solid elaboration in how far different training conditions will affect “the illusion”.

Our interpretation of the illusion is based on the assumption that object recognition in the brain is learned, and that such learning typically occurs under natural conditions—where occlusion is caused by physical, visible objects. From this perspective, the network’s reduced performance under the "invisible occluder" condition is consistent with expectations and reflects the absence of cues normally available during learning. To better examine this, we performed additional analysis, which is now included in the main text.

To further test whether the observed behavior is simply due to an out-of-distribution effect, we created a new set of test stimuli—distinct from the training set—that maintains a similar ratio of visible to invisible object regions (Fig. 4e). Since the invisible occlusion in the control set is organized as a grid, we do not expect lower classification accuracy, despite the invisible occlusion.

Figure 4: Bregman’s illusion. Bregman’s illusion is commonly used to demonstrate the concept of border ownership. **a-b)** Original illustration of the Bregman’s illusion, illustrating that visible ink blot helps with recognition of the letters. **c-d)** Similarly, visible occlusion in our experiment seems to help with recognition task and recovering the digit’s boundaries. Out-of distribution control stimuli, where the background is a random color, and the invisible occlusion is organized as a grid rather than noisy splotchy texture. **f)** Although visible occlusion appears to initially (i.e., pre-attention) hinder classification accuracy, attention helps the model to integrate occlusion boundaries to achieve higher performance. Note: Colors for all input sequences have been inverted to improve visualization.

Furthermore, the high classification accuracy on the control stimuli, supports our claim that the illusion is not solely a result of unfamiliar input, but instead arises from the absence of expected structural features. We will further investigate border-ownership in the next section.

Further, the authors just claim that they observe the illusion in the model without offering an explanation in how far the model can offer an explanation of the illusion in the brain.

To address this point and comments made previously, we performed an elaborate analysis on figure-ground separation and border ownership (See *Figure-Background Separation*). We would like to thank the reviewer for this comment, since it encouraged us to make this very valuable addition to our study.

With respect to the neurophysiological results some effort is made to compare model neural activity

with neural recordings. However, the curve tracking data and multiplicative attentional tuning data are not really the most informative data of attention in the brain, and I am not sure if the authors aim to suggest, that brain area V1 is comparable to their first model layer. More relevant data sets would comprise biased-competition experiments or dynamic receptive field shifts.

We agree that the curve-tracing and multiplicative tuning paradigms are not definitive ways to characterize attention in the brain. However, we chose these paradigms because they have been widely used in seminal studies of visual attention and provide a useful starting point for assessing attentional modulation in our model [Roelfsema et al., 1998].

Our intention was not to claim a one-to-one mapping between specific brain areas (e.g., V1) and particular layers of our model, but rather to show that certain attentional modulation patterns observed in the brain—such as gain-like effects and task-dependent tuning changes—are also present in our architecture. We acknowledge that stronger comparisons could be made using datasets from biased-competition paradigms or dynamic receptive field shifts, and we see this as a valuable direction for future work.

Additionally, our model builds on a growing body of evidence showing that convolutional neural networks can learn hierarchical representations that align with neural activity in the visual cortex, from V1 through higher-order areas like V4 [Khaligh-Razavi and Kriegeskorte, 2014; Schrimpf et al., 2018; Yamins and DiCarlo, 2016]. While we have not conducted a formal representational similarity analysis here, we believe our model exhibits a similar hierarchical structure, with early layers more comparable to V1-like representations and deeper layers more aligned with higher visual areas such as V4.

C) The authors do not use their model to explain brain function, but rather seem satisfied when their results can be interpreted in the context of demonstrating attention in the model. However, they do not elaborate on the function and purpose of attention in the brain. They do not answer any pressing research question in the field of attention by using their model. For example, what can we learn from the model in the line tracking experiment about how attention is involved in line tracking? I did not find any hint about how this model can guide and improve understanding of attention in the brain.

We respectfully disagree with the notion that our model does not contribute to understanding attention in the brain. In our view, one of the central and long-standing questions in attention research is: How can a biologically inspired system successfully operate on real-world stimuli while exhibiting core attentional behaviors observed in humans? This is precisely the question we set out to explore. Our model is not intended to explain every mechanistic detail of brain function, but rather to offer a functional account of attention that bridges neuroscience-inspired principles with task-relevant behavior. By constructing a model that performs real-world tasks while exhibiting features such as top-down modulation, figure-ground segregation, and perceptual grouping, we aim to constrain and test hypotheses about the functional principles of attention and how it may emerge from interacting computational mechanisms. In the case of the line-tracking experiment, for example, our model demonstrates how attention can propagate along a spatially extended structure—an effect reminiscent of curve tracing in psychophysical and physiological studies. While we do not claim a direct mechanistic match, the model provides a computational substrate in which such behaviors can be formalized and probed. We recognize that the model does not answer all open questions in attention research, but we believe it offers a step toward building more comprehensive, interpretable systems that can serve as platforms for exploring the function of attention under more plausible conditions.

In summary, the model rather mimics attention in the brain without explaining attention in the brain.

While we acknowledge that our model does not provide a full mechanistic explanation of attention in the brain, our goal is to develop a functional, biologically inspired model that can operate on real-world tasks while exhibiting key attentional behaviors observed in humans and animals. By doing so, we aim to explore how attention might emerge from the interaction of known computational principles, and can help generate and test hypotheses about attentional function under more biologically plausible conditions. We have now made the limitations of our approach more explicit in the Discussion section, to better contextualize the scope and interpretation of our findings. Here is our revised discussion addressing this comment:

Despite being grounded in prior theory, our model extends earlier frameworks in several important ways. First, whereas many classic attention models were demonstrated only on simplified stimuli or isolated

tasks—and often struggled to scale to complex, natural input—our model is designed to operate on naturalistic scenes, successfully recognizing and attending to objects in real-world images. It thereby addresses a key limitation of earlier computational models, which were often rich in theory but not scalable to real-world problems. Second, our architecture unifies the three canonical attention behaviors—orienting, filtering, and search—within a single model. This multitask embedding goes beyond prior models that typically tackled these attentional functions in isolation. Finally, by incorporating biologically inspired circuit mechanisms—notably top-down feedback, recurrence, lateral communication, divisive normalization, and neuromodulatory gating—our model replicates various neurophysiological findings, such as attention-mediated neuromodulation, attention-invariant tuning, and border-ownership tuning. By integrating these elements, our model achieves a level of biological plausibility and functional breadth that distinguishes it from earlier models. Computationally, we showed that our model can leverage attention to accelerate learning in the presence of ambiguous backgrounds and correctly mask spurious features in the input, leading to more robust predictions. This work shows that bidirectional recurrent gating is a powerful mechanism for modeling and understanding attention.

Furthermore, our model expands upon the hypothesis that the ventral visual stream comprises two distinct yet interconnected pathways: a feature pathway, progressing hierarchically from V1 to higher visual areas such as IT, responsible for parsing visual scenes into increasingly abstract and robust representations [Lamme and Roelfsema, 2000; Roelfsema, 2023]; and an attention pathway, receiving global scene and task-relevant information from higher visual areas, which modulates lower-level neurons via top-down signals. Our results reinforce this hypothesis by demonstrating that neurons within these pathways exhibit properties analogous to biological observations [Jeurissen et al., 2024; Klink et al., 2017; Self et al., 2019]: feature-pathway neurons display orientation-tuned responses similar to Ori-tuned neurons in the primate visual cortex, while attention-pathway neurons reflect border-ownership tuning behavior, selectively enhancing neural responses to preferred object directions. Naturally, our model has limitations in its ability to fully capture the breadth of neural and behavioral phenomena. For instance, it does not incorporate overt attention mechanisms such as saccades and foveation, which are essential components of human visual attention [Itti et al., 2005]. Fortunately, effective models of these mechanisms exist and could, in principle, be integrated into our framework [Mnih et al., 2014]. Additionally, our current implementation of inhibition of return (IOR) is achieved through an explicit training objective rather than biologically grounded mechanisms such as peripheral stimulation and oculomotor activation [Klein, 2000]. However, given the recurrent nature of our network, we believe it can be adapted to incorporate these dynamics. Clearly, while our model describes a range of effects, there are still many findings it cannot yet quantitatively account for. Nonetheless, we believe that the recurrent and modulatory nature of our architecture provides a strong foundation for incorporating these dynamic and mechanistic aspects in future extensions. While our model captures a wide range of attention-related behaviors and neural correlates, there remain many findings it does not yet quantitatively explain—highlighting both its current value and its potential as a foundation for further development.

2) As far as the literature of models of attention is concerned, references to models from the field of computational neuroscience are largely ignored, as they “are generally not scaled to real-world problems and there are thus many behavioural and neural phenomena that current models cannot explain”, according to the view of the authors. However, I do not agree with this view. If models target the implementation and function of attention in the brain, they are supposed to be relevant even though some implementation details may vary. In particular, with regard to the proposed model a comparison among models can help better understanding how the brain may implement top-down, lateral and recurrent processing. Of course, Salehi and co-authors are particularly interested in computational models that operate on images. However, to my view, a comparison among different models is not sufficiently worked out in the present manuscript. A good reference on early models is the scholarpedia article “Computational models of visual attention” from Tsotsos.

Another overview about early and recent deep-learning models is: de Santana Correia, A., Colombini, E.L. Attention, please! A survey of neural attention models in deep learning. *Artif Intell Rev* 55, 6037–6124 (2022).

A quite recent biologically motivated model of attention operating on images is: Burkhardt M, Bergelt J, Gönner L, Dinkelbach HÜ, Beuth F, Schwarz A, Bicanski A, Burgess N, Hamker FH. A large-scale neurocomputational model of spatial cognition integrating memory with vision. *Neural Netw.* 2023 Oct;167:473-488.

Another recent brain-inspired model, but a bit more on the DNN side is: Adeli, H., Ahn, S., & Zelinsky, G. (2023). A brain-inspired object-based attention network for multi-object recognition and visual reasoning. *Journal of Vision*, 23(5), 16-16.

I hope that the above references inspire the authors to better compare the mechanisms implemented in their model to those of other models. I am sure, despite of some differences, that there are certainly many similarities including the three key elements proposed by the authors.

In the revised manuscript, we now cite the suggested works—including the Scholarpedia article by Tsotsos, the survey by de Santana Correia et al. (2022), and the brain-inspired model by Adeli et al. (2023). We have added a more comprehensive overview of the current landscape of attention modeling, highlighting both classical computational neuroscience models and recent developments in brain-inspired deep learning. We also explicitly discuss how the three key elements of our model—top-down, lateral, and recurrent processing—relate to mechanisms proposed in prior models. Many of our architectural choices can be seen as extensions or scalable implementations of earlier ideas, and we now make these connections more explicit in the text. We hope these additions better contextualize our contributions within the broader modeling literature and help clarify the relationship between our approach and existing theoretical frameworks. Here is our revised text from the introduction with respect to this comment:

Given the wide range of attention-related phenomena and the multiple interacting mechanisms involved, modeling attention presents a significant challenge. Yet, building computational models that integrate these components is essential for understanding how attention operates at both cognitive and neural levels [Chun et al., 2011; Tsotsos, 2021]. Over the past decades, numerous models have been proposed, each offering different perspectives on how attention may be implemented in the brain. Computational models of visual attention from neuroscience can be broadly grouped into four paradigmatic hypotheses [Tsotsos and Rothenstein, 2011]: selective routing, saliency map, temporal tagging, and emergent attention. *Selective routing* models conceive of attention as a controlled routing of information through the visual hierarchy (e.g., gating signals that direct a subset of inputs to higher areas) [Fukushima, 1986; Olshausen et al., 1993; Tsotsos et al., 1995]. *Saliency-map* models compute attention by integrating multiple feature maps into a topographic saliency map, where the most conspicuous location wins the competition for attentional selection using a winner-take-all mechanism [Itti and Koch, 2001; Koch and Ullman, 1984]. *Temporal tagging* models exploit time dynamics (oscillations or synchrony) to bind or select attended stimuli [Deco et al., 2002; Grossberg, 1987; Grossberg and Grossberg, 1982; Milner, 1974; Von Der Malsburg, 1994]. In contrast to all of the above, emergent attention models posit that attentional phenomena arise intrinsically from competitive interactions within a large neural system rather than from a single explicit mechanism [Heinke and Humphreys, 2003; Srivastava et al., 2024; Styles, 2006; Thorat et al., 2021]. Although empirical evidence favors each of these hypotheses to varying degrees [Shadlen and Movshon, 1999; Shipp, 2004; Tsotsos et al., 2005], they collectively reflect the underlying complexity of biological attention.

3) Different models are used for different experiments, including different learning regimes. I understand that the authors aim to simplify and take different model versions in different tasks. However, this suggests that there is no unifying model.

We understand the concern and agree that the manuscript could have been clearer on this point. Our goal was to propose a unifying mechanism and architectural framework, rather than a single fixed model instance. While different tasks required slight adaptations in learning regimes or configurations, these were all implemented within the same underlying architecture and with shared principles. To address this, we have now made our intent more explicit in the main text. Additionally, we have added a new section in the Supplementary Materials that outlines how the proposed mechanism can be instantiated across different tasks and datasets, emphasizing its flexibility while preserving architectural consistency.

4) The experimental studies and the function of the model shall be better explained. What are the exact conditions the model has to learn in each task? What properties in the architecture allow the model to learn those tasks? The labels of some results figures do not match well with the labels in Fig. 1, e.g. Target Label. What does pre-attention and post-attention refer to? Please provide for each experiment the particular loss function being used, including their final parameters (alpha, beta) after learning.

In response, we have significantly revised the manuscript to improve the clarity of our experimental descriptions, model functionality, and figure labeling.

- We now provide a clearer explanation of the exact training conditions for each task, including the data used, the supervision available, and any task-specific variations in training regime.
- All figure labels, including references such as "pre-attention" and "post-attention" (Fig 5) have been revised to ensure consistency with Fig. 1 and across the manuscript. We also provide clearer definitions of these terms in the figure captions and main text.
- For each experiment, we now specify the loss functions used, along with their component weights (e.g., α , β). These are summarized in a new table in the supplementary materials.
- Additionally, we have added a supplementary section detailing how to build and configure the model from scratch, to further support reproducibility and clarity.

Figure 5: **Pre-attentive and attentive features.** The model processes the input images over multiple iterations. In the first iteration, it receives a flat, task- and input-agnostic attention map, resulting in feature representations and class predictions that are considered *pre-attentive*. In subsequent iterations, the model incorporates context- and task-dependent information into attention maps, producing *attentive* features and predictions.

We hope these improvements address the reviewer’s concerns and enhance the interpretability of the model and experimental results.

How could the manuscript be improved? If the authors aim at demonstrating their model being a good model of attention in the brain they may improve A, B and C (see above). They may also further elaborate better on the proposed three key elements of attention: top-down process, lateral processes and communication and recurrence. Even though they make this claim already in the third paragraph of their introduction, they do not elaborate the proposed key elements in their manuscript. They could better work out that indeed these three aspects are essential for explaining experimental data and discuss the role of each key element in more detail including a comparison of different implementations of these key elements in different models. However, bear in mind that many models of attention rely on recurrent and top-down processing and use mechanisms of neural gain-modulation. These terms are not new at all. Another direction would be that the authors rather focus on demonstrating their model being a good model of attention in DNNs, putting less emphasis on modeling the brain. So far, DNNs rather ignore methods of attention and it would be interesting to explore, what kind of tasks require attention in DNNs and which tasks could be solved without attention. In the present manuscript, it is only shown that the model can learn tasks, that are somewhat related to attention. This itself is not very surprising, given the strong learning ability of such DNNs. However, the ultimate question is, does the consideration of mechanisms of attention allow DNNs to improve in certain tasks?

We thank the reviewer for this comprehensive and detailed feedback throughout. In response, we have substantially revised the manuscript to clarify both the scope and the contributions of our work. We now more explicitly frame our model as offering a biologically inspired mechanism—rather than a full anatomical account—for attention, grounded in three core elements: top-down modulation, lateral communication, and recurrence. These elements are now described and discussed in greater depth, with clearer justification for their inclusion based on experimental data and prior modeling work. We also added comparisons to existing models that incorporate similar mechanisms, including those in the computational neuroscience literature, and now cite several of the suggested references to better situate our approach. We have restructured the presentation of our experiments to more clearly articulate the learning conditions, architectural contributions, and loss functions used in each task. While we acknowledge that we build on existing concepts, our goal is to demonstrate how a unified, multi-task model grounded

in these well-established principles can operate effectively on real-world stimuli. Finally, while our focus remains on biologically inspired modeling, we also discuss the implications of our findings for deep neural networks more broadly, including the potential of attention mechanisms to enhance task performance in structured, multi-object visual environments.

Reviewer #2

Review of Modeling Attention and Binding in the Brain through Bidirectional Recurrent Gating

Summary: Salehi et al. propose a novel deep learning attention model. A key challenge with modeling attention is the ability to incorporate both feedforward and feedback (top-down) connectivity. Salehi et al overcome this challenge by developing a model with a bidirectional recurrent gating mechanism. The model exhibits many known attentional phenomena on simple (MNIST) and more complex (COCO, CelebA) datasets. The results show that multi-task training enables the model to “attend” to relevant aspects of the input, thereby increasing its performance. Moreover, model neurons show effects associated with attention in biological neurons. The model is interesting, and several aspects of the results are well-substantiated.

We sincerely appreciate the reviewer’s positive feedback, valuable suggestions, and thorough review of the manuscript.

There are a few places where the manuscript could be improved. I have listed my suggestions, both major and minor, below:

Major points

Motivation:

1) A key effect of biological attention is the speeding up of reaction times. Showing the trained model performs faster (lesser number of ‘iter.’s) when using attention could strengthen the model’s biological plausibility. In other words, attention maps converge faster when this network is used than models without the Attention pathway.

We thank the reviewer for this suggestion. However, we are not currently aware of a comparable baseline model without the attention mechanism that would allow for a fair and direct comparison in terms of convergence speed. While this does not directly address the reviewer’s suggestion, we have instead aimed to highlight a complementary benefit of attention in our model, that attention improves learning. The new experiment illustrates how attention facilitates more efficient learning, offering another dimension of biological relevance of attention. Here is our new experiment added in the supplementary materials inspired by the comment above:

Attention improves learning

Our hypothesis is that feedforward neural networks—i.e., networks lacking an attention pathway—perform well when the foreground (target object) can be easily segregated from the background. In such cases, bottom-up feature integration is often sufficient to marginalize irrelevant background information. However, when the background shares similar features with the foreground, we expect feedforward networks to struggle with accurate classification.

To test this, we constructed several composite datasets derived from CIFAR-100 images [Krizhevsky, Hinton, et al., 2009], where the background for each target object is composed of patches from other images (Fig. 6a). The composite datasets are structured as follows: (Set i): The target image is randomly placed among 16×16 patches from randomly selected distractor images (Fig. 6i). (Set ii): Similar to Set i, but the target image itself is divided into four patches (without shuffling), and all patches (including the target) are visually separated by noise frames (Fig. 6ii). (Set iii): The background is composed of 1×1 pixel patches randomly sampled from eight distractor images, effectively creating a noisy background (Fig. 6iii). (Set iv): All background patches come from the same image, repeated eight times, resulting in a more coherent background (Fig. 6iv). (Set v): Spatial cues are presented before the main stimulus (for 2 iterations), while the composites could be taken from any of the previous sets (i–iv) (Fig. 6v). This set is only used with the attention-based model.

We trained three separate networks, each using the same backbone architecture and all trained solely for classification (i.e., no attention loss was applied—even for the model with attention pathway). The three training scenarios were as follows: (0) A control network without attention, trained on composites from Set iii (Fig. 6iii); (1) A second feedforward network, trained on composites from Set ii (Fig. 6ii); and (2) A network with attention pathway, trained on Set ii composites with a mix of samples both with and without spatial cues (Fig. 6ii, v). For validation and testing, we used composites from Set i to assess

Figure 6: Image composites with varying background complexity. For better visualization, we used STL-10 images in the figure, as CIFAR-100 images are lower in resolution. Although here the targets are centered in samples, during training and evaluation the targets were randomly placed. **i)** Target image is intermixed with large patches of random distractor images from the same dataset. **ii)** all the patches, including the target image are visually separated with noise frames. **iii)** the patches of distractor images are of size 1×1 , effectively creating a noisy background. **iv)** the patches of distractor images are from the same image, repeated 8 times. **v)** the image composite is preceded by 2 iterations of spatial cue.

generalization under consistent conditions. Our results show that the network with attention not only converges faster during training (Fig. 7a), but also achieves higher classification accuracy and greater robustness to background noise (Fig. 7b). Moreover, the control condition confirms that the reduced performance in feedforward networks stems from the presence of ambiguous background features, rather than suboptimal hyperparameters (Fig. 7a).

Figure 7: Attention improves learning. **a)** Validation accuracy and cross-entropy (CE) loss during training. The attention-based model—trained on Set ii with a mix of samples both with and without spatial cues—achieves higher accuracy and faster convergence compared to the model without attention. The black dotted line shows the performance of a control model (feedforward-only) trained on Set iii. Chance-level performance is indicated by the red dashed line. **b)** The attention-based model also generalizes better to out-of-distribution samples from Set iv.

2) The abstract and the Introduction mentions that the study replicates the neural phenomenon of “relatively late onset attention”, while none of the results appear to substantiate this claim (the word “late-onset” is not mentioned anywhere else in the manuscript). Could the authors clarify and elaborate on what they mean by this phenomenon? Are they referring to the fact that top-down attention takes longer to deploy?

The term “relatively late onset attention” describes the delay observed before attention takes effect—specifically, the time required for the system to form pre-attentive representations and the signal to reach higher layers, and to enable the top-down neuro-modulation [Lamme and Roelfsema, 2000]. This is aligned with the idea that top-down attention, in contrast to bottom-up processes, takes longer to deploy due to its reliance on contextual or higher-level information ([Zipser et al., 1996]). To avoid ambiguity, we

have now removed the term “relatively late onset attention” from the manuscript and replaced it with “pre-attentive representations” when appropriate. We also removed the term from the abstract and used “border-ownership tuning” to emphasize the newly added results and experiment.

3) The model does not currently appear to be able to fit behavioral or neural data. While many model architectures may be able to qualitatively explain behavioral patterns to varying degrees the ability to also quantitatively fit at least some aspects of behavior in actual experimental tasks would be a valuable addition. Similarly, quantitatively fitting some aspects of neural responses would also be an important addition.

We agree that the ability to quantitatively fit behavioral and neural data would significantly enhance the impact and applicability of our model. In the current work, our primary focus was on capturing a range of visual attention behaviors in a biologically inspired yet relatively simple architecture. While these behaviors are demonstrated qualitatively, we acknowledge that adding quantitative comparisons—particularly with behavioral benchmarks or neural recordings—would strengthen the model’s relevance. Achieving this, however, would likely require incorporating additional time-dependent mechanisms, such as local dynamic tuning [Kubilius et al., 2019] or distributed working memory systems [Christophel et al., 2017], which are currently beyond the scope of our present model. We believe such extensions would help the model perform better on quantitative metrics like Brain Score [Schrimpf et al., 2018]. Additionally, while our newly added figure-ground segregation experiment further supports the biological plausibility of the model, we see this as a stepping stone toward more detailed quantitative validations, which we aim to explore in future work.

4) More broadly, such an addition could also make the model widely relevant clinically, for example to help provide a basis for understanding mechanisms of attention disorders. This could enable quantifying inter-participant or inter-group differences in attention through their model, for example, by fitting the behavior of neurodivergent and neurotypical populations and comparing the model weights between the two groups. Such an extension could be discussed as motivation for such modeling.

While our current study did not directly aim to model or reproduce visual impairments or attention disorders, we agree that this is an exciting and valuable direction for future research. Extending our model to fit behavioral data from both neurotypical and neurodivergent populations—potentially by comparing differences in model parameters or learned representations—could provide important insights into the neural mechanisms underlying attention deficits. Although such modeling lies beyond the scope of the current manuscript, we now include the following paragraph in the Discussion section to emphasize this possibility and encourage future exploration: Crucially, incorporating attention into models of visual cognition is increasingly recognized as essential for bridging biological and artificial neural systems, particularly in the context of studying visual impairments [Schiatti et al., 2023]. While our model does not currently include overt attention mechanisms such as gaze shifts, it offers a promising framework for investigating deficits rooted in covert attention or feature-integration processes. We appreciate the reviewer’s suggestion, which has helped us better articulate the broader relevance and future potential of our work.

Model architecture and training:

5) The paper is missing comparisons with other sota models: a. Comparisons with existing models which explain fewer aspects of attention would strengthen the argument of multi-task training. b. Showing that existing architectures like transformers or conv-lstms perform worse than the proposed model would provide stronger evidence for biological relevance of the model

We appreciate the reviewer’s important point regarding comparisons with existing state-of-the-art models. However, we would like to clarify that the primary goal of this work is not to achieve state-of-the-art performance, but rather to develop a biologically inspired model that captures a wide range of attention-related behaviors across both cognitive and neural domains. We have revised the manuscript to more explicitly state this focus. Specifically, our objective is to build a unified model that can perform tasks typically given to humans and animals in laboratory settings, while also aligning with known neural mechanisms of attention—rather than optimizing for benchmark performance. As such, we see our contribution as complementary to heuristic architectures such as transformers or conv-LSTMs, which may excel in performance metrics but often lack biological interpretability. We now explicitly highlight this

distinction in the Introduction and Discussion section to clarify the scope and positioning of our work. Here is our revised text from introduction:

In this paper, we introduce a biologically inspired model of attention that integrates *bottom-up*, *top-down*, and *lateral interactions* through a *bidirectional recurrent gating* mechanism. Our modeling approach subscribes to the emergent-attention tradition, hypothesizing that a neural network with the right architecture and complexity, equipped with the fundamental components of attention [Knudsen, 2007], and trained on reasonable objectives would result in "attentive behavior". Our framework is informed by a wide body of experimental evidence supporting the role of top-down modulation and recurrence in perception, figure-ground segregation, visual search, and binding [Bouvier and Treisman, 2010; Cavanagh et al., 2023; Freeman et al., 2003; Kar et al., 2019; Kietzmann et al., 2019; Lamme and Roelfsema, 2000; Posner, 2023; Roelfsema, 2023; Self et al., 2015; Wolfe and Horowitz, 2017]. The proposed architecture builds on U-Net [Ronneberger et al., 2015], augmented with biologically motivated components: *divisive normalization* [Reynolds and Heeger, 2009] via layer normalization [Ba et al., 2016], *working memory* using dense recurrent layer [Elman, 1990; Hochreiter and Schmidhuber, 1997; Jordan, 1997], attention-driven *neuromodulation*, and top-down context inputs. It is important to emphasize that our objective is not to improve on artificial neural networks per se, but to use them as computational instruments for exploring attention mechanisms inspired by the brain [Doerig et al., 2023; B. A. Richards et al., 2019]. Our multitask-embedding framework supports flexibility across various attention-related tasks. These include object recognition, perceptual grouping through spatial cueing, symbolic orienting, pop-out, figure-ground separation, inhibition of return, and top-down visual search—covering all three canonical attention axes of *orienting*, *filtering*, and *searching* [Ward, 2008]. We also scale the model to naturalistic images (e.g., from COCO and CelebA). Finally, we present multiple experiments to assess whether the model’s internal representations resemble known neural patterns such as attention-invariant tuning, gain modulation, and border ownership. Together, our results suggest that this architecture provides a computationally tractable framework for unifying a wide range of behavioral and neural findings related to object-based attention.

6) The authors could clarify why the task signal passed to every layer in the Attention pathway. Is there evidence that all the different cortical regions receive such task information or is this a model-specific artifice to avoid task information being lost in the deeper layers?

We agree that the widespread propagation of the task signal across all layers may not have strong biological justification and could be seen as a modeling convenience. Based on this feedback, we re-evaluated our approach and conducted additional experiments. For the MNIST model, we re-trained the network with the task signal provided only at the top layer. This change led to no loss in test performance, while also reducing model complexity and the number of parameters. We have updated the manuscript to reflect this change. For the COCO model, our original setup already limited the task signal injection to the bottleneck and top layers. All new experiments in the revised manuscript now follow this more biologically plausible configuration, with task information introduced only at higher levels of the attention pathway. We appreciate the reviewer’s suggestion, which helped us improve both the interpretability and efficiency of the model.

7) A suggestion: In the MNIST experiments, why is the RNN layer used in the Attention pathway? RNN in the Feature pathway could hold the information required to perform IoR-type tasks.

The RNN layer was originally included in the Attention pathway for architectural symmetry, but we agree that including recurrence in the Feature pathway could also plausibly support information retention relevant for IoR-like tasks. To explore this, we implemented the reviewer’s suggestion by removing the RNN from the Attention pathway and instead placing it in the Feature pathway. While overall classification performance remained comparable, we observed a noticeable drop in attention accuracy. This suggests that recurrence in the Attention pathway may play a distinct role—potentially by maintaining a memory of previously attended locations, which facilitates better integration of attention over time. We have added this observation to the manuscript. We appreciate the reviewer’s input, which helped us further examine and clarify the functional contributions of each pathway RNN.

8) Could the authors provide the following details:

a. What was the “time” i.e., iter used for different tasks for different datasets?

b. How were the values for the hyper-weights α_i and β_i chosen for different tasks?

We added this information to the text and supplements. We also included a more elaborate explanation on the IOR experiment.

c. How was the model trained for the IoR task? Line 176 (Figure 4b) states that the model is trained to “avoid returning to previously attended digits”. In the sequential classification task, the model must avoid attending to a formerly classified digit, which automatically forces it to attend to the remaining ones. Even though target labels and target attention maps are not provided wouldn’t BPTT guarantee this behavior?

A similar point was also raised by Reviewer #3. We agree that inhibition of return (IoR) in the brain is likely arising from mechanisms such as peripheral stimulation, oculomotor activation, or both [Klein, 2000]. As such, our explicit implementation of IoR does differ from the biological mechanisms that give rise to this effect. That said, we believe that incorporating such mechanisms into our model is feasible and could be achieved without major structural changes. In our current implementation, the model is explicitly trained to perform IoR behavior. We have now added a detailed description of the training process to the supplementary materials for clarity:

IOR Experiment Training the Inhibition of Return (IOR) task requires a more nuanced approach compared to other tasks, where each input sequence typically has a single target label and attention map, allowing for straightforward optimization via cross-entropy (CE) and mean squared error (MSE) losses. In the IOR task, however, each input contains multiple target objects, and we explicitly aim to avoid imposing any fixed order of attention. This necessitates a training strategy that supports flexible attention sequencing while still enforcing accurate classification and attention for each object.

We describe the training procedure for a 3-object composite input, where the network is given 2 iterations per object, resulting in 6 iterations total. The input sequence—comprising repeated presentations of the same 3-object image—is fed to the network, which produces a sequence of predicted class labels and attention maps, one per iteration. Up to this point, the process mirrors that of other tasks. The key difference lies in how the loss is computed.

Specifically, we extract the outputs from the 2nd, 4th, and 6th iterations. At the 2nd iteration, we compare the predicted attention map to all three target attention maps using a sum-of-differences metric and identify the object with the closest match (i.e., the one most attended to by the network). This object is designated as the first target, and its corresponding label and attention map are used to compute the CE and MSE losses for that iteration. At the 4th iteration, we exclude the first selected object and compare the predicted attention map to the remaining two targets. The closest match defines the second attended object, which is used for computing losses. The process is repeated at the 6th iteration, where only one object remains. This dynamic matching approach enables the network to learn a valid attention sequence without enforcing a fixed order, while implicitly guiding it to avoid re-attending to previously selected objects—thereby capturing the core behavior of inhibition of return.

We acknowledge this important distinction in the revised discussion and appreciate the reviewer’s suggestion, which highlights the valuable directions for more biologically grounded future modeling:

Additionally, our current implementation of inhibition of return (IOR) is achieved through an explicit training objective rather than biologically grounded mechanisms such as peripheral stimulation and oculomotor activation [Klein, 2000].

9) Did the images used for training the model for Spatial cue task in the COCO dataset always comprise multiple animals? If not, the object recognition task could be trivial for the model to solve as it could have seen the target attention map in the Spatial cue task.

You are correct that not all images used in the Spatial Cue task contained multiple animals. However, if the model were overfitting or “cheating” by memorizing target attention maps, this would be reflected in poor generalization performance on the test set. To address this, we ensured that the final results—both qualitative (visualizations) and quantitative (metrics)—are reported solely on a held-out test subset. Importantly, none of the test images were seen by the model during training or validation, across any of the tasks. Thus, any potential shortcut the model could have exploited during training would not carry

over to the test evaluation. The model’s consistent performance on the test set suggests that it is not relying on memorized attention patterns, but rather learning a generalizable solution.

Results and inferences:

10) Could the authors elaborate further regarding figure 1b? What were the exact signals used for partially supervised training for different tasks? Specifically, isn’t object recognition always partially supervised?

We agree that the term “partial supervision” required further clarification, particularly in the context of object recognition tasks. To address this, we have added additional explanatory text in the main manuscript to clearly define what we mean by “partial supervision” for each task. In brief, object recognition in our setup is indeed always partially supervised in the sense that the model is provided with the class label but not an explicit attention map. In contrast, for some tasks—such as spatial cueing—the training includes both classification and attention loss signals. We now explicitly state which signals are used for training each task. Additionally, we have revised all the main figures, including Figure 1b, to more clearly indicate the supervision provided per task.

11) For the MNIST pop-out task, how was the “correct” attention map determined? Was it a spotlight over the digit’s location or in the exact shape of the digit?

For the MNIST pop-out task, we did not use any target attention map during training—the model learned to perform the task without explicit supervision on where to attend. However, for evaluation purposes, we did compare the model’s attention output to a target attention map to quantify accuracy. We have now made this distinction clearer in both the main text and figure captions. Additionally, we’ve updated the relevant figures—including Figure 8—to explicitly indicate which tasks used evaluation-only attention targets and how those were defined.

Figure 8: Multitask training on MNIST composites (Part 2/3). Results for a single model trained on seven tasks simultaneously. The figure includes input and output signals, as well as the target signals. Target signals that are not used for training (e.g., attention maps for pop-out task and class labels for top-down search task), but only for evaluation, are framed by dashed outlines and marked by the subscript *evaluation*. Here we present the results for: a) *pop-out saliency*, and b) *top-down visual search*.

12) Attribution methods like CAM/grad-CAM are able to produce salience or attention heat maps despite not training the network to segment. Comparing the proposed model with these alternatives would strengthen its relevance to the machine learning community.

We chose not to include comparisons with attribution methods such as CAM or Grad-CAM, because these techniques serve a fundamentally different purpose from the attention mechanisms used in our model. Attribution methods are typically applied post hoc to interpret or visualize the decision-making process of already trained networks. In contrast, our model uses attention as an integral part of the computation itself—modulating representations during training and inference to support task performance. We were

also concerned that drawing direct comparisons might lead to confusion, as it could imply a conceptual equivalence between post-training saliency explanations and biologically inspired attention mechanisms. Nevertheless, we acknowledge that attribution-based saliency maps can sometimes appear visually similar to attention maps.

13) Attention-invariant tuning:

a. Figure 10 – panel a: Could the authors clarify which layers these neurons were chosen from?

Yes. We added this information in the figure’s caption and the text. We have also included a new analysis and plot to the Figure 9. that further elaborates on the claim that neurons in deeper layer tend to have a more complex tuning curve compared to those in earlier layers.

Figure 9: Attention-invariant tuning. a) Orientation tuning curves of four neurons from layers 1, 3, 5, and 7 (from left to right) of our model trained on the curve-tracing task. For the analysis of the attention-invariant tuning, we only consider neurons with Gaussian-like tuning curves (e.g., the two plots on the left) and reject the rest. b) The ratio of neurons that respond to the target bar (i.e., the target bar is in their receptive field) increases in deeper layers, while their orientation tuning curves become more complex (i.e., less Gaussian-like). c) Attention increases the response amplitude but has little to no effect on the width, asymptote, and preferred orientation of the tuning curves. c-i) Our results from the penultimate feature layer of the model compared to c-ii) findings from V4 cortical neurons in macaque monkeys [McAdams and Maunsell, 1999]. The red arrows and values show the median. The graphs are styled similarly to those in [McAdams and Maunsell, 1999].

b. Figure 10 – panel b-i: Because the model architecture uses multiplicative attention, this result appears trivial. Could the authors elaborate?

The reviewer is correct that the use of multiplicative attention could make the observed effect appear straightforward or even expected. However, our choice of multiplicative modulation was deliberate—motivated by findings in neuroscience suggesting that attention operates through gain-like, multiplicative effects on neuronal responses. While this mechanism might suggest a predictable outcome,

we believe the results remain meaningful for several reasons. First, the effects we report emerge from the penultimate (deepest convolutional) layer of the network, which is shaped by a cascade of preceding nonlinear transformations. Given this depth and complexity, it is not trivial to guarantee or analytically predict that the multiplicative modulation will yield the desired attentional behavior at that stage. Second, by experimentally validating these effects in our biologically inspired architecture, we reinforce the relevance of this mechanism for building interpretable and plausible models of attention. We have clarified this motivation and added further explanation in the revised text.

c. Could the authors clarify how they quantified Gaussian-likeness for Figure 10?

The method used to quantify Gaussian-likeness in Figure 10 is described in the Supplementary Materials. In response to the reviewer’s comment, we have now expanded the explanation in the supplement to make the procedure more transparent. Briefly, we selected this method because it provides a more robust estimate compared to approaches like cyclic-Gaussian curve fitting. The updated supplement now includes details on the metrics and fitting procedure used.

For the invariant tuning experiment, we used the following method to accept or reject a tuning curve based on its shape—specifically, whether it is bell-shaped. Starting with the raw tuning curve y of length 180 (i.e., defined over a domain of 0° to 179°), we first apply a 1D Gaussian filter to smooth the curve. Next, we compute basic tuning attributes using simple algebra: the asymptote, amplitude, and preferred orientation. Based on the half-width at half-maximum (HWHM) of the curve, we estimate a reference Gaussian curve and calculate the Mean Euclidean Distance (MED) between this reference and the smoothed curve. We retain only those neurons for which both the target and distractor orientation tuning curves yield an MED smaller than 0.01. The following code calculates the curve attributes—namely amplitude, asymptote, width, and preferred orientation—and returns the mean Euclidean distance between the given tuning curve and a corresponding bell-shaped curve with matching parameters.

14) Please clarify the target attention map for the first timestep for the task in Figure 17 d.

For the task shown in Figure 17d, as with all experiments in this section, the model was trained solely using classification loss—no target attention map was provided during training, including at the first timestep. The network learns to generate attention maps implicitly, as a means of optimizing task performance. We have now made this clearer in both the figure caption and the main text, explicitly stating that attention in this case emerges as a learned internal mechanism rather than being guided by supervised attention targets.

15) Could the authors further elaborate on the results and inferences for the multi-modal top-down visual search experiments (lines 722-725)?

16) On a related note, Figure 19 appears to be lacking a caption! Moreover, the second column shows that the model failed to capture “yellow”. Could the authors comment on why could that be?

Thank you for your comment. Based on your feedback and interest in the multi-modal top-down visual search experiment, we have re-implemented the experiment with three modalities and significantly expanded the corresponding section in the manuscript:

We trained a model based on the Bidirectional Recurrent Gating (BRG) mechanism to simultaneously perform recognition and top-down visual search across three modalities: shape, color, and texture. Each input prompt specifies one target class for each modality—resulting in a three-part search query for shape, color, and texture. Input images are composed of 16 objects arranged in a 4×4 grid (Fig. 10a). The shapes consist of nine distinct classes: triangle, square, circle, cross, hexagon, pentagram, heart, hexagram, and crescent. Colors are drawn from six categories: red, green, blue, yellow, cyan, and magenta. Textures are generated from three stochastic processes: Salt-and-pepper, sampled from a normal distribution, Soft-spotchy, generated via correlated noise, and Structured pattern, produced by repeating a randomly sampled kernel to form a pattern. Due to the randomness in texture generation, instances from the same texture class can vary significantly (Fig. 10b). While the full combinatorial space includes $9 \times 6 \times 3 = 162$ unique objects, our multi-modal approach reduces the effective search complexity. By concatenating the one-hot encodings of the three target categories into a single 18-dimensional prompt vector, the network performs targeted search across only $9 + 6 + 3 = 18$ individual labels. Each input image may contain zero, one, or multiple instances matching the specified combination of features. The model is trained to locate and attend only to objects that satisfy all three target properties (Fig. 10c–e).

For the top-down search task, the model is trained on three-iteration input sequences. The training objective is consistent with previous tasks: minimizing the cross-entropy loss for the recognition output and minimizing the mean squared error (MSE) between predicted and target attention maps, given both the input image and the associated prompt. The model achieves 99% accuracy in selecting the correct object during the top-down search task, with a precision of 94%. For the recognition task, the network reaches 99% classification accuracy across all three modalities (shape, color, and texture) and 97% attention accuracy. To ensure generalization and examine against overfitting, a few specific feature combinations—"triangle + red + salt & pepper", "square + green + structured", and "circle + blue + splotchy"—were explicitly excluded from the training dataset.

Figure 10: **Top-down multi-modal visual search.** a) Example input image for the multi-modal search task, composed of 16 objects arranged in a 4×4 grid. b) Each object is defined by a unique combination of shape, color, and texture. c-e) Example input images, corresponding prompts, and resulting attention maps, illustrating different scenarios of object presence. The input prompt is a concatenated one-hot vector encoding the target shape, color, and texture classes. Note: Input image colors have been inverted for better visualization.

(Minor Points) Typos and corrections:

- a. Figure 1 caption: duplicate word “see”
- b. Line 148: duplicate round bracket for figure reference & inconsistent style
- c. Line 206: specie -> species
- d. Figure 6: panels b and d are not mentioned in the main text
- e. Figure 8 caption: duplicate round brackets when referring to panel b-ii

Thank you for your time and pointing these mistakes out. We have now modified the text and figures accordingly.

Reviewer #3

This study presents a neural architecture integrating bottom-up and top-down visual processing through bidirectional recurrent gating. While the model successfully unifies multiple aspects of visual attention in a single framework - a notable advance over previous single-task approaches - several fundamental questions remain unaddressed.

We sincerely appreciate the reviewer’s positive feedback, valuable suggestions, and thorough review of the manuscript. Thank you!

Major points:

Multi-task Learning Analysis The paper demonstrates multi-task capability but lacks critical analysis of its implications:

Thank you for pointing out these very important concerns. To better respond to the comments, we added a series of experiments and analysis to the supplementary materials which we enumerate here:

Single- versus Multi-task training

To deepen our understanding of the multi-task framework, we conducted three additional sets of experiments designed to explicitly evaluate the effects of multi-task learning on performance, interference/facilitation dynamics, shared representations, and scalability. These include: (1) A comparison between multi-task and single-task models (Fig. 11); (2) Transfer learning from single-task pretraining to a multi-task setting (Figs. 13, 14); and (3) Representational Similarity Analysis to examine the emergence of shared representations (Figs. 15, 16). Together, these experiments offer a more comprehensive view of the strengths and limitations of our multi-task learning paradigm.

Multi-task versus Single-task learning

We performed single-task training for all tasks in both the MNIST and COCO families to evaluate whether the network could solve each task independently, and to assess the impact of multi-task learning on cross-task generalization. We used the same architecture across all MNIST-based tasks, and similarly across all COCO-based tasks. Training epochs were adjusted to ensure that each single-task model received a comparable number of updates to the multi-task model. Hyperparameters were kept consistent within each task family, with the exception of the symbolic orienting task, where the learning rate was reduced to 0.0001 to facilitate convergence. The results are presented in Fig. 11.

Across both datasets, the network is able to learn each task in isolation. However, for MNIST tasks involving partial supervision—indicated by dashed outlines in the figure—multi-task learning provides a clear performance advantage. For example, in the symbolic-orienting task (trained only with cross-entropy loss, without supervision for attention maps) and the top-down search task (trained only with MSE loss, without classification labels), multi-task learning improves both attention accuracy and classification performance.

The results for COCO tasks are particularly interesting. In the recognition task, the network successfully learns where to attend despite receiving no explicit supervision for attention maps. In contrast, the top-down search task fails to recover correct labels when trained in isolation, likely due to the task’s inherently ambiguous label-to-object mapping. These results suggest that multi-task training can provide helpful inductive structure—especially when supervision is partial or the task is under-constrained.

Here we would like to note that the attention maps for the single-tasks that were trained only through classification looks reasonable, despite the low attention accuracy for the given ground truth (Fig. 12).

Single-task Pretraining, Multi-task Transfer Learning

To better understand the interplay between tasks in our multi-task learning framework, we conducted a set of transfer learning experiments with two key questions in mind: (1) Does pretraining on single tasks facilitate future multi-task training? (2) To what extent does this facilitation depend on the pretraining task or the downstream tasks?

Figure 11: Multi-task versus Single-task learning. a) Results for the MNIST family. The model performs equally well on single-tasks that receive both classification and attention error signals during training (e.g. the perceptual grouping task) compared to the multi-task training. However, for the partially supervised tasks such as object recognition and top-down search, the single-task model achieves high accuracy on the supervised component but does not match the performance of the multi-task model. b) Results for the COCO task family, showing similar trends. Dotted lines indicate chance-level classification accuracy, and dashed lines indicate chance-level attention accuracy.

Figure 12: Attention maps for single-tasks trained only on classification. a) Object recognition (MNIST digit), b) Pop-out saliency (MNIST digit), c) Symbolic orienting (MNIST digit), d) Object recognition (COCO animal).

Answering these questions provides insight into factors such as computational efficiency gains or costs, task interference or facilitation, the emergence of shared representations, and how performance scales with increasing task complexity. To explore this, we first train our model (including the task-embedding layer) on a single task until learning plateaus. We then continue training the same model on all tasks simultaneously—including the pre-trained task. This procedure is repeated independently for the three COCO tasks (Fig. 13) and the seven MNIST tasks (Fig. 14).

The COCO results in Fig. 13 suggest several interesting and encouraging patterns. Models pre-trained on recognition or perceptual grouping tasks readily adapt to additional tasks with little to no degradation in performance. In contrast, a model pre-trained on the top-down search task does not show the same facilitation for recognition. Another notable observation is the speed at which pre-trained models

learn new tasks. Finally, in all cases, the network retains performance on the original pretraining task, indicating no catastrophic forgetting under this regime.

Figure 13: Multi-task learning on a pre-trained COCO model. Attention and classification (validation) accuracy during multi-task transfer learning for models pre-trained on single tasks. **left column)** Model pre-trained on the recognition task facilitates rapid learning of additional tasks, though classification accuracy initially drops. This dip may be due to the large learning rate at the peak of the warm-up schedule, as we use warm-up and cool-down learning rate scheduling. **middle column)** Model pre-trained on the perceptual grouping task enables faster and more effective multi-task transfer learning compared to the other two tasks. This may be because the model received both classification and attention supervision during single-task pretraining. **right column)** Model pre-trained on top-down visual search does not yield effective transfer learning; classification accuracy for the composite search task and attention accuracy for the recognition task remain suboptimal.

The results for MNIST paint a different picture (Fig. 14). We selected three representative cases to illustrate that not all tasks are equally facilitative. Specifically, models pre-trained on IOR or tracking either lose performance on the original task or impede learning of new tasks. Additionally, both of these pretraining tasks interfere with the performance of visual search—even in a model that was originally pre-trained on visual search itself (Fig. 14, right column). These findings highlight that certain tasks may introduce interference dynamics in the multi-task setting, depending on how their representations interact with others.

Based on the results shown in Fig. 13 and Fig. 14, we draw the following conclusions: 1) *Aligned tasks*: Some tasks facilitate each other, meaning that learning one can enhance or accelerate learning of the other. We refer to these as aligned tasks. Our results suggest that recognition and perceptual grouping fall into this category. 2) *Orthogonal tasks*: Other tasks appear to interfere with one another, such that learning one can hinder learning of the other. We refer to these as orthogonal tasks. Inhibition of return and top-down search appear to behave this way, suggesting that additional sub-networks or task-specific dynamics may be required to support them effectively. 3) *Backbone potential*: These results also point to the possibility of constructing a "backbone model" pre-trained on core tasks such as recognition and perceptual grouping. Pretraining on such foundational tasks may help improve and accelerate the learning of new, aligned tasks added later in the training process.

Figure 14: **Multi-task learning on a pre-trained MNIST model.** Attention and classification (validation) accuracy during multi-task transfer learning for models pre-trained on single tasks. **left column)** Model pre-trained on inhibition of return (IOR) task fails to accommodate multi-task transfer learning, particularly for visual search task. **middle column)** Model pre-trained on object tracking does not enable strong transfer learning, although it retains performance on the original tracking task. **right column)** Model pre-trained on top-down visual search struggles to maintain its performance when new tasks are introduced.

Representational Similarity Analysis

Continuing our investigation of single-task versus multi-task learning, a plausible concern is whether the network implicitly splits into separate, parallel single-task pathways during training. To evaluate this possibility—and to assess whether shared representations emerge across tasks—we conducted a Representational Similarity Analysis (RSA; [Kriegeskorte et al., 2008]) using the Pearson correlation coefficient as the similarity metric.

For this analysis, we trained a model based on the bidirectional recurrent gating mechanism on three distinct tasks: (1) object recognition, (2) perceptual grouping via spatial cueing, and (3) top-down visual search. We used the STL-10 dataset [Coates et al., 2011], which comprises 10 image classes with 500 training samples per class, and a fixed subset of its validation set for evaluation. Since feature representations are spatially equivariant, we opted not to use the COCO dataset (we could not create image composites for the three tasks where the target object is always at the same location and same size). STL-10, with its smaller sample size, was chosen intentionally to increase the risk of overfitting, thereby providing a more sensitive test of whether truly shared representations emerge. It also provides us with control over location and size of target objects.

The trained model achieved classification accuracies of 70%, 74%, and 75% for object recognition, perceptual grouping, and top-down search, respectively. Attention accuracy reached 94% for perceptual grouping and 86% for top-down search. Notably, the high classification accuracy on the top-down search task—which was not explicitly trained using cross-entropy loss—strongly suggests that the network leveraged feature representations learned from the other two tasks, indicating the presence of shared internal representations.

The stimuli used for RSA were constructed such that the target object was always centered (Fig. 15a),

Figure 15: Representation Similarity Analysis. a) Example input samples used for RSA, constructed with target objects consistently positioned at the center of the image. b) Averaged cross-task similarity analysis of neural activity in the pre-attentive (left) and attentive (right) phases. The results are from the penultimate layer of the model trained on the STL-10 dataset composites. c) Attentive representation similarity matrix for 200 samples from the object recognition task, revealing strong neural correlation clusters at two levels: within the same class (left) and across the same superclass (right). d) Superclass correlations observed during training. Shown are output attention maps from the top-down search task for two validation samples at epoch 8, illustrating the model’s early-stage ability to attend to relevant object categories (left: animals, right: vehicles), despite not yet achieving high overall search accuracy.

although during training these objects were randomly positioned within composites. We first analyzed both pre-attentive and attentive neural activities (i.e., feature representations) for each input at the penultimate layer of the feature pathway. We then computed cross-task representational similarities for both attention phases across all three tasks and all input samples (Fig. 16). We expected lower cross-task similarities in pre-attentive representations (Fig. 16b, lower triangle) and higher similarities in attentive representations (Fig. 16b, upper triangle). Each entry in the similarity matrices represents the Pearson correlation coefficient between the feature representations of two samples from two different tasks in either the pre-attentive or attentive phase. To better quantify these trends and highlight the difference between attention phases, we averaged the diagonal values across all task combinations and samples (Fig. 15b). The results confirm our hypothesis: pre-attentive representations are largely dissimilar across tasks (Fig. 15b, left), while attentive representations show significantly greater similarity (Fig. 15b, right). This suggests that attention consistently suppresses irrelevant features and background activity across tasks, leading to shared task-invariant representations for the same target object. These findings further support the conclusion that the network leverages its attention pathway to solve a variety of tasks by dynamically aligning its internal representations toward the relevant stimulus features.

No comparison of performance between multi-task and equivalent single-task networks.

We have now included a detailed comparison of single-task vs. multi-task training (Section: Multi-task versus Single-task Learning; Fig. 11). All tasks were trained individually using the same architecture, number of updates, and hyperparameters (except for one task requiring LR tuning). Our key findings are:

Figure 16: Representation Similarity Matrix. Results are from the penultimate layer of the model trained on the STL-10 dataset composites. Lower triangle: Cross-task similarity matrices of pre-attentive neural representations. Upper triangle: Cross-task similarity matrices of attentive neural representations. The observed increase in both auto- and cross-task correlations during the attentive phase strongly supports our hypothesis that the attention pathway is the primary mechanism for top-down neural modulation in our multi-task paradigm.

- Tasks with full supervision (e.g., classification + attention) perform similarly in both paradigms.
- Tasks with partial supervision (e.g., recognition without attention supervision, or vice versa) benefit significantly from the multi-task setup due to cross-task signal sharing.
- For both MNIST and COCO models, attention enables learning even when not directly supervised.

These results validate that multi-task learning improves performance especially when supervision is incomplete or indirect.

No evaluation of computational efficiency gains/costs

We address this concern in our transfer learning experiments (Section: Single-task pre-training, Multi-task transfer-learning; (Figs. 13, 14). By comparing learning trajectories:

- Pre-trained models on single "aligned" tasks (e.g., recognition and perceptual grouping) allow faster convergence on multi-task learning.
- Orthogonal tasks (e.g., inhibition of return, tracking) tend to slow convergence and may reduce performance on the pre-trained task.

Since we are using the same model and same training hardware (i.e., CPU, GPU, and memory), the

learning curves offer a proxy for training efficiency. We find that aligned tasks reduce the total training cost to reach a given accuracy and can significantly accelerate learning on other tasks.

No investigation of potential task interference or facilitation effects

This aspect is already addressed in our transfer learning experiments. By pre-training the model on a single task and then evaluating its performance during multi-task fine-tuning, we observed:

- **Facilitation:** Tasks like recognition and perceptual grouping improve the learning of new tasks, even without explicit supervision for them (Fig. 13).
- **Interference:** Tasks like IOR and tracking hinder new task learning or degrade pre-trained performance (Fig. 14).

These results led us to define aligned and orthogonal tasks, based on whether they facilitate or interfere with each other. This distinction provides a framework for anticipating task dynamics in multi-task settings.

No analysis of whether shared representations emerge across tasks

We conducted a Representation Similarity Analysis (RSA) on a model trained on three tasks (Section: Representation Similarity Analysis; Figs. 15, 16). The findings demonstrate that:

- Pre-attention representations differ across tasks and the same task.
- Post-attention representations show increased similarity, indicating that the attention pathway acts as a unifying mechanism across tasks.

Furthermore, performance on tasks not explicitly supervised (e.g., classification for top-down search) supports the hypothesis of internal feature sharing.

Scalability Concerns

The reported 80% classification accuracy needs context: How many classes (I guess 10)? How does this compare to state-of-the-art?

This is correct—the original tasks based on MNIST and COCO involve 10 object classes. To address this concern more directly, we have now added a new experiment using CIFAR-100, which involves 100 classes, thereby testing the model’s ability to handle increased task complexity and object variability. Additionally, our multi-modal experiment effectively spans a combinatorial space of 9 shapes \times 6 colors \times 3 textures, resulting in a large number of potential stimulus configurations, even though these are composed from relatively simple synthetic elements. We would like to emphasize that competing with machine learning state-of-the-art (SOTA) performance was not our primary objective. Rather, our focus is on understanding how biological attentional mechanisms can support flexible multi-task behavior, which are not typically addressed by SOTA classification models. Nonetheless, we agree that future benchmarking against more complex datasets like ImageNet could help clarify the model’s scope and limitations.

What are the specific challenges for scaling this approach to larger datasets or more complex tasks?

Based on our findings and design experience, we identify three key challenges and one major opportunity in scaling our approach:

Challenge 1. Data Availability: While larger datasets generally aid learning, our current multi-task paradigm requires not only target labels but also attention supervision (e.g., segmentation maps or spatial cues) for some tasks. This constrains the range of applicable datasets. However, we believe this is a temporary limitation, as new datasets that support both recognition and attention-related signals are emerging—most notably the FlyingObjects dataset [Peters et al., 2024], which is specifically designed for dynamic multi-task visual reasoning.

Challenge 2. Memory Bottlenecks from Recurrency: Although our model converges with relatively few iterations per task, the recurrent nature of the architecture introduces memory constraints that limit batch size and increase compute time. We see two promising directions to mitigate this: (1) Algorithmic

approaches such as adjoint sensitivity methods could potentially improve the memory constraints, (2) Architectural adaptation to a transformer-like model with top-down attention control, which could retain the task-routing benefits of our approach while improving scalability and parallelization.

Challenge 3. Inter-task Alignment and Completeness: As discussed earlier, certain tasks—such as Inhibition of Return (IOR) and tracking—are likely to rely on primary mechanisms outside the scope of our current model. Scaling to more complex tasks may require incorporating additional specialized modules to reflect the diversity of neural substrates involved in cognitive processing.

Opportunity: Task Design as an Implicit Regularizer: A major advantage we observed is that multi-task training with well-aligned tasks not only accelerates convergence but can also serve as an implicit form of regularization, reducing the reliance on fully supervised labels or spatial attention maps. This opens a promising avenue for designing multi-task setups to compensate for supervision gaps and support efficient generalization—especially in scenarios with limited ground-truth attention data.

How does computational cost scale with additional tasks or increased complexity?

Thank you for this important point. We found that this study is bottlenecked by memory rather than runtime constraints. Although I have to admit that our hardware included NVIDIA A100 GPUs, which are more than enough for our training. So to make an informed response, we performed the following analysis, which is also now added to the supplementary materials.

Computational Scaling with Task Count and Iterations

To empirically evaluate how computational cost scales with task count and complexity, we conducted a controlled grid of training scenarios, systematically varying: (1) the number of tasks (from 1 to 5), and (2) the number of recurrent iterations per task (from 1 to 5). For this analysis, we used our COCO model (2 million learnable parameters). All experiments were run on a single NVIDIA A100 80GB GPU. Compute times were averaged over 10 epochs, each consisting of 32 mini-batches with 128 samples per mini-batch and trained on both CE and MSE losses. To ensure consistency, all reported times were normalized relative to the training time of the feature-path through a single feedforward pass (i.e., no recurrency and no attention) of 3.97 seconds. Our key findings are as follows: (1) Adding a second task increases compute time, but additional tasks beyond two have minimal impact (Fig. 17). This jump from one to two task is primarily due to the embedding layers introduced for multi-task conditioning; and (2) Increasing the number of recurrent iterations per task (reflecting task complexity or temporal depth) results in sublinear growth in compute time. This suggests that our recurrent mechanism is computationally more efficient than a naïve unrolling strategy.

Figure 17: Computational scaling with number of task and iterations Normalized average training time per epoch per task in \log_2 scale as a function of number of tasks and number of recurrent iterations per task. While compute time for each task increases with the addition of a second task due to fixed overhead from task-embedding layers, it plateaus for three or more tasks—likely. In contrast, increasing the number of recurrent iterations leads to a sublinear growth in compute time, indicating the efficiency of the recurrent gating mechanism over naïve unrolling.

On the other hand, memory usage—both CPU and GPU—increases with longer sequence lengths due to the need to store intermediate recurrent states and gradients. It is also important to note that, in our current setup, input samples are dynamically generated during training (particularly for multi-object

COCO scenes). This introduces additional CPU overhead that is not inherent to the model architecture and could be optimized separately.

In summary, while sequence length contributes more significantly to compute cost than the number of tasks, the system demonstrates reasonable scalability even as task count and temporal complexity increase.

Minor issues

The demonstrated behavior may not truly represent IOR, which requires temporal history effects. Alternative mechanisms could produce similar task performance.

A similar point was also raised by Reviewer #2. We agree that inhibition of return (IOR) in the brain is likely an emergent phenomenon arising from mechanisms such as peripheral stimulation, oculomotor activation, or both [Klein, 2000]. As such, our explicit implementation of IOR indeed differs from the biological mechanisms that give rise to this effect. That said, we believe that incorporating such mechanisms into our model is feasible and could be achieved without major structural changes. We have included the following in our discussion:

Furthermore, our current implementation of inhibition of return (IOR) is achieved through an explicit training objective rather than emerging from temporal history effects or biological mechanisms like peripheral stimulation and oculomotor activation [Klein, 2000].

The learning mechanism should be explicitly acknowledged as biologically implausible. The claim about observing moving objects providing supervised learning signals is problematic - this represents unsupervised learning.

If the concern is regarding the use of BPTT, we have explicitly acknowledged its biological implausibility in the Discussion section. If the comment refers to our statement about motion providing a learning signal, we agree that our original wording was unclear. We have revised the text to better reflect our intent:

Naturally, our model has limitations in its ability to fully capture the breadth of neural and behavioral phenomena. For instance, it does not incorporate overt attention mechanisms such as saccades and foveation, which are essential components of human visual attention [Itti et al., 2005]. Fortunately, effective models of these mechanisms exist and could, in principle, be integrated into our framework [Mnih et al., 2014]. Additionally, our current implementation of inhibition of return (IOR) is achieved through an explicit training objective rather than biologically grounded mechanisms such as peripheral stimulation and oculomotor activation [Klein, 2000]. However, given the recurrent nature of our network, we believe it can be adapted to incorporate these dynamics. Clearly, while our model describes a range of effects, there are still many findings it cannot yet quantitatively account for. Nonetheless, we believe that the recurrent and modulatory nature of our architecture provides a strong foundation for incorporating these dynamic and mechanistic aspects in future extensions. While our model captures a wide range of attention-related behaviors and neural correlates, there remain many findings it does not yet quantitatively explain—highlighting both its current value and its potential as a foundation for further development.

Next, our model uses Backpropagation Through Time (BPTT) to update its weights. While BPTT is a powerful and widely used optimization method in machine learning, there is no direct evidence that the brain employs backpropagation—particularly not in a BPTT-like setting where information would need to propagate backward through time [Lillicrap et al., 2020]. Nevertheless, we argue that our mechanism provides a strong inductive bias enabling attentive behaviour to emerge in our network and to perform well on multiple attention tasks, even if the underlying learning rule is not biologically grounded.. However, Biologically plausible alternatives to gradient descent have been extensively studied [Bengio et al., 2015; B. A. Richards and Kording, 2023], and backpropagation itself has been shown to produce brain-like representations in visual tasks [Cadena et al., 2019; Khaligh-Razavi and Kriegeskorte, 2014; Kriegeskorte et al., 2008; Kubilius et al., 2019; Rajalingham et al., 2018]. Similarly, while weight sharing in convolutional layers lacks direct biological realism—despite being inspired by the organization of simple and complex cells in the visual cortex [Fukushima, 1988]—recent work have proposed augmentations or wake-sleep cycles to train networks resembling emergent weight sharing [Pogodin et al., 2021]. In other

words, we view backpropagation not as a biological mechanism per se, but as an effective computational tool for training our biologically inspired architecture.

We now avoid describing these cues as providing supervised signals and instead frame them as auxiliary or indirect signals that support unsupervised or self-supervised learning processes.

While the model qualitatively reproduces human-like behaviors, quantitative comparisons to human performance are absent but this claim is made. The claim is incorrect and the comparisons would be stronger with a quantitative comparison (but I understand that the authors did not do this).

We agree that our claim was overstated and appreciate your feedback. Other reviewers have raised similar concerns, and we apologize for the misrepresentation of our results. We have revised the manuscript to clarify that our findings are qualitative and do not include a quantitative comparison to human behavior.

* Fig 1. Variables are undefined (in this section) and chance level should be added to fig 1b).

Thank you for noticing. We added the information in the caption.

* Figure 2a-c: Consider adding training phase for clarity

This was very helpful in making the experiments more clear, thank you! We modified figures 2, 3, and 4 accordingly.

* Super et al. (2010) citation misrepresents the literature - Lamme & Roelfsema (2000) would better support the recurrent processing claim.

We changed the text accordingly.

* Task Implementation: Clarification needed: Is task cycling performed per stimulus? Can stimuli be randomized?

Currently we did train on single-task mini-batches given that each task could have different sequence lengths and optimization parameters. But one could use the newly developed operators in Pytorch and Tensorflow such as Jagged and Ragged tensor operators which would help handling sequences with different lengths in the same batch. We added a paragraph in the supplementary materials on this issue:

Since different tasks involve varying numbers of iterations and task-specific hyperparameters, we trained the model using mini-batches containing stimuli from a single task at a time. During training, batches were interleaved across tasks to ensure that no two consecutive batches were from the same task. This approach simplifies optimization but imposes a constraint on batching. In principle, this constraint could be lifted if all tasks shared the same number of iterations and the IOR task—due to its intermediate prediction steps—were excluded. Alternatively, recent developments in deep learning frameworks, such as *Ragged* or *Jagged* tensor operators in PyTorch and TensorFlow, now allow for efficient batching of variable-length sequences. These tools could enable more flexible multi-task batching in future implementations.

Reviewer #3 (Remarks on code availability)

well organized GitHub with clear notebooks. Most data is form other repositories and this is well referenced.

References

- Araujo, A., Norris, W., & Sim, J. (2019). Computing receptive fields of convolutional neural networks [https://distill.pub/2019/computing-receptive-fields]. *Distill*. <https://doi.org/10.23915/distill.00021>
- Ba, J. L., Kiros, J. R., & Hinton, G. E. (2016). Layer normalization. *arXiv preprint arXiv:1607.06450*.
- Baldauf, D., & Desimone, R. (2014). Neural mechanisms of object-based attention. *Science*, *344*(6182), 424–427.
- Beck, D. M., & Kastner, S. (2009). Top-down and bottom-up mechanisms in biasing competition in the human brain. *Vision research*, *49*(10), 1154–1165.
- Behrmann, M., Zemel, R. S., & Mozer, M. C. (1998). Object-based attention and occlusion: Evidence from normal participants and a computational model. *Journal of Experimental Psychology: Human Perception and Performance*, *24*(4), 1011.
- Bengio, Y., Lee, D.-H., Bornschein, J., Mesnard, T., & Lin, Z. (2015). Towards biologically plausible deep learning. *arXiv preprint arXiv:1502.04156*.
- Bichot, N. P., Heard, M. T., DeGennaro, E. M., & Desimone, R. (2015). A source for feature-based attention in the prefrontal cortex. *Neuron*, *88*(4), 832–844.
- Bouvier, S., & Treisman, A. (2010). Visual feature binding requires reentry. *Psychological science*, *21*(2), 200–204.
- Buschman, T. J., & Kastner, S. (2015). From behavior to neural dynamics: An integrated theory of attention. *Neuron*, *88*(1), 127–144.
- Cadena, S. A., Denfield, G. H., Walker, E. Y., Gatys, L. A., Tolia, A. S., Bethge, M., & Ecker, A. S. (2019). Deep convolutional models improve predictions of macaque V1 responses to natural images. *PLoS Computational Biology*, *15*(4), e1006897.
- Carrasco, M. (2011). Visual attention: The past 25 years. *Vision research*, *51*(13), 1484–1525.
- Cavanagh, P., Caplovitz, G. P., Lytchenko, T. K., Maechler, M. R., Tse, P. U., & Sheinberg, D. L. (2023). The architecture of object-based attention. *Psychonomic Bulletin & Review*, *30*(5), 1643–1667.
- Christophel, T. B., Klink, P. C., Spitzer, B., Roelfsema, P. R., & Haynes, J.-D. (2017). The distributed nature of working memory. *Trends in cognitive sciences*, *21*(2), 111–124.
- Chun, M. M., Golomb, J. D., & Turk-Browne, N. B. (2011). A taxonomy of external and internal attention. *Annual review of psychology*, *62*(1), 73–101.
- Chun, M. M., & Wolfe, J. M. (2005). Visual attention. *Blackwell handbook of sensation and perception*, 272–310.
- Coates, A., Ng, A., & Lee, H. (2011). An analysis of single-layer networks in unsupervised feature learning. *Proceedings of the fourteenth international conference on artificial intelligence and statistics*, 215–223.
- Deco, G., Pollatos, O., & Zihl, J. (2002). The time course of selective visual attention: Theory and experiments. *Vision research*, *42*(27), 2925–2945.
- Doerig, A., Sommers, R. P., Seeliger, K., Richards, B., Ismael, J., Lindsay, G. W., Kording, K. P., Konkle, T., Van Gerven, M. A., Kriegeskorte, N., et al. (2023). The neuroconnectionist research programme. *Nature Reviews Neuroscience*, *24*(7), 431–450.
- Elman, J. L. (1990). Finding structure in time. *Cognitive science*, *14*(2), 179–211.
- Feldman, J. (2013). The neural binding problem(s). *Cognitive Neurodynamics*, *7*, 1–11.
- Field, D. J., Hayes, A., & Hess, R. F. (1993). Contour integration by the human visual system: Evidence for a local “association field”. *Vision Research*, *33*(2), 173–193. [https://doi.org/10.1016/0042-6989\(93\)90156-Q](https://doi.org/10.1016/0042-6989(93)90156-Q)
- Freeman, E., Driver, J., Sagi, D., & Zhaoping, L. (2003). Top-down modulation of lateral interactions in early vision: Does attention affect integration of the whole or just perception of the parts? *Current Biology*, *13*(11), 985–989.
- Fukushima, K. (1986). A neural network model for selective attention in visual pattern recognition. *Biological Cybernetics*, *55*(1), 5–15.
- Fukushima, K. (1988). Neocognitron: A hierarchical neural network capable of visual pattern recognition. *Neural Networks*, *1*(2), 119–130.
- George, D., Lazaro-Gredilla, M., Lehrach, W., Dedieu, A., & Zhou, G. (2020). A detailed mathematical theory of thalamic and cortical microcircuits based on inference in a generative vision model. *Biorxiv*, 2020–09.
- Gilbert, C. D., & Li, W. (2013). Top-down influences on visual processing. *Nature reviews neuroscience*, *14*(5), 350–363.

- Gray, C. M., König, P., Engel, A. K., & Singer, W. (1989). Oscillatory responses in cat visual cortex exhibit inter-columnar synchronization which reflects global stimulus properties. *Nature*, *338*(6213), 334–337.
- Grossberg, S. (1987). A psychophysiological theory of reinforcement, drive, motivation, and attention. In *Advances in psychology* (pp. 3–81, Vol. 42). Elsevier.
- Grossberg, S., & Grossberg, S. (1982). Biological competition: Decision rules, pattern formation, and oscillations. *Studies of Mind and Brain: Neural Principles of Learning, Perception, Development, Cognition, and Motor Control*, 379–398.
- Han, S., Jiang, Y., Mao, L., Humphreys, G. W., & Gu, H. (2005). Attentional modulation of perceptual grouping in human visual cortex: Functional mri studies. *Human Brain Mapping*, *25*(4), 424–432.
- Heinke, D., & Humphreys, G. W. (2003). Attention, spatial representation, and visual neglect: Simulating emergent attention and spatial memory in the selective attention for identification model (saim). *Psychological review*, *110*(1), 29.
- Hochreiter, S., & Schmidhuber, J. (1997). Long short-term memory. *Neural computation*, *9*(8), 1735–1780.
- Hommel, B., Chapman, C. S., Cisek, P., Neyedli, H. F., Song, J.-H., & Welsh, T. N. (2019). No one knows what attention is. *Attention, Perception, & Psychophysics*, *81*, 2288–2303.
- Itti, L., & Koch, C. (2001). Computational modelling of visual attention. *Nature Reviews Neuroscience*, *2*(3), 194–203.
- Itti, L., Rees, G., & Tsotsos, J. K. (2005). *Neurobiology of attention*. Elsevier.
- Jeurissen, D., van Ham, A. F., Gilhuis, A., Papale, P., Roelfsema, P. R., & Self, M. W. (2024). Border-ownership tuning determines the connectivity between v4 and v1 in the macaque visual system. *Nature communications*, *15*(1), 9115.
- Jordan, M. I. (1997). Serial order: A parallel distributed processing approach. In *Advances in psychology* (pp. 471–495, Vol. 121). Elsevier.
- Kar, K., Kubilius, J., Schmidt, K., Issa, E. B., & DiCarlo, J. J. (2019). Evidence that recurrent circuits are critical to the ventral stream’s execution of core object recognition behavior. *Nature Neuroscience*, *22*(6), 974–983.
- Khaligh-Razavi, S.-M., & Kriegeskorte, N. (2014). Deep supervised, but not unsupervised, models may explain IT cortical representation. *PLoS Computational Biology*, *10*(11), e1003915.
- Kietzmann, T. C., Spoerer, C. J., Sörensen, L. K., Cichy, R. M., Hauk, O., & Kriegeskorte, N. (2019). Recurrence is required to capture the representational dynamics of the human visual system. *Proceedings of the National Academy of Sciences*, *116*(43), 21854–21863.
- Klein, R. M. (2000). Inhibition of return. *Trends in Cognitive Sciences*, *4*(4), 138–147.
- Klink, P. C., Dagnino, B., Gariel-Mathis, M.-A., & Roelfsema, P. R. (2017). Distinct feedforward and feedback effects of microstimulation in visual cortex reveal neural mechanisms of texture segregation. *Neuron*, *95*(1), 209–220.
- Knudsen, E. I. (2007). Fundamental components of attention. *Annu. Rev. Neurosci.*, *30*(1), 57–78.
- Koch, C., & Ullman, S. (1984). *Selecting one among the many: A simple network implementing shifts in selective visual attention*. (tech. rep.). MIT Cambridge Artificial Intelligence Lab.
- Kriegeskorte, N., Mur, M., & Bandettini, P. A. (2008). Representational similarity analysis-connecting the branches of systems neuroscience. *Frontiers in Systems Neuroscience*, *2*, 4.
- Krizhevsky, A., Hinton, G., et al. (2009). Learning multiple layers of features from tiny images.
- Kruijne, W., Bohte, S. M., Roelfsema, P. R., & Olivers, C. N. (2021). Flexible working memory through selective gating and attentional tagging. *Neural Computation*, *33*(1), 1–40.
- Kubilius, J., Schrimpf, M., Kar, K., Hong, H., Majaj, N. J., Rajalingham, R., Issa, E. B., Bashivan, P., Prescott-Roy, J., Schmidt, K., et al. (2019). Brain-like object recognition with high-performing shallow recurrent anns. *arXiv preprint arXiv:1909.06161*.
- Lamme, V. A. (1995). The neurophysiology of figure-ground segregation in primary visual cortex. *Journal of neuroscience*, *15*(2), 1605–1615.
- Lamme, V. A., & Roelfsema, P. R. (2000). The distinct modes of vision offered by feedforward and recurrent processing. *Trends in neurosciences*, *23*(11), 571–579.
- Lillicrap, T. P., Santoro, A., Marris, L., Akerman, C. J., & Hinton, G. (2020). Backpropagation and the brain. *Nature Reviews Neuroscience*, *21*(6), 335–346.
- Lindsay, G. W. (2020). Attention in psychology, neuroscience, and machine learning. *Frontiers in Computational Neuroscience*, *14*, 29.

- Lindsay, G. W. (2021). Convolutional neural networks as a model of the visual system: Past, present, and future. *Journal of cognitive neuroscience*, *33*(10), 2017–2031.
- Maunsell, J. H. (2015). Neuronal mechanisms of visual attention. *Annual review of vision science*, *1*(1), 373–391.
- McAdams, C. J., & Maunsell, J. H. (1999). Effects of attention on orientation-tuning functions of single neurons in macaque cortical area V4. *Journal of Neuroscience*, *19*(1), 431–441.
- Milner, P. M. (1974). A model for visual shape recognition. *Psychological review*, *81*(6), 521.
- Mnih, V., Heess, N., Graves, A., & Kavukcuoglu, K. (2014). Recurrent models of visual attention. *arXiv preprint arXiv:1406.6247*.
- Moore, T., & Zirnsak, M. (2017). Neural mechanisms of selective visual attention. *Annual Review of Psychology*, *68*, 47–72.
- Olivers, C. N., & Roelfsema, P. R. (2020). Attention for action in visual working memory. *Cortex*, *131*, 179–194.
- Olshausen, B. A., Anderson, C. H., & Van Essen, D. C. (1993). A neurobiological model of visual attention and invariant pattern recognition based on dynamic routing of information. *Journal of Neuroscience*, *13*(11), 4700–4719.
- O’Reilly, R. C., Wyatte, D., Herd, S., Mingus, B., & Jilk, D. J. (2013). Recurrent processing during object recognition. *Frontiers in Psychology*, *4*, 124.
- Paneri, S., & Gregoriou, G. G. (2017). Top-down control of visual attention by the prefrontal cortex: functional specialization and long-range interactions. *Frontiers in neuroscience*, *11*, 545.
- Peters, B., Butkus, E., Retchin, M. H., & Kriegeskorte, N. (2024). Flyingobjects: Testing and aligning humans and machines in gamified object vision tasks. *Journal of Vision*, *24*(10), 1053–1053.
- Pogodin, R., Mehta, Y., Lillicrap, T., & Latham, P. E. (2021). Towards biologically plausible convolutional networks. *Advances in Neural Information Processing Systems*, *34*, 13924–13936.
- Posner, M. I. (1980). Orienting of attention. *Quarterly journal of experimental psychology*, *32*(1), 3–25.
- Posner, M. I. (2016). Orienting of attention: Then and now. *Quarterly journal of experimental psychology*, *69*(10), 1864–1875.
- Posner, M. I. (2023). The evolution and future development of attention networks. *Journal of Intelligence*, *11*(6), 98.
- Posner, M. I., Snyder, C. R., & Davidson, B. J. (1980). Attention and the detection of signals. *Journal of experimental psychology: General*, *109*(2), 160.
- Rajalingham, R., Issa, E. B., Bashivan, P., Kar, K., Schmidt, K., & DiCarlo, J. J. (2018). Large-scale, high-resolution comparison of the core visual object recognition behavior of humans, monkeys, and state-of-the-art deep artificial neural networks. *Journal of Neuroscience*, *38*(33), 7255–7269.
- Reynolds, J. H., & Heeger, D. J. (2009). The normalization model of attention. *Neuron*, *61*(2), 168–185.
- Richards, B. A., Lillicrap, T. P., Beaudoin, P., Bengio, Y., Bogacz, R., Christensen, A., Clopath, C., Costa, R. P., de Berker, A., Ganguli, S., et al. (2019). A deep learning framework for neuroscience. *Nature neuroscience*, *22*(11), 1761–1770.
- Richards, B. A., & Kording, K. P. (2023). The study of plasticity has always been about gradients. *The Journal of Physiology*, *601*(15), 3141–3149.
- Robertson, L. C. (2003). Binding, spatial attention and perceptual awareness. *Nature Reviews Neuroscience*, *4*(2), 93–102.
- Roelfsema, P. R. (2023). Solving the binding problem: Assemblies form when neurons enhance their firing rate—they don’t need to oscillate or synchronize. *Neuron*, *111*(7), 1003–1019.
- Roelfsema, P. R., Lamme, V. A., & Spekreijse, H. (1998). Object-based attention in the primary visual cortex of the macaque monkey. *Nature*, *395*(6700), 376–381.
- Ronneberger, O., Fischer, P., & Brox, T. (2015). U-net: Convolutional networks for biomedical image segmentation. *International Conference on Medical Image Computing and Computer-assisted Intervention*, 234–241.
- Rutishauser, U., Walther, D., Koch, C., & Perona, P. (2004). Is bottom-up attention useful for object recognition? *Proceedings of the 2004 IEEE Computer Society Conference on Computer Vision and Pattern Recognition, 2004. CVPR 2004.*, *2*, II–II.
- Schiatti, L., Gori, M., Schrimpf, M., Cappagli, G., Morelli, F., Signorini, S., Katz, B., & Barbu, A. (2023). Modeling visual impairments with artificial neural networks: A review. *Proceedings of the IEEE/CVF International Conference on Computer Vision*, 1987–1999.
- Scholl, B. J. (2001). Objects and attention: The state of the art. *Cognition*, *80*(1-2), 1–46.

- Schrimpf, M., Kubilius, J., Hong, H., Majaj, N. J., Rajalingham, R., Issa, E. B., Kar, K., Bashivan, P., Prescott-Roy, J., Geiger, F., et al. (2018). Brain-score: Which artificial neural network for object recognition is most brain-like? *BioRxiv*, 407007.
- Self, M. W., Jeurissen, D., van Ham, A. F., van Vugt, B., Poort, J., & Roelfsema, P. R. (2019). The segmentation of proto-objects in the monkey primary visual cortex. *Current Biology*, 29(6), 1019–1029.
- Self, M. W., Mookhoek, A., Tjalma, N., & Roelfsema, P. R. (2015). Contextual effects on perceived contrast: Figure-ground assignment and orientation contrast. *Journal of Vision*, 15(2), 2–2.
- Shadlen, M. N., & Movshon, J. A. (1999). Synchrony unbound: A critical evaluation of the temporal binding hypothesis. *Neuron*, 24(1), 67–77.
- Shipp, S. (2004). The brain circuitry of attention. *Trends in cognitive sciences*, 8(5), 223–230.
- Singer, W. (2001). Consciousness and the binding problem. *Annals of the New York Academy of Sciences*, 929(1), 123–146.
- Srivastava, S., Wang, W. Y., & Eckstein, M. P. (2024). Emergent human-like covert attention in feed-forward convolutional neural networks. *Current Biology*, 34(3), 579–593.
- Styles, E. (2006). *The psychology of attention*. Psychology Press.
- Super, H., Romeo, A., & Keil, M. (2010). Feed-forward segmentation of figure-ground and assignment of border-ownership. *PLoS one*, 5(5), e10705.
- Thiele, A., & Bellgrove, M. A. (2018). Neuromodulation of attention. *Neuron*, 97(4), 769–785.
- Thorat, S., Aldegheri, G., & Kietzmann, T. C. (2021). Category-orthogonal object features guide information processing in recurrent neural networks trained for object categorization. *arXiv preprint arXiv:2111.07898*.
- Treisman, A. (1996). The binding problem. *Current Opinion in Neurobiology*, 6(2), 171–178. [https://doi.org/https://doi.org/10.1016/S0959-4388\(96\)80070-5](https://doi.org/https://doi.org/10.1016/S0959-4388(96)80070-5)
- Treisman, A. (1998). Feature binding, attention and object perception. *Philosophical Transactions of the Royal Society of London. Series B: Biological Sciences*, 353(1373), 1295–1306.
- Treisman, A., & Gelade, G. A. (1980). A feature-integration theory of attention. *Cognitive Psychology*, 12, 97–136. <https://api.semanticscholar.org/CorpusID:353246>
- Tsotsos, J. K., & Bruce, N. D. B. (2008). Computational foundations for attentive processes [revision #91149]. *Scholarpedia*, 3(12), 6545. <https://doi.org/10.4249/scholarpedia.6545>
- Tsotsos, J. K., & Rothenstein, A. (2011). Computational models of visual attention [revision #171311]. *Scholarpedia*, 6(1), 6201. <https://doi.org/10.4249/scholarpedia.6201>
- Tsotsos, J. K. (2021). *A computational perspective on visual attention*. MIT Press.
- Tsotsos, J. K., Culhane, S. M., Wai, W. Y. K., Lai, Y., Davis, N., & Nuflo, F. (1995). Modeling visual attention via selective tuning. *Artificial Intelligence*, 78(1-2), 507–545.
- Tsotsos, J. K., Itti, L., & Rees, G. (2005). A brief and selective history of attention. *Neurobiology of attention*, 50003–3.
- van Bergen, R. S., & Kriegeskorte, N. (2020). Going in circles is the way forward: The role of recurrence in visual inference. *Current Opinion in Neurobiology*, 65, 176–193.
- Von Der Malsburg, C. (1994). The correlation theory of brain function. In *Models of neural networks: Temporal aspects of coding and information processing in biological systems* (pp. 95–119). Springer.
- Von der Malsburg, C. (1995). Binding in models of perception and brain function. *Current opinion in neurobiology*, 5(4), 520–526.
- Von der Malsburg, C. (1999). The what and why of binding: The modeler’s perspective. *Neuron*, 24(1), 95–104.
- von der Malsburg, C. (2024). How are segmentation and binding computed and represented in the brain? *Cognitive Processing*, 1–6.
- Walther, D., Rutishauser, U., Koch, C., & Perona, P. (2005). Selective visual attention enables learning and recognition of multiple objects in cluttered scenes. *Computer Vision and Image Understanding*, 100(1-2), 41–63.
- Ward, L. M. (2008). Attention [revision #185343]. *Scholarpedia*, 3(10), 1538. <https://doi.org/10.4249/scholarpedia.1538>
- Wolfe, J. M., & Horowitz, T. S. (2004). What attributes guide the deployment of visual attention and how do they do it? *Nature reviews neuroscience*, 5(6), 495–501.
- Wolfe, J. M., & Horowitz, T. S. (2017). Five factors that guide attention in visual search. *Nature Human Behaviour*, 1(3), 0058.
- Yamins, D. L., & DiCarlo, J. J. (2016). Using goal-driven deep learning models to understand sensory cortex. *Nature neuroscience*, 19(3), 356–365.

Zipser, K., Lamme, V. A., & Schiller, P. H. (1996). Contextual modulation in primary visual cortex. *Journal of Neuroscience*, *16*(22), 7376–7389.

Letter

Dear Editor,

We are pleased to re-submit our revised manuscript, "Modeling Attention and Binding in the Brain through Bidirectional Recurrent Gating", for your consideration. We would like to express our sincere gratitude to you and to all three reviewers for their time and their thoughtful, constructive, and detailed feedback provided on our work.

We have performed an extensive revision to address all the concerns raised during the review process. We believe these changes have substantially strengthened the paper, clarified its contributions, and also better contextualized its relevance for the neuroscience community.

A primary focus of our revision was to strengthen the connection between our computational model and theoretical and experimental neuroscience, a key point raised by the reviewers. To this end, we have rewritten the Introduction and Discussion sections to move beyond demonstrating behaviors to offering mechanistic explanations and testable hypotheses. The new discussion now explicitly frames our work as a contribution to major theoretical debates, including the unification of attentional phenomena, the neural basis of feature binding, and the circuit-level implementation of active biased competition. We have also added a dedicated section outlining specific, falsifiable predictions that could directly guide future experimental work.

Furthermore, in response to the second reviewer's valuable feedback, we have added a significant new set of results. This new section details a series of psychophysical experiments showing that our model qualitatively reproduces canonical human behavioral findings, including the effects of perceptual load, the phenomenon of inattention blindness, and the attentional modulation of perceived contrast. We believe these new results provide further evidence for the model's cognitive plausibility and significantly enhance the relevance of our work.

We are very grateful for the opportunity to improve our manuscript. The reviewers' insightful feedback has been instrumental in helping us refine our arguments and better articulate the significance of our findings. We are confident that the manuscript is now substantially improved and makes a stronger contribution to the field.

Best regards,

Saeed Salehi, Ari S. Benjamin, Prof. Klaus-Robert Müller, and Prof. Konrad P. Kording

Reviews

We would like to cordially thank the reviewers for the constructive and insightful comments on our paper. We feel that they have greatly helped us to improve the manuscript. Our responses are in blue-colored text. The revised texts from the manuscript are in gray-colored text.

Reviewer #1

Salehi and co-authors aim at modelling attention in the brain by means of a deep neural architecture using bidirectional neural gating in the model. The training relies on backpropagation through time.

The clarity of the manuscript has improved significantly during the revision. In particular, the teaching and input signals used in the individual studies are now more comprehensible. Overall, the proposed model represents a substantial improvement over more traditional deep learning architectures, which often neglect attentional phenomena. However, I remain unconvinced that the current approach offers substantial insight into the mechanisms of attention in the brain.

We thank the reviewer for their time and for their thoughtful and detailed feedback on our revised manuscript. We appreciate their positive comments regarding the improved clarity. We also hope that this new round of revisions demonstrates a similarly significant improvement in response to the reviewer's valuable feedback.

While some of my earlier concerns have been addressed, others remain unresolved. My remaining concerns are as follows:

1) I criticized that in the introduction, models and data are combined in a very unreflected way. I still spotted such sentences in the revised manuscript which are:

“Empirical research reveals that selectively focusing on portions of the visual scene produces neural responses that are more localized, sparser, and less noisy [Posner et al., 1980; Tsotsos and Bruce, 2008].” These references do not support the claim. Posner et al. (1980) is a behavioral study, and Tsotsos and Bruce (2008) is a computational modeling paper. Neither analyzed neural responses.

„In the context of learning, behavioral evidence shows that focusing on task-relevant stimuli improves generalization [Rutishauser et al., 2004; Walther et al., 2005].“ These references are primarily computational. They do not provide behavioral evidence.

We sincerely thank the reviewer for their meticulous reading and for pointing out these specific citation misalignment. We have corrected these specific sentences. We have also made an extensive overhaul of the introduction and discussion section to ensure the claims are now accurately supported by their corresponding references.

2) While the tone throughout the manuscript is mostly balanced, I find the following statement in the abstract to be an overstatement and recommend removing it: “Most importantly, our proposed model unifies decades of cognitive and neurophysiological findings of visual attention into a single principled architecture.” The model is a valuable and interesting contribution to computational modelling of attention. However, it cannot reasonably be described as a unification of decades of empirical findings. Such a claim overreaches the evidence presented and may raise unrealistic expectations.

We thank the reviewer for this important feedback. We agree that the original wording was an overstatement and that we failed to articulate our intended message which has led to the misunderstanding, and we have removed the sentence from the abstract.

Our goal was not to claim a complete theory of all attentional phenomena, but to argue that a core contribution of our work is providing a unifying computational account for several key modes of attention that are often studied and modeled in isolation. As we now elaborate in the second paragraph of the revised Discussion, our work addresses the central challenge of unifying diverse attentional phenomena (spatial, feature-based, and object-based) within a single framework. Our contribution is to propose and demonstrate that a single core mechanism, bidirectional recurrent gating, implemented in one architecture, can successfully perform the three canonical attention behaviors of orienting, filtering,

and search. This result provides computational support for the hypothesis that these different modes of attention may not rely on fundamentally separate mechanisms, but can emerge from the same set of core computations operating over different neural representations.

We have rewritten the abstract and the discussion to state this more precisely. We hope the reviewer finds this revised framing to be a more accurate and appropriately scoped representation of our work.

3) A key concern remains that the model is used primarily to demonstrate attentional behaviors, rather than to explain them. The manuscript does not articulate how the model sheds light on the function or purpose of attention in the brain. It also does not appear to address any major open research questions in attention neuroscience.

We thank the reviewer for this crucial feedback, which gets to the heart of what a computational model should offer to neuroscience. We agree that the previous version of our manuscript focused more on demonstrating attentional phenomena than on explaining their underlying mechanisms or addressing key open questions. Addressing this gap was the primary goal of our revision. For example, we have rewritten the Discussion section to frame our work as an explanatory and hypothesis-driven contribution. To this end, the new Discussion (summarized below) now explicitly addresses several major open questions in the field:

The Unification Problem: How can diverse attentional phenomena (spatial, feature, object) be unified by a common set of principles? We address this by proposing bidirectional recurrent gating as a candidate mechanism. Our model demonstrates that a single architecture can perform all three canonical attention tasks, supporting the hypothesis that they emerge from shared computations rather than distinct modules.

The Binding Problem: How does the brain bind features into coherent objects? Our model provides a working implementation of the "binding by firing rate enhancement" theory. It offers a computational explanation for how top-down attentional modulation can selectively amplify an object's features to bind them together and segregate them from the background.

The Circuit-Level Implementation of Attention: How do top-down signals work to implement active biased competition? We propose a specific, testable computational flow to explain this. In our model, top-down signals induce a multiplicative response gain in feature-selective neurons. This amplified signal then interacts with local divisive normalization circuits to ensure that attended information wins the competition for neural representation. This provides a concrete mechanistic explanation for our model of attention at the circuit level.

By explicitly framing our model as a proposed solution to these open questions, we hope the revised manuscript now offers substantial insight into the mechanisms of attention. We kindly direct the reviewer to the last two paragraphs of introduction and the first few paragraphs of the new Discussion, which are dedicated to articulating these explanatory points.

In their response letter, the authors state their research question as: "How can a biologically inspired system successfully operate on real-world stimuli while exhibiting core attentional behaviors observed in humans?" While this is indeed a legitimate research question, it is one focused on building a model, rather than one that can guide or be guided by experimental neuroscience. In its current form, the model does not clearly serve as a tool for hypothesis generation or for interpreting neural data. For example, while the model shows strong performance on grouping and segmentation tasks, this is likely a result of supervised training on pre-segmented data, rather than an emergent property rooted in biologically plausible mechanisms.

We thank the reviewer for raising this critical point about the model's utility as a tool for neuroscience. We agree that a model's value is measured by its ability to generate testable hypotheses and interpret data, not just to replicate behaviors. A major focus of our revision was to explicitly articulate how our model presents testable predictions and serves to provide evidence for the emergent attention hypothesis. To address the concern that the model is not a tool for hypothesis generation, we have added a new section to the Discussion dedicated entirely to its specific, falsifiable predictions. These include:

1- Task Invariance in Early Visual Pathways: The model predicts that while high-level areas show task-

specific signals, neurons in early- to mid-level sensory pathways should exhibit relatively stable tuning properties across different attentional tasks.

2- Experience-Dependent Learning: The model predicts that an organism’s learning history will shape its attentional representations. We hypothesize that animals trained on tasks that do not require object integration will show impoverished neural signatures for binding when later tested on more complex tasks.

3- Modeling Causal Interventions: The model’s modular architecture allows it to be used *in silico* to predict the network-level and behavioral consequences of specific perturbations, such as simulating the effects of pharmacological blockers by altering corresponding modulatory interactions.

Regarding the specific concern that strong performance on segmentation is solely a result of supervised training on pre-segmented data, we would like to clarify two points:

First, while some tasks use segmentation maps for training, a significant number of our key results emerge without this level of supervision. For example, in the object recognition with noise, pop-out saliency, symbolic orienting, multi-modal shape recognition, and feature-masking (CelebA) experiments, the model learns to produce the correct attention map purely from the classification objective and the multitask paradigm.

Second, we have addressed the limitation of using supervised signals in a dedicated paragraph in the Discussion. We agree with the reviewer that biological organisms learn without explicit segmentation maps. We discuss how the supervision in our model acts as a proxy for richer, more plausible learning signals, such as spatiotemporal consistency from object motion or depth information from stereo-vision, and we argue that our recurrent architecture is well-suited to incorporate these self-supervised signals in future work.

The discussion would benefit greatly from addressing how the model could inform future experiments or contribute to theoretical debates about attention in the brain.

We thank the reviewer for this constructive advice. This recommendation became the guiding principle for our revision. The Discussion section has been rewritten to explicitly address how our model informs future experiments and contributes to theoretical debates. We believe that the new Discussion section now makes a much stronger case for our model’s value as a tool for both theoretical and experimental neuroscience, and we hope the reviewer finds that it directly addresses their concerns.

In conclusion, the manuscript shows meaningful progress in the domain of modelling attention in deep neural networks. However, it still falls short in offering novel insights into the biological or cognitive mechanisms of attention. Strengthening the connection to neuroscience—either through more precise referencing or by framing the model in a way that supports experimental interpretation—would significantly enhance the impact of the work.

We sincerely thank the reviewer for their constructive feedback. We believe the revised version makes a substantially stronger case for our model’s ability to offer novel insights into the biological and cognitive mechanisms of attention. We hope they will agree that these extensive revisions have successfully strengthened the paper’s connection to neuroscience and better articulate its contribution to understanding the mechanisms of attention.

Reviewer #2

Salehi et al. present a novel model and introduce innovative methods for incorporating established attentional findings into a machine learning framework. The model exhibits many expected patterns of behavior, although it is not strongly motivated by brain architecture or evolutionary principles.

The authors have addressed many of my concerns through additional experiments and by providing more detailed descriptions of their methodology and I thank them for the careful response. The revised figures enhance clarity, and I appreciate their comprehensive outline of the training procedure for the IoR task.

These are a few additional suggestions, major and minor.

We thank the reviewer for their continued engagement with our work and for their positive and encouraging feedback. We also thank them for their insightful comment regarding the model's motivation from brain architecture. This is a crucial point, and we have sought to further strengthen this connection in the current revisions, particularly in the Discussion. We will now address their additional suggestions in detail.

Major points

1. It would help to target the model to address an important open question in the attention literature, to make systematic predictions about mechanisms of unexplained behavioral or neural phenomena or to provide an integrative or synthetic view of these diverse phenomena. Morgan et al (2025) provides a recent example of such a model: <https://www.biorxiv.org/content/10.1101/2024.11.09.622721v1>

We thank the reviewer for this excellent and constructive suggestion, and for pointing us to the work of Morgan et al. (2025). We agree that a key measure of a model's contribution is its ability to address open questions and provide a synthetic view of the field. This has been the central focus of our revision. To this end, and as detailed in our response to Reviewer 1, we have rewritten the Introduction and Discussion sections to better frame our work in this context. The discussion now explicitly addresses how our model provides an integrative view on several open questions, including:

- How diverse attentional phenomena (spatial, feature, object-based) can be *unified* by a single underlying mechanism.
- How *binding* can emerge from attentional modulation via "binding by firing rate enhancement".
- How *active biased competition* is implemented at the circuit-level through the interaction of multiplicative gain and divisive normalization.

Furthermore, inspired by the reviewer's suggestion and the provided reference, we took the opportunity to further strengthen the connection between our model and human behavioral phenomena. We have therefore conducted a new set of experiments to test if our model could replicate canonical psychophysical findings from the human attention literature. These new results, presented in a new subsection of the Results section ("Psychophysical results"), demonstrate that our model qualitatively reproduces:

1. Attention's role in enhancing contrast sensitivity (contrast gain).
2. The increase in perceived contrast for attended stimuli.
3. The effects of perceptual load and a clear demonstration of inattention blindness.

Psychophysical results

Psychophysical studies provide a powerful framework for characterizing attentional mechanisms and their influence on perception [Carrasco, 2011], making them critical for evaluating the biological relevance of computational models [Morgan et al., 2025]. Drawing on this empirical foundation, we designed three experiments (contrast-detection, contrast discrimination, and orientation change-detection) to test whether our model's behavior aligns with established human findings. For all experiments, we employed a spatial cueing paradigm. The model was presented with a 3x3 grid containing Gabor patches as target and distractor stimuli, and Gaussian patches as a spatial cues to orient attention. To mimic peripheral dynamics, targets appeared only in the eight off-center locations (Fig. 1a). While the same base architecture was used for all experiments, each model was trained independently on its specific task. For the analysis, we used Sigmoid function to fit the data and 50% threshold for reporting. Detailed

specifications of the training procedures and stimulus parameters are available in the supplementary materials.

Attention enhances contrast sensitivity

A key finding in psychophysics is that transient covert attention can enhance performance in visual discrimination tasks [Cameron et al., 2002]. This enhancement manifests as a decrease in the contrast threshold required for perception, resulting in a leftward shift of the psychometric function (Fig. 1b). This effect is considered a signature of the "contrast gain" mechanism of attention [Reynolds et al., 2000]. To test whether our model exhibits this behavior, we designed a cued localization task. In each trial, a transient cue oriented the network's attention to a random location. Subsequently, a Gabor patch with variable contrast appeared in 50% of trials at one of the peripheral locations, independent of the cue's position. The model was trained to output the location of the Gabor patch if present, or signal its absence. We then constructed psychometric functions by measuring performance as a function of target contrast, comparing trials where the target appeared at the cued (i.e., valid) location versus at an uncued (i.e., invalid) location. Our results replicate the established psychophysical findings. As shown in Figure 1d, attention markedly improved the model's performance. Critically, the psychometric function for the attended condition ($th_{50\%} = 0.21$) is shifted to the left compared to the unattended condition ($th_{50\%} = 0.25$). This demonstrates a clear reduction in the contrast threshold, consistent with the contrast gain mechanism reported in humans [Cameron et al., 2002].

Attention enhances perceived contrast

To test if our model accounts for attention's influence on perception, we sought to replicate the seminal work of Carrasco, Ling, and Read [Carrasco et al., 2004]. Their study demonstrated that transient covert attention increases the perceived contrast of a stimulus, a phenomenon quantified by measuring the Point of Subjective Equality (PSE). We implemented a cued, contrast comparison task. On each trial, a spatial cue was presented, followed by two Gabor patches of varying contrast at different locations (Fig. 1e). The model's objective was to identify which of the two patches had a higher contrast. To measure the PSE during evaluation, we designated one Gabor as the "standard" (with a fixed contrast of 0.1 a.u.) and the other as the "test" (with variable contrast). We then assessed the probability that the model would select the test patch as higher-contrast, both when the test patch itself was cued and when the standard patch was cued. The model's behavior qualitatively replicated the key human finding. As shown in Figure 1f, attending to the test patch systematically increased the likelihood of it being reported as having higher contrast (standard-cued: $th_{50\%} = 0.11$, neutral-cued: $th_{50\%} = 0.10$, and test-cued: $th_{50\%} = 0.09$). Meaning that attention increases perceived contrast for the attended stimulus, even when it is objectively less contrastive than the unattended stimulus. This outcome, which demonstrates that attention enhances apparent contrast in our model, is consistent with the original findings of Carrasco et al. (compare with Fig. 1g).

Perceptual load and inattention blindness

Two foundational concepts in attention research are perceptual load [Lavie, 1995] and inattention blindness [Simons, 2007; Simons, 2000]. Perceptual Load Theory posits that the ability to process task-irrelevant information depends on the demands of the primary task; high load consumes attentional resources, leaving little capacity for processing all the relevant stimuli [Lavie, 1995]. A direct consequence of this is inattention blindness: the striking failure to notice a salient, unexpected object when attention is engaged by a demanding task [Cartwright-Finch and Lavie, 2007; Macdonald and Lavie, 2008]. To determine if our model's behavior is consistent with these phenomena, we designed a cued orientation change-detection task. In the task, a spatial cue was followed by the presentation of one to eight Gabor patches (Fig. 1h). On 50% of trials, one of these patches would change its orientation, and the model's objective was to detect and output this change. During training, the cue's location was correlated with the target patch, indicating the target's location with 50% validity to encourage its use. Our evaluation yielded two key results. First, we found a clear effect of perceptual load: as the number of distractor patches increased, the model's ability to detect the orientation change systematically decreased, as illustrated by the rightward shift in the psychometric functions (Fig. 1i) (50% threshold as a function of number of distractors; 0: $th_{50\%} = 7.9^\circ$, 1: $th_{50\%} = 8.8^\circ$, 3: $th_{50\%} = 9.5^\circ$, and 7: $th_{50\%} = 10.0^\circ$). Second, to test for inattention blindness, we analyzed performance based on cue validity. When the cue correctly located the target patch (valid cue), performance was highest. However, when the cue

Figure 1: Psychophysical results a) Gabor patches (targets/distractors) can only appear in the eight off-center locations. b) Schematic illustration of the contrast gain mechanism, where attention shifts the psychometric function to the left, lowering the contrast threshold. c) Trial sequence for the cued localization task. A cue precedes the stimuli, appearing at either the target's location (valid cue) or a different location (invalid cue). d) Model results showing contrast gain. The psychometric function for the attended (validly cued) condition is shifted to the left compared to the unattended condition, indicating a reduced contrast threshold. e) Trial sequence for contrast discrimination. The cue either precedes the 'test' patch, the 'standard' patch, or be neutral (i.e., center). f) The model's results qualitatively match the human data, showing a similar Point of Subjective Equality (PSE) shift and demonstrating that attention increases the apparent contrast of the cued stimulus. The PSE is shown by horizontal gray line intersecting the three curves at 50%. g) Original human data showing how attention alters perceived contrast. Figure is reproduced from Carrasco et al. [Carrasco et al., 2004] for Gabor patches with 2 cpd and standard patch contrast of 6%. h) Trial sequence for the orientation change-detection task. The number of distractor patches varies from 0 to 7 to control the perceptual load. i) The effect of perceptual load on performance. For neutrally cued trials, change-detection accuracy decreases as the number of Gabor patches (i.e., number of distractors) increases. j) Inattentional blindness effect shown for a set size of two patches. Performance is highest with a valid cue and significantly impaired when an invalid cue directs attention to a distractor, falling well below the neutral cue baseline.

directed attention to a distractor (invalid cue), performance dropped dramatically, falling well below the neutral-cued condition (Fig. 1j) (valid-cued: $th_{50\%} = 8.2^\circ$, neutral-cued: $th_{50\%} = 8.8^\circ$, and invalid-cued: $th_{50\%} = 10.1^\circ$). This profound impairment in detecting a visible change when attention is misdirected is a clear analogue of inattention blindness.

We are very grateful for the reviewer’s suggestion, which directly motivated this new section. We believe these new psychophysical results significantly strengthen the manuscript by showing that our proposed mechanisms not only perform complex tasks but also operate under constraints that are remarkably similar to those of the human visual system, thereby providing a more cohesive bridge to experimental work.

2. In symbolic reorienting task (MNIST), can the same digit appear more than once in an image during training? Please clarify.

Thank you for this clarifying question. In the training protocol for the symbolic reorienting task reported in the manuscript, the distractor digits were sampled from a set that explicitly excluded the target digit’s class. Therefore, the same digit class did not appear as both a target and a distractor in the same training trial. However, to test the robustness of our model, we also ran experiments where distractors were sampled with replacement (i.e., allowing the target digit to potentially appear as a distractor). We found no significant difference in the model’s performance under this condition.

3. In the top-down search task (MNIST), was it necessary to explicitly provide the target attention map? Given that the model is trained across multiple tasks, shouldn’t it have developed an internal representation of what the digit ‘8’ looks like and thus be capable of generating the attention map autonomously?

We thank the reviewer for this intriguing question. The reviewer is correct that the model develops internal representations of digits from other tasks. However, the top-down visual search task has a unique design. The identity of the target digit is provided as an input prompt. If we were to use a classification loss for this task, the model would achieve 100% accuracy by learning a trivial "Clever Hans/shortcut" solution: simply outputting the label from the prompt. This would prevent the model from learning the cognitive process we aimed to model. Therefore, to ensure the model learns to actually perform the search, the only training objective used for this specific task is the loss on the target attention map. Without the attention map as a target, there would be no learning signal to guide the model for this task.

Furthermore, the model needs to be trained simultaneously on the "top-down search" task prompt itself. The model must learn to associate the search-task ID with the specific behavior of locating a prompted object in the visual input.

That being said, our results strongly support the reviewer’s intuition that the model leverages its existing "knowledge" (i.e., learned representations). As we demonstrate in the Supplementary Materials, if the model is pretrained on object recognition alone, it can subsequently learn the top-down visual search task in very few training epochs. This suggests that the necessary object representations are already in place, and the supervision on the attention map primarily teaches the model how to access and apply this knowledge in response to a new task demand.

Minor points

1. Space after period: Lines 123 and 764
2. Figure 3a caption: the word ‘trained’ appears to be mistakenly included or misplaced
3. Bracket after panel number in captions of Figure 4, Figure 7 and Figure 26
4. Figure 6 caption: c-d -> c-e, bold f
5. Figure 8c) “purelyd” -> “purely”
6. Figure 10c) did the authors mean “F-neurons”
7. Line 379: B capitalized
8. Line 760: The word “Table” missing from the reference
9. Formatting: inconsistent casing across section titles and across figure titles
10. Figure 14: “;” placed incorrectly before ‘width’, fist -> first
11. Lines 845-846: better to refactor

12. Line 989: spacing after open bracket
13. Figure 24: caption for panel e is missing
14. Figure 25: disctractor -> distractor
15. Details of panels c-e could be added in Figure 29

We are very grateful to the reviewer for their time and attention to detail and for providing these helpful corrections. We have carefully gone through the manuscript and have corrected all of the noted typos and errors in the text and figures. Regarding point 11 (Lines 845-846: better to refactor), we were not sure what exactly was requested but we did remove the second part of the sentence which was ambiguous.

Reviewer #3

I had some major points that the authors addressed throughly

multi-task learning analysis -> added substantial experiments including single vs. multi-task training comparisons, transfer learning studies, and representational similarity analysis).

efficiency -> scaling analyses showing how their approach scales with task count and complexity.

scaling -> conducted experiments with CIFAR-100

and more (task interference, shared representations etc).

So in sum they have taken my comments and added substantial changes and updates.

We sincerely thank the reviewer for their positive assessment of our work and revision. We appreciate their continued engagement and will now address their remaining points.

The authors also show that certain tasks can be aligned or orthogonal to each other. I would be interested in reading in the discussion speculations on the source of this. Would it be possible to predict for two tasks if they are one or the other?

We thank the reviewer for this thought-provoking question. They are correct that understanding the principles that govern task alignment and interference in our model is a critical next step. While a systematic investigation is beyond the scope of this work, we agree that it is an important topic for discussion. As suggested, we have now added a paragraph discussing and speculating on the sources of these effects:

- **Competition for Specialized Resources:** We hypothesize that task interference depends on the degree to which tasks compete for the same limited computational resources. For example, tasks requiring dynamic, state-dependent processing, such as Inhibition of Return (IOR) and object tracking, appear to compete for the finite resources of the model's recurrent working memory systems. The model's current architecture lacks certain biological systems (e.g., distributed working memory, a dedicated dorsal stream), forcing tasks that would rely on these systems to compete for the available architectural resources.
- **A second source of interference may be inherent to the tasks themselves,** specifically the potential for conflict between bottom-up (stimulus-driven) and top-down (goal-driven) signals. A task where the goal aligns with stimulus saliency (e.g., "attend to the pop-out item") is likely aligned with the network's default processing. Conversely, a task requiring a search for a non-salient item creates competition between the bottom-up drive to attend to a salient distractor and the top-down search goal.
- **Predicting Task Relationships:** Regarding the reviewer's question about *predicting* this relationship, we believe this is possible in principle. A predictive framework would likely require analyzing each task's demands along the axes described above: (1) the specific computational resources required (e.g., feedforward feature extraction vs. recurrent state maintenance) and (2) the degree of conflict between top-down goals and bottom-up sensory evidence. We speculate that tasks occupying similar points in this abstract "task space" would be more likely to interfere with one another.

Beyond that I have no additional comments and enjoyed reading the paper very much. I think it is a very relevant step in the field.

We thank the reviewer for their encouraging and positive assessment and delighted to hear that they enjoyed reading the paper.

Two final remark, I can imagine the authors are thinking about ablation studies in the future?

We are planning a series of ablation and extension experiments to precisely quantify the contribution of each architectural component—such as divisive normalization and distributed working memory—to the model's performance across different tasks. We have also made our code publicly available on GitHub, hoping this would encourages further investigation and collaborative exploration of the model's properties.

Also, is the repository up to date with this version of the submission? Looking at it, it might be of the last version?

We thank the reviewer for catching this. They are correct that the repository was not yet synced with the latest revisions, and we apologize. We will push the final, updated code to the public repository.

Best, H.Steven Scholte

Reviewer #3 (Remarks on code availability):

The code has a readme file and clear instruction on how to run, and explanation of the repository structure and a tutorial on how to run everything. It also has a demo.

I suspect (but have not run it) that the repository is of the first submission and not updated yet.

We thank the reviewer for catching this. They are correct that the repository was not yet synced with the latest revisions, and we apologize. We will push the final, updated code to the public repository.

References

- Cameron, E. L., Tai, J. C., & Carrasco, M. (2002). Covert attention affects the psychometric function of contrast sensitivity. *Vision research*, *42*(8), 949–967.
- Carrasco, M. (2011). Visual attention: The past 25 years. *Vision research*, *51*(13), 1484–1525.
- Carrasco, M., Ling, S., & Read, S. (2004). Attention alters appearance. *Nature neuroscience*, *7*(3), 308–313.
- Cartwright-Finch, U., & Lavie, N. (2007). The role of perceptual load in inattention blindness. *Cognition*, *102*(3), 321–340.
- Lavie, N. (1995). Perceptual load as a necessary condition for selective attention. *Journal of Experimental Psychology: Human perception and performance*, *21*(3), 451.
- Macdonald, J. S., & Lavie, N. (2008). Load induced blindness. *Journal of Experimental Psychology: Human Perception and Performance*, *34*(5), 1078.
- Morgan, J., Albanna, B., & Herman, J. P. (2025). A recurrent vision transformer shows signatures of primate visual attention. *arXiv preprint arXiv:2502.10955*.
- Reynolds, J. H., Pasternak, T., & Desimone, R. (2000). Attention increases sensitivity of v4 neurons. *Neuron*, *26*(3), 703–714.
- Simons, D. J. (2007). Inattention blindness [revision #91372]. *Scholarpedia*, *2*(5), 3244. <https://doi.org/10.4249/scholarpedia.3244>
- Simons, D. J. (2000). Attentional capture and inattention blindness. *Trends in cognitive sciences*, *4*(4), 147–155.

Response to the reviewers

We sincerely thank the reviewers for their time, careful evaluation, and constructive feedback throughout the review process. Their insights have been invaluable in improving our manuscript, and we are grateful for their role in its eventual acceptance.

Response to Reviewer #1:

We thank the reviewer for their thoughtful and balanced assessment of our work. We also value the reviewer's perspective on the limitations of current deep neural network approaches in explaining the neural mechanisms of visual attention. We agree that bridging the gap between data-driven models and biologically grounded explanations remains an important direction for future research, and we hope our work positively contributes in the future of the field.

Response to Reviewer #3:

We are grateful for the reviewer's positive evaluation. We sincerely appreciate their comments and feedback that helped us better evaluate the model's multitask capacity and limitations. We also value their attention and feedback on the code availability and functionality, and we share their enthusiasm that making our models available can facilitate further progress by the broader research community.

Best regards,

Saeed Salehi